# Deep-Pathfinder: A boundary layer height detection algorithm based on image segmentation

Jasper S. Wijnands[1], Arnoud Apituley[1], Diego Alves Gouveia[1], and Jan Willem Noteboom[1]

[1]Royal Netherlands Meteorological Institute (KNMI), De Bilt, The Netherlands

**Correspondence:** Jasper S. Wijnands (jasper.wijnands@knmi.nl)

**Abstract.** A novel atmospheric layer detection approach has been developed based on deep learning techniques for image segmentation. Our proof-of-concept estimated the layering in the atmosphere, distinguishing between pollution rich layers closer to the surface and cleaner layers aloft. Knowledge of the spatio-temporal development of atmospheric layers, such as the mixing boundary layer height (MBLH), is important for the dispersion of air pollutants and greenhouse gases, and for assessing the performance of numerical weather prediction systems. Existing lidar-based layer detection algorithms typically do not use the full resolution of the available data, require manual feature engineering, often do not enforce temporal consistency of the layers, and lack the ability to be applied in near real-time. To address these limitations, our Deep-Pathfinder algorithm represents the MBLH profile as a mask and directly predicts it from an image with backscatter lidar observations. Deep-Pathfinder was applied to range-corrected signal data from Lufft CHM15k ceilometers at five locations of the operational ceilometer network in the Netherlands. Input samples of $224 \times 224$ pixels were extracted, each covering a 45-minute observation period. A customised U-Net architecture was developed with a nighttime indicator and MobileNetV2 encoder for fast inference times. The model was pre-trained on 19.4 million samples of unlabelled data and fine-tuned using 50 days of high-resolution annotations. Qualitative and quantitative results showed competitive performance compared to two reference methods: the Lufft and STRATfinder algorithms, applied to the same dataset. Deep-Pathfinder enhances temporal consistency and provides near real-time estimates at full spatial and temporal resolution. These properties make our approach valuable for application in operational networks, as near real-time and high-resolution MBLH detection better meets the requirements of users, such as in aviation, weather forecasting and air quality monitoring.

## 1 Introduction

The atmospheric boundary layer (ABL) is the lowest part of the troposphere which is influenced directly by meteorological mechanisms near the surface, including heat transfer, evaporation and transpiration, and terrain induced flow modification (Stull, 1988). Various sub-layers can be identified within the ABL, each with different properties. For example, one of the processes observed in the ABL is the vertical mixing of air. In our study, the same definition to characterise layers is used as described in Kotthaus et al. (2023). The term mixing boundary layer (MBL) is used to refer to the ABL sub-layer closest to the ground. Its height (MBLH) may indicate either the convective boundary layer (CBL) height or stable boundary layer (SBL) height, whichever is present at the given moment. The MBLH terminology is applied when no information on atmospheric

stability is available to differentiate between SBL and CBL (Kotthaus et al., 2023, p. 435). MBLH is not constant, but varies throughout the day and can range from less than 100 metres to a few kilometres, depending on the surface sensible heat flux, temperature difference over the inversion layer at the top, temperature gradient of the atmosphere above the mixing layer, and subsidence (Ouwersloot and Vilà-Guerau de Arellano, 2013). Accurate estimates of the MBLH are important for several applications and purposes. For example, a shallow mixing layer results in a larger concentration of air pollutants near the surface, affecting population health through increased risk of respiratory diseases. Similarly, MBLH also affects the dispersion of greenhouse gases throughout the low atmosphere for which accurate estimates are needed. Further, research on the parametrisation of the atmosperhic boundary layer remains essential to the improvement of numerical weather prediction (NWP) systems (Edwards et al., 2020). Therefore, the availability of reliable MBLH estimates would be useful to test and improve NWP accuracy.

The MBLH is not easily and accurately identified in real-time. Existing methods for MBLH detection are commonly based on (i) thermodynamic, (ii) wind and turbulence, or (iii) aerosol characteristics (Kotthaus et al., 2023). These methods are complementary, as each approach has different advantages and limitations for capturing certain features of the MBLH (Collaud Coen et al., 2014). Further, Milroy et al. (2012) showed the inherent link between thermodynamic and aerosol backscatter profiles based on a comparison of remote and in-situ data. Thermodynamic methods, such as the parcel method (Holzworth, 1964) and the bulk Richardson method (Vogelezang and Holtslag, 1996), use temperature and humidity profiles. These methods provide good baseline performance and are frequently used as a reference for the development of new methods. Methods based on wind or turbulence attempt to measure the height of the layer where buoyancy-driven or shear-driven turbulence takes place. The measurement of wind and turbulence can be performed using various instruments, including sodars, radar wind profilers and Doppler lidars. Finally, aerosol-based MBLH methods attempt to observe the result of the mixing process via a proxy, as aerosol properties of different atmospheric layers can vary due to recent mixing processes. Hence, the boundaries of atmospheric layers can be approximated by detecting changes in aerosol properties. Generally, a rapid drop in aerosol content occurs beyond the top of the mixing layer. This phenomenon can be observed using ceilometers that employ the lidar (light detection and ranging) measurement principle, emitting short laser pulses and measuring the back-scattering by aerosols to eventually support estimating particle concentrations at different altitudes. The laser beam and field of view of the receiver telescope often have a no-overlap area, resulting in a blind zone at low altitudes. Further, many lidar systems have an incomplete-overlap region, which can in principle be corrected (Hervo et al., 2016). Possibilities and limitations depend mainly on the instrument optical design, but estimating the height of shallow mixing layers can be challenging. Aerosol-based detection is also difficult when no or limited differences in aerosol content exist between the mixing layer and the layer directly above. Finally, with high relative humidity aerosol particles grow in size, leading to increased backscatter which may result in a layer detection where no layer is (i.e., faulty layer attribution). The high humidity layer also does not necessarily coincide with the stable nighttime surface inversion, meaning MBLH retrieval during night by use of ceilometer backscatter data can be strongly biased. Further investigation of these mechanisms is beyond the scope of this paper. For further reading, refer to Kotthaus et al. (2023, section 3.3.2 and references therein) or Collaud Coen et al. (2014). A variety of methods has been developed that derive MBLH from backscatter profiles. For example, regions showing substantial change in the attenuated backscatter have been detected based

on negative vertical gradients and inflection points (e.g., Sicard et al., 2006), wavelet covariance transform (e.g., Cohn and Angevine, 2000), and spatio-temporal variance (e.g., Menut et al., 1999).

Not all methods for MBLH detection take the temporal progression of the MBLH into account. Point-based detection models (i.e., at a specific time) have the advantage that more labelled data is available for model fitting. However, these methods occasionally experience sudden jumps in the MBLH profile from one layer to another. This undesirable behaviour could be reduced by setting limits for the maximum altitude difference between successive MBLH estimates (Martucci et al., 2010). Several other methods addressed this issue by reinforcing temporal consistency through path optimisation mechanisms from graph theory, initially developed in Pathfinder (de Bruine et al., 2017), and subsequently further enhanced, such as in PathfinderTURB (Poltera et al., 2017), CABAM (Kotthaus and Grimmond, 2018) and STRATfinder (Kotthaus et al., 2020). For these type of approaches, it is common to reduce the temporal resolution of the input data to one or two-minute segments. This reduces noise in the data, which is important for these gradient-based methods.

Some studies have developed approaches based on machine learning to further improve detection accuracy. For example, unsupervised methods such as cluster analysis have been used to detect the boundary layer based on backscatter data (Toledo et al., 2014). Further, Rieutord et al. (2021) compared the use of k-means clustering and AdaBoost. The accuracy of these two approaches varied substantially across measurement sites. However, the (initial) application of the machine learning methods showed potential and various suggestions for future research were made to further improve performance. Min et al. (2020) applied clustering algorithms for post-processing the results of several existing MBLH detection algorithms. Further, Allabakash et al. (2017) used fuzzy logic to combine the range-corrected signal-to-noise ratio, the vertical velocity, and the Doppler spectral width of the vertical velocity to identify MBLH from a radar wind profiler. Bonin et al. (2018) also applied fuzzy logic to combine data from different scanning strategies of a Doppler lidar, determining where turbulent mixing is present. Several studies also used techniques from computer vision, such as edge detection, to identify layer boundaries (e.g., Haeffelin et al., 2012; Patel et al., 2021; Vivone et al., 2021).

Various studies have combined remote sensing information with other atmospheric variables. For example, gradient boosted regression trees were used by de Arruda Moreira et al. (2022) to predict the MBLH estimated with microwave radiometer data based on the MBLH estimated with ceilometer data and several atmospheric variables. Krishnamurthy et al. (2021) used the random forest algorithm to combine Doppler wind lidar MBLH estimates using the method by Tucker et al. (2009) with various meteorological measurements such as surface relative humidity, air temperature, soil moisture, and turbulence kinetic energy. These approaches have been shown to generally improve prediction accuracy, although the use of multiple data sources may complicate the large-scale implementation in existing real-time detection networks, such as E-Profile (Haefele et al., 2016).

In summary, several methodological challenges still remain. Few methods incorporate temporal information to avoid jumps between layers. This is an issue for some of the machine learning methods described above. In contrast, many methods that are not based on machine learning require expert knowledge to manually set modelling thresholds (e.g., for nighttime detection, instrument- and site-specific tuning). This also extends to the manual specification of guiding restrictions for layer selection. An open research question for MBLH detection is how to combine the advantages of different methods in a single approach. In particular, it would be beneficial to (i) promote temporal consistency of the MBLH profile, (ii) use the full resolution of

the ceilometer, and (iii) limit manual feature engineering, specification of rules for layer selection, and site-specific tuning parameters. Further, not all existing methods can be used for real-time detection, which is an important quality for operational use. These challenges form the basis for the Deep-Pathfinder MBLH detection algorithm described in this paper.

## 2 Materials and methods

Deep-Pathfinder has a similar goal to other algorithms that attempt to find the path the MBLH follows based on ceilometer observations (e.g., Pathfinder, PathfinderTURB), although using a completely different approach based on deep learning. Our study proposes to process lidar data from ceilometers using computer vision techniques for image segmentation. Image segmentation has been used in many domains, including scene understanding for autonomous vehicles (Guo et al., 2021) and medical image analysis to detect various types of cancer (e.g., Dong et al., 2017). The concept of the new Deep-Pathfinder al-

gorithm is to represent the 24-hour MBLH profile as a mask (i.e., black indicating the MBL, white indicating the residual layer or free troposphere) and directly predict the mask from an image with range-corrected ceilometer observations (see Fig. 1). This promotes temporal consistency of MBLH estimates, while using the maximum resolution of the ceilometer. It also limits manual feature engineering, meaning there is no need to explicitly define the important characteristics that will be used to create the MBLH profile (e.g., which values represent a cloud).

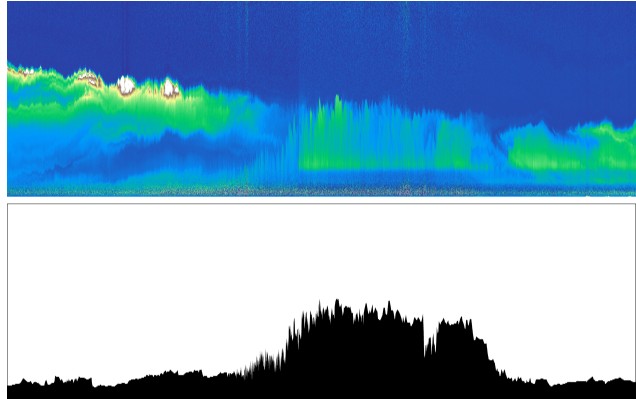

**Figure 1.** Concept of the Deep-Pathfinder model, showing an input image of the range-corrected lidar signal (top) and a corresponding mixing layer mask (bottom). The plot axes and colour scale are not supplied to the model and have therefore been omitted.

Machine learning can estimate the link function between input and output images from historical data. Given the large-scale availability of ceilometer data and the high temporal and spatial resolution at which it is recorded, this domain is very suitable for machine learning approaches such as deep learning. However, the main challenge for a deep learning approach is the limited availability of annotated data. In particular, annotated data is generally not available for extended consecutive time periods, except for MBLH estimates generated by other methods (e.g., ECMWF reanalysis). Further, annotated data is laborious to

obtain, especially at high temporal resolution. Therefore, our research aimed to extract domain knowledge from unlabelled data, reducing requirements for the amount of annotated data.

## 2.1 Data

A large dataset with unlabelled ceilometer data in NetCDF format (∼125 GB) was downloaded from the KNMI Data Platform (https://dataplatform.knmi.nl/). This dataset contained backscatter profiles from ceilometers in the KNMI observation network
and the full dataset available at the time of this research was obtained, containing measurements from June 2020 to February 2022. Data was available at five locations in the Netherlands: Cabauw, De Kooy, Groningen Airport Eelde, Maastricht Aachen Airport and Vlissingen. Throughout the observation period, each location operated a CHM15k ceilometer from manufacturer Lufft, which is a one-wavelength backscatter lidar at 1064 nm. The ceilometers recorded data continuously at 12-second temporal and 10-meter vertical resolution. For the vertical profile the 'beta_raw' variable was used (firmware v1.010 and
v1.040) as it readily provided a normalised range-corrected signal (RCS), with instrument-specific overlap correction (usable above 80–100 metres altitude) and harmonisation between different CHM15k systems. Note that prior research has indicated that the built-in overlap correction of the CHM15k is not perfect (Hervo et al., 2016). The region of incomplete overlap is generally an issue to be considered in lidar applications, in particular when the lidar data is used to derive or estimate physical quantities that rely on the optical parameters included in the lidar data, such as optical extinction. However, the Deep-Pathfinder
approach does not rely on optical information, as is explained in the following sections. Due to the training process, Deep-Pathfinder implicitly takes the incomplete overlap into account and does not require an additional overlap correction to be applied before analysis (see Hervo et al., 2016).

Model pre-training (see Sect. 2.4) used all ceilometer data from Cabauw, De Kooy, Groningen Airport Eelde, Maastricht Aachen Airport and Vlissingen. Model fine-tuning used a subset of data from Cabauw, De Kooy and De Bilt and some days
outside the June 2020 to February 2022 period (e.g., for validation), which were manually annotated. The majority of annotated data was from Cabauw, located in the western part of the Netherlands (51.971° N, 4.927° E) at 0.7 m below mean sea level. At this site a large set of instruments is operated to study the atmosphere and its interaction with the land surface (see Fig. 2). Here, KNMI and partners have been carrying out a continuous measurement program for atmospheric research since 1972 (see https://cabauw.knmi.nl). The measurement program has evolved with increasingly advanced measurement tech-
niques and instruments, often in collaboration with other institutes and universities. For this study, the CHM15k ceilometer was used together with ancillary data such as relative humidity (RH) information that was collected at the 10, 20, 40, 80, 140 and 200 m levels of the 213 metres high meteorological tower (Bosveld, 2020). Other data sources for annotation included MBLH information from ECMWF, based on bulk Richardson number (ECMWF, 2017), which was obtained via the Cloudnet model output for Cabauw (CLU, 2022) and indicated the general atmospheric conditions. Further, MBLH estimates from the
manufacturer's layer detection algorithm were included for reference purposes. This proprietary algorithm, based on wavelet covariance transform, identified multiple candidate layers from the RCS data, where the lowest identified layer can typically be interpreted as the MBLH. Model evaluation used MBLH estimates from two reference methods: (i) Lufft's wavelet covariance transform algorithm and (ii) a state-of-the-art detection algorithm, STRATfinder (Kotthaus et al., 2020). At the time of

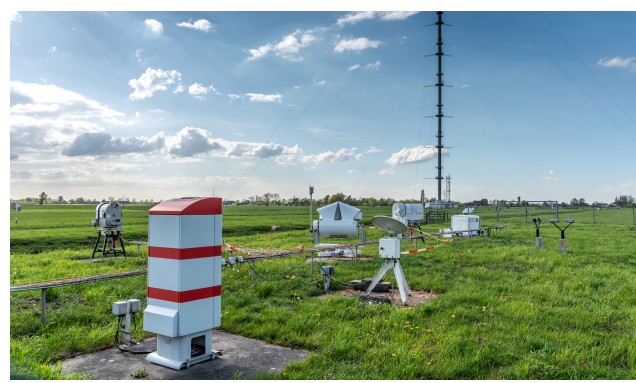

**Figure 2.** The ceilometer (in front) and 213-meter mast (background) at the same location at Cabauw support high-quality annotations. This site also includes a scanning microwave radiometer (left), a scanning cloud radar, a micro rain radar and a second microwave radiometer (center), a scanning Doppler wind lidar and two distrometers (right).

publication, STRATfinder was still in active development and the STRATfinder data for this study was received from Institute Pierre Simon Laplace (IPSL) on 6 May 2022.

## 2.2 Pre-processing and annotation

Various pre-processing steps were performed on the ceilometer data, using Python and OpenCV (Bradski, 2000). First, the RCS was capped to [0, 1e6] $m^2$ counts $s^{-1}$ and rescaled to [0, 1]. This corresponds to an attenuated backscatter interval of about [0, 2.4–3.4] $Mm^{-1}$ $sr^{-1}$, depending on the specific ceilometer used and its respective calibration constant provided by E-Profile (Wiegner and Geiß, 2012; Haefele et al., 2016). The spatial range was cropped to a maximum altitude of 2240 meter, while retaining the 10-meter spatial resolution. A total of 7200 time steps was selected using the original temporal resolution of 12 seconds, capturing a 24-hour period of data. The resulting data was stored as a 16-bit grayscale image with 224×7200 pixels.

The software package `labelme` (Wada, 2022) was used for manually annotating a small part of the data (i.e., layer attribution). This tool enabled the creation of custom masks for image segmentation, including the export of selected points to JSON format. Specifically, for a single day of data, many consecutive points were selected to follow small-scale changes such as convective plumes, intrusions and extrusions visible in the RCS data. The resulting JSON data was converted to a black and white mask at the same 224×7200 pixels resolution as the input image. High-resolution annotations were created for 50 days in 2019, 2020 and 2021. Representative cases under a variety of atmospheric conditions were selected for annotation to cover a broad range of boundary layer dynamics. The captured atmospheric conditions included clear days with a distinct CBL, cold days with a shallow boundary layer, days with cloud cover, days with multiple cloud layers, and precipitation events.

The annotation process started with a visual inspection of the RCS data and corresponding gradient fields (see Fig. 3). Gradient estimation identified the location of layer boundaries in a consistent manner, leading to several candidate layers at some time steps. This information on potential layer boundaries was combined with thermodynamic information and manufacturer

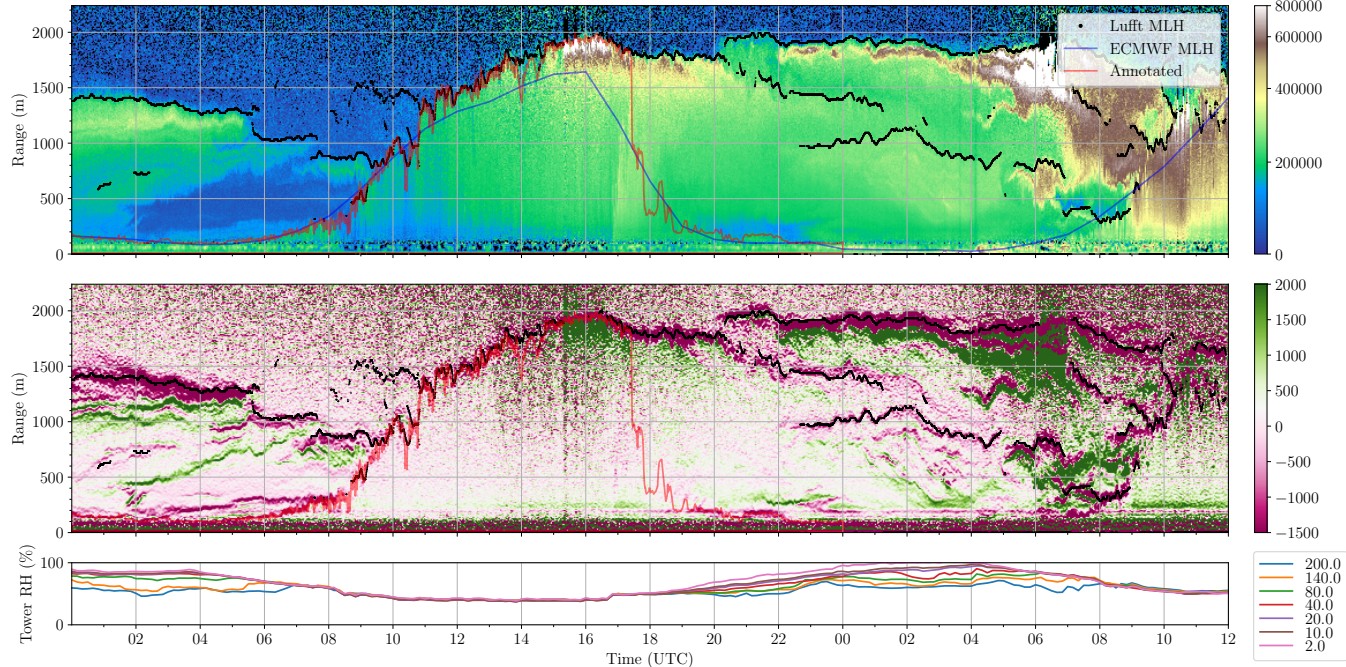

**Figure 3.** Combining data sources for 8 June 2021 (left part of the plot) and the first half of 9 June 2021, to annotate the MBLH on 8 June. The top plot shows the RCS data with layers detected by the ceilometer (black lines), the thermodynamic MBLH from ECMWF (blue line) and the annotated MBLH (red line). The middle plot shows the negative (magenta) and positive (green) gradients of the RCS data, with layers detected by the ceilometer (black lines) and the annotated MBLH (red line). The bottom plot contains the measured relative humidity by the Cabauw mast at different altitudes (various colours). Sunrise was at 03:21 UTC and sunset at 19:55 UTC.

MBLH estimates to enhance the layer selection process. Note that the attenuated backscatter retrievals during nighttime are not equivalent to the SBL or nocturnal boundary layer (NBL) defined by thermodynamics. Our MBLH definition (see Sect. 1) has been adopted for consistency with existing methods for MBLH detection and for comparison of results with these reference methods.

        The layer attribution started with identification of the nocturnal layer structure before sunrise by tracking backwards in time
from the high-confidence MBLH identified after sunrise. At this point, the RH profile confirmed when the unstable boundary layer rose above 200 m (which generally coincided with the convective ejection patterns in the RCS). During nighttime, the pollution rich layers may drop to very low altitudes into the incomplete-overlap region of the CHM15k ceilometer and vertical mixing in fact predominantly ceases to exist. However, the concentration levels of pollutants remain layered and, therefore, Cabauw mast measurements were used to aid in the identification of the presence and height of nocturnal layers. When RH
values at various heights were similar, we considered this to be a more or less homogeneous nocturnal layer, and the layer height was approximated as the altitude where the RH values of higher layers with respect to the lower layers started to diverge. For example, Fig. 3 indicates a MBLH between 40 and 140 meters around 4am UTC on 8 June, which slowly rose to 200 meters

around 7am. Note that for annotated layer heights below 100 metres it would be more difficult for the detection algorithm to recover the exact layer height, because of the no-overlap region of the ceilometer. If the MBLH of the following day was clearly visible in RCS data, it was followed backward in time to identify its formation after sunset.

Under less complex atmospheric conditions, the CBL plumes can often be identified in great detail from sharp layer edges. To take advantage of the high temporal and spatial resolution of the RCS, the MBLH under such conditions was annotated following the ejection patterns inside the entrainment zone (EZ). Hence, the average MBLH was located somewhere inside the EZ, while the amplitude between the local minimum and maximum MBLH provided an indication of its thickness. The CBL was usually confidently annotated as the MBLH until late afternoon or early evening, when the boundary layer was fully developed. For cases with convective clouds forming on the top of the boundary layer or low stratiform clouds with no clear aerosol layer underneath, the apparent cloud top height was annotated as the MBLH. Periods with rain were annotated with a value of 0, as the MBLH is undefined during precipitation.

The transition region from the daytime convective (mixed) boundary layer to a neutrally stratified residual layer (RL) with a stable NBL below, may not be clear from aerosol data (Schween et al., 2014). To complete the annotations in this region, the thermodynamic data informed the gradual decline towards the nighttime MBLH. The annotators looked for a path (layer edge), albeit with low gradient, that could connect the closest previously annotated neutrally stratified RL to the daytime CBL. An example of this process can be seen in Fig. 3 between 4pm and 8pm. If the aerosol profile was too smooth, a sudden jump to the nighttime MBLH was annotated.

Due to the physical processes leading to vertical mixing, the RCS profile shows distinct differences during the day and at night. To differentiate between the SBL and CBL, a nighttime variable was included. Specifically, sunrise and sunset times in UTC were computed for the corresponding date and stored with the images using a nighttime variable. This assists the model in distinguishing whether an estimate of the SBL or CBL is expected. In summary, one labelled sample consisted of a 24-hour pre-processed RCS image, a nighttime variable, and a corresponding annotated mask. All samples were converted to TensorFlow's TFRecord format for modelling purposes (Abadi et al., 2015).

## 2.3 Model architecture

The Deep-Pathfinder algorithm is based on the U-Net architecture (Ronneberger et al., 2015), which is a frequently used model for image segmentation tasks. U-Net extracts features from an input image using consecutive convolutional layers (i.e., the encoder). From the latent space representation (i.e., a representation of compressed data), the dimensions are increased again to obtain an output mask (i.e., the decoder). Skip connections connect corresponding layers in the encoder and decoder to increase details in the generated mask. This generic U-Net architecture was adapted to the task of MBLH detection. The encoder was based on MobileNetV2 (Sandler et al., 2018), which was originally developed for constrained compute environments such as mobile and embedded devices. This was chosen to ensure fast inference times for potential operationalisation, as MobileNetV2 was developed specifically for low latency inference. The input dimensions of the RCS image were 224×224 pixels, representing a time period of 44 minutes and 48 seconds and a fixed altitude range of 0 to 2240 meters, preserving the temporal and spatial resolution of the ceilometer data. Using a sequence of MobileNetV2 layers, a 7×7 block with 320

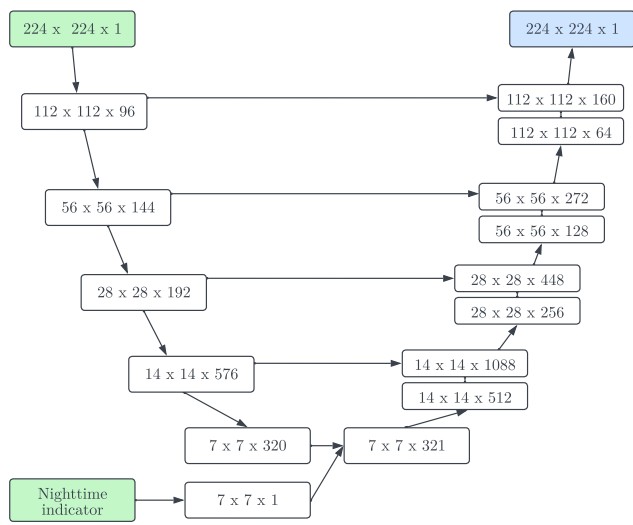

**Figure 4.** Deep-Pathfinder neural network architecture with input (green), output (blue), skip connections (horizontal arrows), encoder (downward arrows) and decoder (upward arrows from $7\times7\times321$). In each box, the first two numbers indicate the spatial dimensions and the third indicates the number of channels, determining the capacity of the neural network layer.

features was extracted from the input image. Further, the U-Net architecture was adapted to incorporate different boundary layer dynamics before and after sunset. Specifically, a nighttime indicator was added to the extracted features as an additional channel, indicating whether the sample mainly occurred inside or outside the sunrise to sunset window for the date and time of the sample. This resulted in a latent space with dimensions $7\times7\times321$. The architecture used several transposed convolutional layers to decode the latent space and obtain a $224\times224$ pixels output image. Layer depths (i.e., the number of features per layer) were obtained through experimentation. For each pixel a single output value was produced, representing (i) the RCS value during pre-training, or (ii) an MBL indicator during transfer learning. A graphical representation of the neural network architecture is presented in Figure 4.

## 2.4 Model calibration

The model calibration process consisted of two steps: (i) pre-training and (ii) fine-tuning. While annotated data is laborious to obtain, unlabelled data contains readily available and valuable information on the typical patterns observed in lidar signals. Therefore, part of the network was pre-trained using all data to aid the calibration on limited annotated samples afterwards. The unsupervised pre-training task was to reconstruct the ceilometer data, meaning the input RCS image was also used as the target image. This was implemented by removing the skip connections and nighttime indicator from the neural network architecture to create an autoencoder with an equivalent structure. Removing the skip connections was necessary as information would otherwise flow directly from input to output without passing through the encoder/decoder structure, which is undesirable behaviour for the pre-training task. Unlabelled ceilometer data (see Sect. 2.1) was used to train this autoencoder.

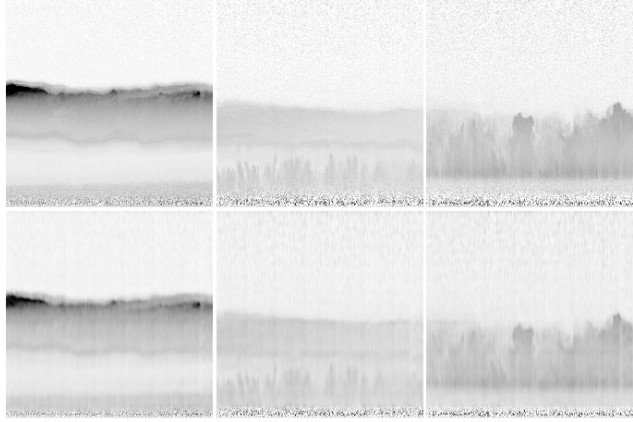

**Figure 5.** Visualisation of the completed pre-training process, showing in each column a grayscale input image (top) and the image reconstructed by the autoencoder (bottom).

In total, 19.4 million different samples of 224×224 pixels were extracted from the unlabelled data through cropping. Given the temporal resolution of 12 seconds, a total of 6976 different images can be extracted from a full day of data. Model calibration was performed with TensorFlow r2.6 using NVIDIA A100 GPUs of the Dutch National Supercomputer Snellius. The binary crossentropy loss function was used for model calibration in combination with the Adam optimizer (Kingma and Ba, 2014) with a learning rate of 1e-3. After approximately 11 million iterations (about five days), the model reached convergence with a loss close to 0 for the image reconstruction task (see Fig. 5). The pre-training task resulted in a calibrated encoder and decoder network. Only the encoder weights were retained to initialise the Deep-Pathfinder architecture with skip connections, extracting valuable features from RCS images without the use of annotated data.

Subsequently, transfer learning was used to fine-tune the pre-trained model for the task of mask prediction. Transfer learning is a learning paradigm for adapting a model to perform a similar task, exploiting the knowledge captured during the previous task. For the mask prediction task, the neural network model now had to predict an annotated mask of 224×224 pixels. The model was provided with both the RCS image and the nighttime indicator as inputs, while the corresponding annotated mask was used as the target image. For illustration purposes, Fig. 6 shows three sample training image pairs. Samples were grouped in batches for computation purposes. These batches were constructed by randomly selecting a day from the training data and subsequently extracting a random batch of 16 image pairs of 224×224 pixels from the full 24-hour RCS image and corresponding annotated mask. Although optimal randomisation would be obtained if one batch contained samples from various days, this implementation choice resulted in limited pre-processing overhead and full GPU utilisation, while still iterating through the full set of annotated images. As 50 days of ceilometer data were annotated, the training set contained approximately 350,000 samples to fine-tune the deep learning model. Typically, an experiment required less than 50 epochs of training on labelled data for transfer learning. A small validation dataset for one additional annotated day ($n = 1396$ samples, including 765 daytime and 631 nighttime images) was used to tune model hyperparameters, such as the learning rate, batch size, and layer depth multiplier. The validation set did not contain data from any of the days present in the training set. The

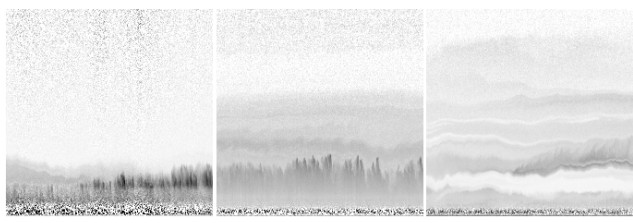

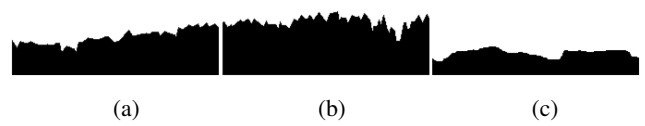

(a)       (b)       (c)

**Figure 6.** Three samples of image pairs for model fine-tuning, consisting of a grayscale input image (top), a nighttime indicator (not shown) and a manually annotated output mask (bottom). The presented samples are from Cabauw on (a) 15-Sep-2019 at 8:49am–9:34am UTC, (b) 03-Sep-2021 at 9:33am–10:18am UTC, and (c) 14-Jun-2021 at 9:46pm–10:31pm UTC.

main selection criterion for model evaluation was the mean accuracy of the generated masks in the validation set, providing a quantitative scoring mechanism for different experiments. In addition, predictions for a full validation day were visualised by creating a 24-hour prediction mask (see Sect. 2.5) to get additional insights on model behaviour. For example, models with comparable validation accuracy could show differences in smoothness of the decline in MBLH estimates after sunset. This

qualitative information provided secondary input in the model selection process, after candidate models were selected based on high validation accuracy. Finally, a test set was constructed with ceilometer data not used for mask prediction, spanning the second half of 2020 at Cabauw, to obtain the unbiased performance of the final tuned model.

## 2.5 Model inference and post-processing

Model inference using the calibrated model resulted in a generated mask with for each pixel a prediction of (near) white or

265 black, or a grey tone in case of uncertainty. The generated masks correspond to a 45-minute period, rather than providing a 24-hour sequence of MBLH values. Therefore, several post-processing steps were required. First, multiple output masks were generated via model inference. Specifically, the input image was repeatedly shifted by five pixels (i.e., one minute) and processed by the model, until the entire day was processed. This led to overlapping predictions for each time step. Multiple predictions at each unique time and altitude combination were arithmetically averaged to obtain a full 24-hour mask. Sub-

270 sequently, an MBLH profile was extracted from the 24-hour mask using the following method. Each predicted mask was processed column-wise, identifying the MBLH at time step $t$ independent of other time steps. A loss function was formulated to evaluate the plausibility of every possible pixel $p \in \{1, 2, \ldots, 224\}$ to represent the MBLH, for a fixed time $t$. Note that model predictions display varying degrees of uncertainty, with pixel predictions being either black, white or any grey tone in between (i.e., the result of the pixel-wise sigmoid function at the end of the neural network). The loss function took into account model

uncertainty by proportionally penalising the number of pixels below $p$ that were predicted as non-black, plus the pixels above $p$ that were predicted as non-white. MBLH at time $t$ was estimated as the value $\hat{p}$ that minimised this loss, multiplied by the spatial resolution of 10 metres.

## 3 Results

### 3.1 Qualitative assessment

Deep-Pathfinder performance was compared to MBLH estimates from STRATfinder and manufacturer Lufft for all days in the test set, showing the out-of-sample performance on new data not used for model calibration. The selected days in Fig. 7 contain varying conditions, including the typical growth of the CBL during the day, periods of precipitation, low clouds, hardly visible decay after sunset, multiple cloud layers, and a day without strong convection. The CBL was typically captured well by all three methods, with minimal differences in MBLH between them. In Figs. 7a and 7b, before sunrise, the Lufft algorithm 285 jumped between several aerosol layers which developed inside the residual of the mixing layer of the day before. Deep-Pathfinder and STRATfinder identified the nighttime MBLH around 100–200 meters altitude, although STRATfinder estimates were at a constant level slightly above the actual MBLH due to guiding restrictions in the algorithm. Instead, Deep-Pathfinder was still able to process the noisy signal in the incomplete-overlap region. Another difference between Deep-Pathfinder and STRATfinder was that Deep-Pathfinder followed short-term fluctuations in MBLH more closely than STRATfinder due to 290 the use of high-resolution input data. All algorithms had difficulties capturing the decline in MBLH around sunset, which is a typical limitation for MBLH detection based on aerosol observations (see Wang et al., 2012; Schween et al., 2014). For example, a sudden jump in MBLH is visible in Fig. 7b for both Deep-Pathfinder and STRATfinder, although at a different time.

For complex atmospheric conditions, a considerable amount of MBLH estimates of the Lufft algorithm were missing due to quality control flags. An example is provided in Fig. 7c, where Lufft estimates were only available after 8pm UTC and 295 not during low cloud conditions. In most cases, Deep-Pathfinder and STRATfinder were still able to provide appropriate MBLH estimates. In a few cases, STRATfinder predictions were missing due to quality control flags (e.g., Fig. 7d). During the precipitation event in Fig. 7a around 7 to 9pm UTC, Deep-Pathfinder has been trained to predict 0 (i.e., not applicable), while Lufft predictions jumped to about 2,500 meter altitude. The example of Fig. 7e shows that for multiple cloud layers Deep-Pathfinder and STRATfinder typically followed a different layer. Hence, in case of multiple cloud layers, users should be aware 300 that the methods may produce different MBLH estimates. When a clear CBL was not apparent (e.g., Fig. 7f), Deep-Pathfinder and STRATfinder obtained similar estimates, although in Fig. 7f both were far above the stable nighttime surface inversion.

### 3.2 Correlation analysis

A statistical assessment of overall agreement between the algorithms was performed through a correlation analysis. For the July to December 2020 test period, the Pearson correlation was computed between the time series of each pair of algorithms. 305 Deep-Pathfinder and STRATfinder scored an average correlation of 0.706, based on 250,000 corresponding records of data

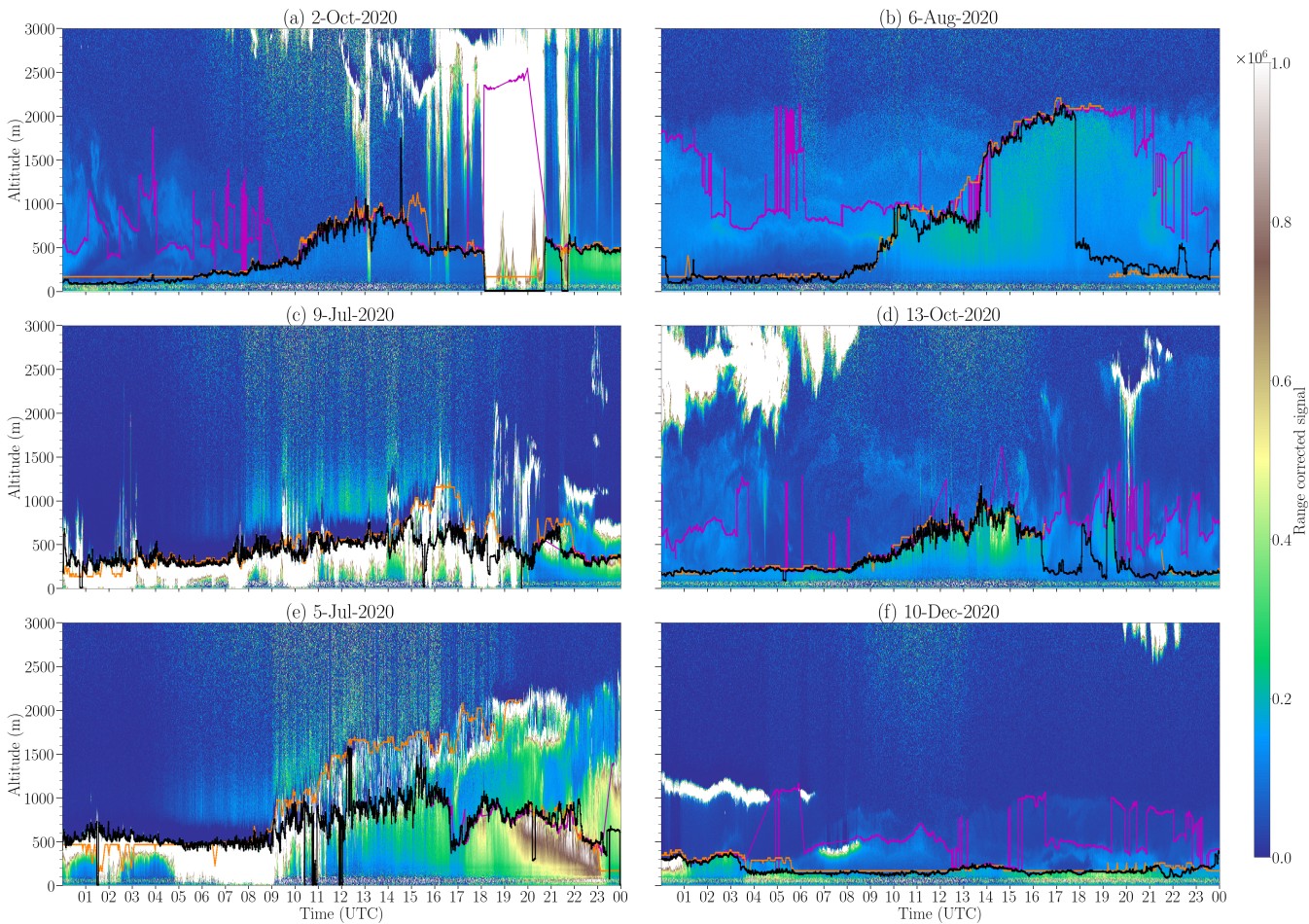

**Figure 7.** Performance comparison of Deep-Pathfinder (black), STRATfinder (orange) and Lufft (purple) on selected days at Cabauw.

(see Table 1). In contrast, the Lufft algorithm obtained a substantially lower correlation with either method. Alignment of the algorithms was also not constant throughout the test period. Table 2 shows the distribution of the number of days the correlation was in a pre-defined range, indicating that on the majority of days the correlation between Deep-Pathfinder and STRATfinder was within $[0.6, 0.8)$ or $[0.8, 1]$. These daily fluctuations can be partly explained by the amount of cloud cover. To illustrate

this point, the daily cloud overcast fractions were computed for all dates in the test set, looking only for clouds below 2245 meters (i.e., the vertical range captured by our model). Table 3 shows that on days with no or few clouds the Deep-Pathfinder and STRATfinder algorithms were more closely aligned, based on Pearson correlation and mean absolute difference statistics. Note that achieving the highest possible correlation was not the goal of our study, as otherwise STRATfinder annotations could have been used for model training. This would have led to better alignment between the methods, although without the

capability to create high-resolution predictions.

**Table 1.** Pearson correlation between the time series of Deep-Pathfinder, STRATfinder and Lufft in the July to December 2020 test set.

|  | Deep-Pathfinder | STRATfinder | Lufft |
|---|---|---|---|
| Deep-Pathfinder | 1 |  |  |
| STRATfinder | 0.706 | 1 |  |
| Lufft | 0.425 | 0.325 | 1 |

**Table 2.** For each pair of algorithms, this table lists the total number of days that the Pearson correlation was within the specified ranges.

| Correlation | Deep-Pathfinder vs. STRATfinder | Deep-Pathfinder vs. Lufft | STRATfinder vs. Lufft |
|---|---|---|---|
| $[-1.0, -0.8)$ | 0 | 0 | 0 |
| $[-0.8, -0.6)$ | 0 | 4 | 5 |
| $[-0.6, -0.4)$ | 0 | 2 | 7 |
| $[-0.4, -0.2)$ | 1 | 10 | 19 |
| $[-0.2, 0.0)$ | 5 | 20 | 27 |
| $[0.0, 0.2)$ | 13 | 27 | 19 |
| $[0.2, 0.4)$ | 20 | 22 | 20 |
| $[0.4, 0.6)$ | 27 | 40 | 29 |
| $[0.6, 0.8)$ | 48 | 25 | 16 |
| $[0.8, 1.0]$ | 47 | 7 | 11 |

### 3.3 Diurnal MBLH patterns

The performance of the MBLH detection methods across different seasons can provide insights related to model robustness in terms of showing expected behaviours. Fig. 8 shows the mean MBLH estimates throughout the day for the different algorithms, for each month in the test set. The interquartile range (i.e., $25^{th}$ to $75^{th}$ percentile) of the MBLH estimates is also included in this figure. For consistency, the temporal resolution for this analysis was reduced to one minute for all methods. A gradual decline in peak MBLH can be observed from July and August towards December. On some days, STRATfinder reached higher peak values than Deep-Pathfinder. In case of multiple cloud layers, our annotations typically followed the lower layer, while STRATfinder followed the higher layer. For example, this behaviour can be observed in Fig. 7e. Further, Deep-Pathfinder was able to capture low layers at night better than STRATfinder, although it was also limited by the no-overlap region of the ceilometer.

Importantly, performance varied for the different phases of the diurnal development of the ABL. In the early morning, the ML grows into a stably stratified, unmixed NBL at the surface (i.e., roughly 4–8h UTC). This is typically followed by fast

**Table 3.** Comparison of Deep-Pathfinder and STRATfinder estimates for different ranges of cloud cover in the test set.

|  | Overall | Cloud overcast range | | |
|---|---|---|---|---|
|  |  | 0–10% | 10–30% | 30–100% |
| Number of days | 161 | 27 | 24 | 110 |
| Pearson correlation | 0.706 | 0.811 | 0.819 | 0.632 |
| Mean absolute difference (m) | 189.0 | 141.6 | 176.7 | 205.3 |

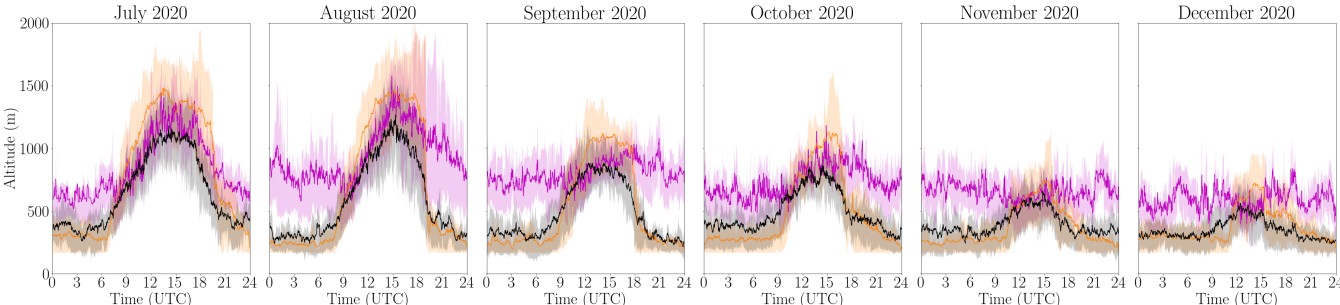

**Figure 8.** Mean diurnal MBLH patterns per month at Cabauw for Deep-Pathfinder (black), STRATfinder (orange) and Lufft (purple), including interquartile ranges.

growth into the more or less neutrally stratified RL (8–12h UTC). In the early afternoon, the fully developed CBL grows slowly into the free troposphere (12–16h UTC). During the late afternoon a more or less sudden breakdown of convective turbulence and thus breakdown of mixing occurs (16–19h UTC). Finally, in the evening and night a new NBL and RL develops (19–4h UTC). Table 4 shows that the mean absolute difference between Deep-Pathfinder and STRATfinder was lowest during the evening, night and early morning growth phases. In contrast, the early and late afternoon is where they were least similar (see explanation above). The Lufft algorithm obtained higher estimates than both other algorithms during the evening, night and early morning growth phases, as it had a tendency to follow aerosol or moist layers in the residual layer (see Sect. 3.1). However, Lufft and Deep-Pathfinder estimates were substantially more similar in the early afternoon. During this period, the manufacturer algorithm can be used more confidently.

### 3.4 Using alternative neural network architectures

The choice of neural network architecture is an important modelling choice in deep learning research. For example, various alternative neural network architectures have been developed based on U-Net. The latest architectures typically obtain higher performance on benchmark datasets than the standard U-Net implementation. Therefore, the performance of some of these new architectures has been investigated, quantifying the impact of the architectural choices on model performance and providing

**Table 4.** Method intercomparison showing the mean absolute difference (metres) for the various phases of development of the ABL, obtained using the test set. Time of day is stated in UTC; note that the local standard time at Cabauw is UTC+1 or UTC+2 (daylight saving time).

| Method | Overall | Time of day (UTC) | | | | |
| --- | --- | --- | --- | --- | --- | --- |
| | | 4–8h | 8–12h | 12–16h | 16–19h | 19–4h |
| Deep-Pathfinder vs. STRATfinder | 189.0 | 127.6 | 178.1 | 251.5 | 316.5 | 156.7 |
| Deep-Pathfinder vs. Lufft | 323.5 | 392.9 | 219.9 | 117.7 | 293.3 | 396.5 |
| STRATfinder vs. Lufft | 369.8 | 437.2 | 285.4 | 220.9 | 306.4 | 435.2 |

insights related to promising directions for future research. Specifically, the performance of Swin-Unet (Cao et al., 2021), UNet 3+ (Huang et al., 2020), Attention U-Net (Oktay et al., 2018), TransUNet (Chen et al., 2021), $U^2$-Net (Qin et al., 2020) and ResUNet-a (Diakogiannis et al., 2020) was investigated. These models have substantial architectural differences. For example,
TransUNet was based on vision transformers, $U^2$-Net used a nested U-structure, and ResUNet-a used residual connections, atrous convolutions and pyramid scene parsing pooling. Model implementations were obtained from the Keras UNet collection (Sha, 2021). After model training, masks were predicted for all samples in the validation set and the MBLH was extracted (see Sect. 2.5). Statistics were computed with respect to the annotations for the validation set, which followed the same annotation process as described in Sect. 2.2. A full evaluation on six months of test data was not performed. However, the results are
indicative of the performance of different model architectures. Note that these experiments did not use any form of unsupervised pre-training. Therefore, the Deep-Pathfinder architecture without pre-training on unlabelled lidar data (referred to as U-Net Nighttime Indicator) was also included for comparison purposes. Further, a simpler architecture without nighttime indicator and no pre-training (U-Net Base) was included, as this indicator was also not implemented for the alternative architectures. Hence, results of the alternative architectures can be directly compared to the U-Net Base model.

Tables 5 and 6 provide the mean absolute error (MAE) and Pearson correlation that were obtained, showing a large variation between different neural network architectures. The best performance was obtained by ResUNet-a and Deep-Pathfinder, followed by $U^2$-Net and TransUNet. Notably, the ResUNet-a architecture obtained better results on the validation set than Deep-Pathfinder. The growth of the MBL in the early and late morning, and the fully developed CBL in the early afternoon were captured best. Correlation was highest during the late morning, when the high temporal resolution forecasts accurately
followed the annotated CBL in the validation set. The breakdown of convective turbulence in the late afternoon (i.e., 16–19h UTC) was the main difficulty the models faced. The correlation analysis also showed that tracking MBLH was more difficult at night, where ceilometer-based detection has its limitations (e.g., see Sect. 2.2).

The U-Net Base architecture performed worse than most of the newer architectures, which was expected due to architectural improvements. Further, both the U-Net Nighttime Indicator and Deep-Pathfinder architectures performed substantially better
than the U-Net Base architecture. This shows the benefits of (i) incorporating sunrise and sunset information explicitly in the model and (ii) unsupervised pre-training on large-scale lidar data to improve feature extraction. For example, overall MAE

**Table 5.** MAE for the MBLH in metres, obtained using the validation set. Neural network architectures have been sorted based on overall score.

| Architecture | Overall | Time of day (UTC) | | | | |
|---|---|---|---|---|---|---|
| | | 4–8h | 8–12h | 12–16h | 16–19h | 19–4h |
| ResUNet-a | 41.3 | 10.2 | 21.6 | 25.1 | 108.8 | 48.5 |
| Deep-Pathfinder | 64.2 | 14.8 | 22.3 | 24.2 | 114.0 | 106.0 |
| U-Net Nighttime Indicator | 74.5 | 16.1 | 32.1 | 23.7 | 162.1 | 112.7 |
| TransUNet | 82.6 | 16.8 | 114.8 | 36.1 | 227.3 | 69.9 |
| $U^2$-Net | 84.8 | 10.2 | 23.8 | 22.5 | 112.9 | 163.3 |
| U-Net Base | 104.5 | 12.8 | 41.9 | 77.1 | 371.9 | 96.2 |
| Attention U-Net | 113.7 | 33.1 | 28.8 | 261.0 | 311.0 | 56.1 |
| UNet 3+ | 125.1 | 27.6 | 40.2 | 291.9 | 321.9 | 66.4 |
| Swin-Unet | 242.7 | 26.5 | 222.8 | 580.4 | 338.1 | 165.7 |

**Table 6.** As Table 5, but for Pearson correlation.

| Architecture | Overall | Time of day (UTC) | | | | |
|---|---|---|---|---|---|---|
| | | 4–8h | 8–12h | 12–16h | 16–19h | 19–4h |
| ResUNet-a | 0.96 | 0.97 | 0.99 | 0.90 | 0.85 | 0.28 |
| Deep-Pathfinder | 0.91 | 0.95 | 0.99 | 0.86 | 0.85 | 0.24 |
| $U^2$-Net | 0.90 | 0.96 | 0.99 | 0.89 | 0.71 | 0.28 |
| U-Net Nighttime Indicator | 0.90 | 0.95 | 0.98 | 0.93 | 0.82 | 0.29 |
| TransUNet | 0.89 | 0.91 | 0.81 | 0.69 | 0.71 | 0.00 |
| Attention U-Net | 0.82 | 0.51 | 0.98 | 0.45 | 0.54 | 0.11 |
| UNet 3+ | 0.79 | 0.65 | 0.95 | 0.44 | 0.59 | 0.15 |
| U-Net Base | 0.78 | 0.94 | 0.95 | 0.65 | 0.42 | 0.19 |
| Swin-Unet | 0.32 | 0.94 | 0.83 | 0.35 | 0.75 | 0.22 |

decreased from 104.5 metres for the U-Net Base model to 74.5 and 64.2 metres for the U-Net Nighttime Indicator and Deep-Pathfinder architectures, respectively.

## 4 Discussion

### 4.1 Annotations and model robustness

Labelling MBLH is a complex and time-consuming task. Further, deep learning methods typically require a very large number of labelled samples. This combination of factors complicates the development of machine learning approaches for MBLH detection. The issue of obtaining sufficient training samples was adressed in our study by unsupervised pre-training and extracting many 45-minute samples from a 24-hour mask through random cropping. Further, image segmentation architectures can be trained using relatively few annotated samples (Ronneberger et al., 2015), making it a suitable approach for this particular application.

Annotating MBLH with high temporal resolution has several advantages. For example, the model will become more responsive to observed changes in MBLH. Further, short-term fluctuations in MBLH could be used to provide an estimate of the thickness of the entrainment zone (Cohn and Angevine, 2000), which cannot be provided by many algorithms. Combining measurements from different sensors, such as Doppler wind lidar, ceilometer and microwave radiometer could further improve the accuracy of annotations. For example, measurements of specific humidity and virtual potential temperature are useful indicators with respect to the height of the mixing layer. Note that using only ceilometer data as model input allows for integration of the algorithm in existing Automatic Lidars and Ceilometers networks (e.g., E-Profile, see Haefele et al., 2016). However, including these additional sensor data sources as model input could also further increase the accuracy of MBLH detection models (e.g., Kotthaus et al., 2023).

To explore model robustness, we have investigated training the model on other data sources than the manually annotated MBLHs. Specifically, annotations for Payerne were obtained from MeteoSwiss (Poltera et al., 2017) to train the deep learning architecture (results not presented here). Our annotations followed small-scale fluctuations in the RCS data closely, while the externally sourced annotations were made once per minute and could be characterised as more stable over time. The settings for the temporal and spatial resolutions of the ceilometer were also different. Hence, for this experiment one pixel represented a duration of 30 seconds and 15 metres of altitude. The results of this experiment indicated it was possible to capture the important characteristics from alternative annotated datasets. The Deep-Pathfinder methodology was robust against differences in annotation methods, leading to different results, but functioning appropriately regardless of the chosen dataset. Hence, the annotations and resolution of the input data mainly determine the quality of the final predictions.

Deep-Pathfinder is not aware of the optical properties in the lidar data, nor of the physical units, because it does not require this information. In that sense, the algorithm can be trained on background, range and laser power corrected backscatter lidar data from other lidars or ceilometers. For different types of ceilometers (e.g., Vaisala CL31), it is recommended to repeat the unsupervised pre-training using unlabelled data of the corresponding instrument. This should not be necessary when using Deep-Pathfinder at other locations with the same instrument type. Several experiments also indicated that unsupervised pre-training of encoder weights outperformed randomly initialising weights or loading ImageNet weights (Deng et al., 2009). This was assessed using the validation set, based on the binary crossentropy loss and the accuracy of the generated masks (results not presented). Instead of unsupervised pre-training, the model could also be pre-trained using MBLH predictions of (i) an

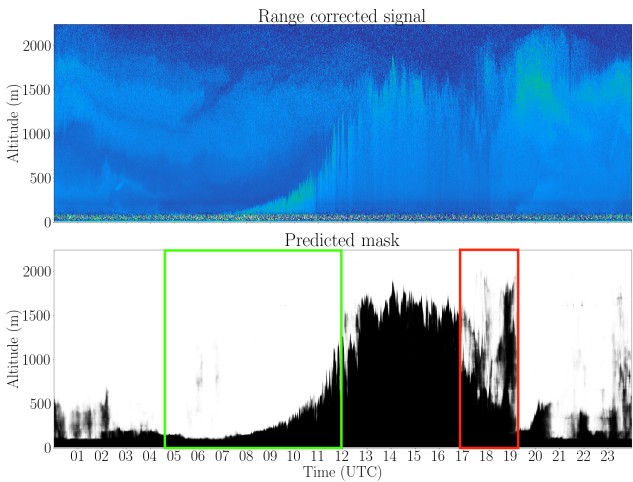

**Figure 9.** Model extensions: deriving quality control flags from unsharp regions in predicted masks. The green box provides an example of high model confidence, while the red box shows an example of lower model confidence.

existing MBLH algorithm, (ii) a numerical weather prediction system, or (iii) synthetic data. This initial step would directly result in a base deep learning model for the task of mask prediction, which can be fine-tuned using limited high-quality labelled data. Fine-tuning can be an iterative process, where the current shortcomings of the model are used to slightly improve specific annotations in the training data. The model can then be retrained using the updated annotations to enhance performance.

## 4.2 Model extensions / future research

We have identified various model extensions that could be investigated in future research. For example, model inference and post-processing leads to a grayscale output mask (see Fig. 9). Gray or blurry areas in these output masks indicate model uncertainty and can be used to develop quality control flags for operational use. Specifically, the value of the loss function during MBLH extraction (see Sect. 2.5) could be considered as an indicator of model confidence. Further, quality control flags could be set for specific atmospheric conditions, for example, so end users can exclude circumstances without vertical mixing. Other examples of useful flags are the occurrence of clouds (e.g., a binary indicator based on any detected cloud base height) and rain, since MBLH is not clearly defined during periods of precipitation.

Instead of using only two output classes (i.e., mixing layer or not), image segmentation methods are suitable for the detection of multiple classes. Extending the Deep-Pathfinder algorithm to multi-class prediction would also be a valuable future research direction. For example, Manninen et al. (2018) developed a method to obtain multiple classes from Doppler wind lidar information, such as convective mixing, non-turbulent, in cloud mixing, and wind shear (see Fig. 10). Obtaining such a set of annotated samples forms the main challenge for implementing this new functionality. In addition, ceilometer measurements can be accompanied with other data, such as profiles of horizontal and vertical wind speed from a Doppler wind lidar. We

envision the input image to have an extra channel to capture both the ceilometer and wind lidar data. Hence, no major changes to the neural network architecture would be required, besides minor changes to the input and final layer.

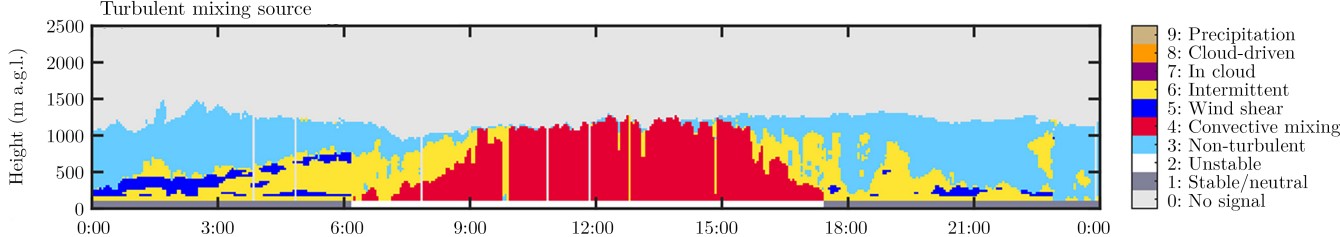

**Figure 10.** Model extensions: multi-class prediction (image from Manninen et al., 2018).

The analyses with alternative neural network architectures indicate that there is potential to further improve model accuracy, especially since these models were not implemented with unsupervised pre-training or additional variables (i.e., nighttime indicator). As U$^2$-Net uses a nested U-structure, it has so many skip connections that it would not be as suitable for the pre-training approach used in our study. In contrast, ResUNet-a performed well on the MBLH detection task and only has a limited number of skip connections. A custom implementation of ResUNet-a with temporary removal of the six skip connections would allow for unsupervised pre-training. Hence, the ResUNet-a architecture is a promising candidate to further improve model accuracy, in future research. Note that accuracy was not the only consideration for choosing the deep learning architecture. ResUNet-a was 8.5 times slower to calibrate than Deep-Pathfinder because of the higher complexity. Computational efficiency is an important consideration for operational use.

## 5    Conclusions

Our study shows that computer vision methods for image segmentation can be adapted to successfully track layers in data recorded by ceilometers. Through the use of unsupervised pre-training on large-scale unlabelled lidar data, appropriate results for MBLH estimation were obtained with only 50 days of annotations. Further, Deep-Pathfinder takes advantage of the full spatial and temporal resolution of the ceilometer, leading to high-resolution MBLH estimates. One challenge for model development is that no ground truth MBLH data is available, although the quality of the labels can be assessed based on different physical parameters (e.g., radiosonde based temperature profiles, radar wind profilers based turbulence profiles, Doppler lidar based wind profiles). This complicates method intercomparison. In comparison with existing MBLH approaches (e.g., rule-based layer selection algorithms), the number of assumptions required for MBLH detection was reduced. The initial structured annotation process (see Sect. 2.2) involves assumptions to determine the exact location of the MBLH. However, in the modelling phase manual feature engineering is avoided, as the mapping between input and label is learned directly from large-scale data.

As shown in previous studies, layer attribution can be improved by taking into account temporal consistency. Although existing path optimisation algorithms have greatly improved the temporal consistency of MBLH estimates, they can only be

evaluated after a full day of ceilometer data has been recorded. Deep-Pathfinder retains the advantages of temporal consistency by assessing MBLH evolution in 45-minute samples. However, our algorithm can also produce real-time estimates, by using the most recent 45 minutes of data and extracting the current MBLH from the right-hand side of the output mask. The availability of real-time MBLH estimates from a large-scale ceilometer network could be used for the advancement of NWP models.

Finally, it makes a deep learning approach as presented here valuable for operationalisation, as near real-time MBLH detection better meets the requirements of operational users in aviation, weather forecasting and air quality monitoring.

*Author contributions.* All authors were involved in conceptualization of the study. DAG, JSW and AA jointly completed the data curation and annotation. JSW designed the methodology, performed the experiments, analysed results and wrote the original draft of the manuscript. AA, DAG and JWN reviewed and edited the manuscript. AA and JWN contributed to funding acquisition for this project.

*Competing interests.* The authors declare that they have no conflict of interest.

*Acknowledgements.* This project was supported by the Dutch Research Council (NWO) through computation resources on the Snellius high-performance computing system (grant number EINF-3035) and by the Ruisdael Observatory for atmospheric science (grant number 184.034.015). The authors would like to acknowledge the anonymous reviewers for their valuable feedback, which helped improve the quality of the original manuscript. The authors would also like to thank Marijn de Haij (KNMI) for support of the ceilometer network in the
460 framework of E-Profile; Simone Kotthaus and Melania Van Hove (IPSL) for providing STRATfinder model predictions throughout 2020 at Cabauw, as well as Giovanni Martucci, Alexander Haefele (MeteoSwiss) and Yann Poltera (ETH Zurich) for providing annotated MBLH estimates and corresponding ceilometer data at Payerne from the PathfinderTURB study.

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
