# Peer review of "Deep-Pathfinder: A boundary layer height detection algorithm based on image segmentation"

_Atmospheric Measurement Techniques, 2023_

## Author Comment (AC1)

**Manuscript: amt-2023-80**

**Title**: Deep-Pathfinder: A boundary layer height detection algorithm based on image segmentation

Dear Reviewer:

Thank you for your time and effort to carefully comment on the manuscript. The manuscript has been thoroughly revised based on all comments. Furthermore, your feedback has been acknowledged in the 'Acknowledgements' section of the paper:

"The authors would like to acknowledge the anonymous reviewers for their valuable feedback, which helped improve the quality of the original manuscript." (see lines 448–449)

Reviewer 1, summary:

This paper addresses a very interesting topic and proposes an interesting solution, which can be applied in several ceilometers stations. However, some conceptual points need to be better described/commented.

A point-by-point response to your feedback is provided below.

General comments:

1. Line 123: This is a conceptual error. The Mixing Layer does not exist during the night. After sunset, due to a stable regime, the ML becomes Stable Boundary Layer and Residual Layer.

This has been corrected. For consistency with existing studies, the same layer definition has now been applied in our paper.

The following has been added to the Introduction section:

"Various sub-layers can be identified within the ABL, each with different properties. For example, one of the processes observed in the ABL is the vertical mixing of air. In our study, the same definition to characterise layers is used as described in Kotthaus et al. (2023). The term mixing boundary layer (MBL) is used to refer to the ABL sub-layer closest to the ground. Its height (MBLH) may indicate either the convective boundary layer (CBL) height or stable boundary layer (SBL) height, whichever is present at the given moment. The MBLH terminology is applied when no information on atmospheric stability is available to differentiate between SBL and CBL (Kotthaus et al., 2023, p. 435)." (see lines 21–26)

Further, the sentences mentioned in the comment were changed as follows:

"During nighttime, the pollution rich layers may drop to very low altitudes into the incomplete-overlap region of the CHM15k ceilometer and vertical mixing in fact ceases to exist. However, the concentration levels of pollutants remain layered and, therefore,

Cabauw mast measurements were used to aid in the identification of the presence and height of nocturnal layers." (see lines 170–173)

2. Line 126: The MLH only occurs during the daytime, because it is a convective process.

MLH has been changed to MBLH, corresponding to the definition added in the Introduction section (see our response to comment #1).

3. Line 127: How is possible to identify the Stable Boundary Layer from RCS if such a layer has only a thermodynamic definition? RCS provides the top of different aerosol layers, which in some cases can be coincident with a stable layer top, but it is not a rule. It is necessary a temperature and/or wind speed profile to identify a stable regime.

This description has been corrected.

The following has been added to the manuscript:

"Note that the attenuated backscatter retrievals during nighttime are not equivalent to the SBL or nocturnal boundary layer (NBL) defined by thermodynamics. Our MBLH definition (see Sect. 1) has been adopted for consistency with existing methods for MBLH detection and for comparison of results with these reference methods." (see lines 164–167)

4. Figure 7: MLH only occurs during the daytime. The MLH does not decrease it becomes Residual Layer and Stable Layer, it is important to present such division in this picture.

In the caption MLH has been changed to MBLH, corresponding to the definition added in the Introduction section (see our response to comment #1).

**Technical comments:**

5. Line 26: "throughout the low atmosphere"

The manuscript has been changed according to this suggestion. (see line 32)

6. Line 26: I recommend changing "Further" per "Consequently", because you used "further" other time in the same phrase

The double use of "further" in this phrase has been corrected. (see lines 32–34)

7. Figure 2: It is necessary to correct this figure, because the Mixing Layer only occurs during the daytime.

We agree that the mixing layer occurs during the daytime. However, being able to differentiate and track different sub-layers during the night is still important as it reviews

characteristics of the ABL development, and being able to consistently follow selected layers is a crucial step for layer attribution and further quality control/flagging.

Other MBLH detection methods also provide profiles for the entire day (e.g., the STRATfinder and Lufft algorithms described in the paper). Therefore, the definition of the mixing layer has been updated to MBLH for consistency with the reference methods (see our response to comment #1). The caption of the figure has also been updated.

8. Line: 119: MLH only occurs during the daytime

MLH has been changed to MBLH, corresponding to the definition added in the Introduction section (see our response to comment #1).

9. Line 124: nocturnal MLH : MLH only occurs during the daytime

The phrase "of the nocturnal MLH" has been removed from the manuscript. (see line 173)

10. Line 125: nocturnal MLH : MLH only occurs during the daytime

This sentence has been changed as follows in the manuscript:

"When RH at various height levels were similar, we considered this to be a more or less homogeneous nocturnal layer, and the layer height was approximated as the altitude where the RH values of higher layers with respect the lower layers started to diverge." (see lines 173–175)

11. Line 139: nocturnal MLH : MLH only occurs during the daytime

This has been revised as follows in the manuscript:

"The transition region from the daytime convective (mixed) boundary layer to a neutrally stratified residual layer (RL) with a stable NBL below, may not be clear from aerosol data (Schween et al., 2014). To complete the annotations in this region, the thermodynamic data informed the gradual decline towards the nighttime MBLH." (see lines 188–190)

Reference:

Schween, J. H., Hirsikko, A., Löhnert, U., and Crewell, S.: Mixing-layer height retrieval with ceilometer and Doppler lidar: from case studies to long-term assessment, Atmospheric Measurement Techniques, 7, 3685–3704, https://doi.org/10.5194/amt-7-3685-2014, 2014.

12. Line 140: Is the figure number correct?

This figure number was indeed incorrect. The order of figures 2 and 3 has been swapped during the revision process, so the reference in the text is now correct. (see line 192)

13. Line 222: This phrase is redundant because the convective boundary layer only occurs during the daytime

This has been corrected by removing "during daytime" from the manuscript.

The adjusted sentence is:

"The CBL was typically captured well by all three methods, with minimal differences in MBLH between them." (see lines 277–278)

14. Line: 225: nocturnal MLH : MLH only occurs during the daytime

MLH has been changed to MBLH, corresponding to the definition added in the Introduction section (see our response to comment #1).

The corresponding sentence in the manuscript has been changed as follows:

"Deep-Pathfinder and STRATfinder identified the nighttime MBLH around 100–200 meters altitude, although STRATfinder estimates were at a constant level slightly above the actual MBLH due to guiding restrictions in the algorithm." (see lines 279–281)

15. Line: 255: nocturnal MLH : MLH only occurs during the daytime

The corresponding paragraph has been rewritten as follows:

"Table 3 shows that the mean absolute difference between Deep-Pathfinder and STRATfinder was lowest during the evening, night and early morning growth phases. In contrast, the early and late afternoon is where they were least similar (see explanation above). The Lufft algorithm obtained higher estimates than both other algorithms during the evening, night and early morning growth phases, as it had a tendency to follow aerosol or moist layers in the residual layer (see Sect. 3.1)." (see lines 321–324)

---

## Author Comment (AC2)

**Manuscript**: amt-2023-80
**Title**: Deep-Pathfinder: A boundary layer height detection algorithm based on image segmentation

Dear Reviewer:

Thank you for your time and effort to carefully comment on the manuscript. The manuscript has been thoroughly revised based on all comments. Furthermore, your feedback has been acknowledged in the 'Acknowledgements' section of the paper:

"The authors would like to acknowledge the anonymous reviewers for their valuable feedback, which helped improve the quality of the original manuscript." (see lines 448–449)

Our response to the comments and suggestions are given in the detailed response below. Here, in the general response, we should like to clarify a few points that are important for the perspective of the comments:

- We consider the presented algorithm based on image-segmentation a proof-of-concept. We chose to apply it in such a way that it could be directly compared to independent algorithms that could be applied to the same data set. Since these algorithms are aimed at estimating the mixed boundary layer height (MBLH), Deep-Pathfinder was trained to do the same. As such, our paper aims to describe the Deep-Pathfinder method itself and the comparison to the STRATfinder and the Lufft algorithms. We agree that backscatter information is not sufficient to fully characterize layers and that the applicability of ceilometer data alone for the nocturnal boundary layer retrievals is debatable. However, being able to differentiate and track different sub-layers during the night is still important as it reviews characteristics of the ABL development, and being able to consistently follow selected layers is a crucial step for layer attribution and further quality control/flagging. We chose to follow the approaches of STRATfinder as the current state of the art. In our revised manuscript we have addressed the correct use of terms as requested, but for the in-depth discussion on the suitability of the use of (backscatter) lidar data, with or without synergy with other co-located data, we should like to refer to other papers.

- The name Deep-Pathfinder was chosen as a reference to the paper by De Bruine et al. (2017) that originates from the same research group in KNMI. Pathfinder was published as a proof-of concept at the time, and was subsequently enhanced and developed by other groups. This has resulted in versions of the algorithm, in particular PathfinderTURB and STRATfinder. The latter is now applied as the MBLH detection algorithm in the framework of EUMETNET E-Profile for the ALS network in a European central processing facility. Our hope is that the Deep-Pathfinder concept shall contribute similarly to the further advancement of MBLH detection for the application in the wider context such as E-Profile.

Reference:

de Bruine, M., Apituley, A., Donovan, D. P., Klein Baltink, H., and de Haij, M. J.: Pathfinder: applying graph theory to consistent tracking of daytime mixed layer height with backscatter lidar, Atmospheric Measurement Techniques, 10, 1893–1909, https://doi.org/10.5194/amt-10-1893-2017, 2017.

Although no comments were related specifically to the abstract, it was updated to reflect the changes made during this revision. The abstract of the manuscript has been rewritten as follows:

"A novel atmospheric layer detection approach has been developed based on deep learning techniques for image segmentation. Our proof-of-concept estimated the layering in the atmosphere, distinguishing between pollution rich layers closer to the surface and cleaner layers aloft. Knowledge of the spatio-temporal development of atmospheric layers, such as the mixing boundary layer height (MBLH), is important for the dispersion of air pollutants and greenhouse gases, and for assessing the performance of numerical weather prediction systems. Existing lidar-based layer detection algorithms typically do not use the full resolution of the available data, require manual feature engineering, often do not enforce temporal consistency of the layers, and lack the ability to be applied in near real-time. To address these limitations, our Deep-Pathfinder algorithm represents the MBLH profile as a mask and directly predicts it from an image with backscatter lidar observations. Deep-Pathfinder was applied to range-corrected signal data from Lufft CHM15k ceilometers at five locations of the operational ceilometer network in the Netherlands. Input samples of 224x224 pixels were extracted, each covering a 45-minute observation period. A customised U-Net architecture was developed with a nighttime indicator and MobileNetV2 encoder for fast inference times. The model was pre-trained on 19.4 million samples of unlabelled data and fine-tuned using 50 days of high-resolution annotations. Qualitative and quantitative results showed competitive performance compared to two reference methods: the Lufft and STRATfinder algorithms, applied to the same dataset. Deep-Pathfinder enhances temporal consistency and provides near real-time estimates at full spatial and temporal resolution. These properties make our approach valuable for application in operational networks, as near real-time and high-resolution MBLH detection better meets the requirements of users, such as in aviation, weather forecasting and air quality monitoring." (see lines 1–17)

A point-by-point response to your feedback is provided below.

1. The authors describe the application of a machine learning based computer vision algorithm to ceilometer backscatter data to derive mixing layer height (MLH). The approach to use computer vision based on neuronal network techniques (NN) and apply it to ceilometer backscatter data is to my knowledge new and the approach is therefore of general interest. Nevertheless the article lacks a deeper investigation of abilities and shortcomings of the algorithm. The evaluation with respect to other existing MLH retrievals based on physical considerations is very short and could be expanded. I therefore recommend major revisions to this work.

Thank you very much for your comments. The manuscript has been thoroughly revised based on the provided suggestions and very useful insights.

The evaluation of the method has been extended. For example, performance is now evaluated for different phases of the ABL development.

The following was added to the manuscript:

"Importantly, performance varied for the different phases of the diurnal development of the ABL. In the early morning, the ML grows into a stably stratified, unmixed NBL at the surface (i.e., roughly 4–8h UTC). This is typically followed by fast growth into the more or less neutrally stratified RL (8–12h UTC). In the early afternoon, the fully developed CBL grows slowly into the free troposphere (12–16h UTC). During the late afternoon a more or less sudden breakdown of convective turbulence and thus breakdown of mixing occurs (16–19h UTC). Finally, in the evening and night a new NBL and RL develops (19–4h UTC)." (see lines 316–321)

The Results section has been expanded to provide results for these ABL development phases. See the response to comment #4 for a more detailed description and results of the comparison with reference methods. Further, see the response to comment #125 for a more detailed description of the additional results for the alternative neural network architectures.

Existing methods based on physical considerations (i.e., bulk Richardson) have been used as a reference for annotation purposes. Further, STRATfinder has been evaluated in comparison to physics-based methods, and we achieved similar results to STRATfinder. Previous research has indicated that thermodynamic detection and aerosol-derived layer heights are inherently linked (Milroy et al., 2012), which has been added to the manuscript:

"These methods are complementary. For example, Milroy et al. (2012) showed the inherent link between thermodynamic and aerosol backscatter profiles based on a comparison of remote and in-situ data." (see lines 37–39)

Reference:

Milroy, C., Martucci, G., Lolli, S., Loaec, S., Sauvage, L., Xueref-Remy, I., Lavrič, J. V., Ciais, P., Feist, D. G., Biavati, G., and O'Dowd, C. D.: An assessment of pseudo-operational ground-based light detection and ranging sensors to determine the boundary-layer structure in the coastal atmosphere, Advances in Meteorology, 2012, 929 080, https://doi.org/10.1155/2012/929080, 2012.

2. Basic idea is that backscatter data from ceilometers provide all the information you need to identify the height of the mixed layer (ML). Ceilometers are automatic lidar systems originally developed to determine cloud base which defines the so called 'ceiling' for visible flight. They are therefore widespread at airports and constitute a rather dense measurement network. These instruments also provide - as an additional by-product - profiles of aerosol backscatter which shows especially on more or less cloud free, calm days typical patterns which give the impression that it is possible to visually 'see' how the boundary layer develops and how far up

turbulent mixing reaches. It is thus a logical step to use computer vision to analyse these patterns and try to identify the mixed layer (ML) or even the development state of the whole atmospheric boundary layer (ABL). There are uncounted studies proposing different approaches to derive MLH from ceilometer backscatter data (for a review see Haeffelin et al 2012, or very recent Kotthaus et al 2023). The authors give here also a short review but with a focus on methods which use multiple instruments and parameters. I am missing arguments why this study falls back to a single parameter from a single instrument. I would recommend to mention in the introduction the principal methods based on backscatter alone (Gradient, Wavelet transform, variance, fit of a function, … ) and give arguments why Neuronal Network based computer vision might perform better.

The paragraph where we provide an overview of other methods had indeed a section describing methods that used multiple instruments and parameters. However, the first half also referred to methods that used backscatter profiles only, for example, Toledo et al. (2014) and Rieutord et al. (2021). For clarification, this paragraph has now been split into two paragraphs.

At the end of the paragraph describing methods that use multiple instruments/parameters, an argument is provided why a single parameter from a single instrument could be beneficial:

"These approaches have been shown to generally improve prediction accuracy, although the use of multiple data sources may complicate the large-scale implementation in existing real-time detection networks, such as E-Profile (Haefele et al., 2016)." (see lines 83–84)

This is also noted in the Discussion section:

"Note that using only ceilometer data as model input allows for integration of the algorithm in existing Automatic Lidars and Ceilometers networks (e.g., E-Profile, see Haefele et al., 2016)." (see lines 372–373)

The following has been added to the Introduction section to describe the principal methods based on backscatter alone:

"A variety of methods has been developed that derive MBLH from backscatter profiles. For example, regions showing substantial change in the attenuated backscatter have been detected based on negative vertical gradients and inflection points (e.g., Sicard et al., 2006), wavelet covariance transform (e.g., Cohn and Angevine, 2000), and spatio-temporal variance (e.g., Menut et al., 1999)." (see lines 54–57)

The final paragraph of the Introduction provides arguments why neural networks for computer vision might perform better than other approaches:

"In summary, several methodological challenges still remain. Few methods incorporate temporal information to avoid jumps between layers. This is an issue for some of the machine learning methods described above. In contrast, many methods that are not based

on machine learning require expert knowledge to manually set modelling thresholds (e.g., for nighttime detection, instrument- and site-specific tuning). This also extends to the manual specification of guiding restrictions for layer selection. An open research question for MBLH detection is how to combine the advantages of different methods in a single approach. In particular, it would be beneficial to (i) promote temporal consistency of the MBLH profile, (ii) use the full resolution of the ceilometer, and (iii) limit manual feature engineering, specification of rules for layer selection, and site-specific tuning parameters. Further, not all existing methods can be used for real-time detection, which is an important quality for operational use. These challenges form the basis for the Deep-Pathfinder MBLH detection algorithm described in this paper." (see lines 85–93)

References:

Cohn, S. A. and Angevine, W. M.: Boundary layer height and entrainment zone thickness measured by lidars and wind-profiling radars, Journal of Applied Meteorology, 39, 1233–1247, https://doi.org/10.1175/1520-0450(2000)039<1233:BLHAEZ>2.0.CO;2, 2000.

Haefele, A., Hervo, M., Turp, M., Lampin, J.-L., Haeffelin, M., and Lehmann, V.: The E-PROFILE network for the operational measurement of wind and aerosol profiles over Europe, in: Proceedings of the WMO Technical Conference on Meteorological and Environmental Instruments and Methods of Observation (CIMO TECO 2016), pp. 1–9, World Meteorological Organization, Madrid, 2016.

Menut, L., Flamant, C., Pelon, J., and Flamant, P. H.: Urban boundary-layer height determination from lidar measurements over the Paris area, Applied Optics, 38, 945–954, https://doi.org/10.1364/AO.38.000945, 1999.

Rieutord, T., Aubert, S., and Machado, T.: Deriving boundary layer height from aerosol lidar using machine learning: KABL and ADABL algorithms, Atmospheric Measurement Techniques, 14, 4335–4353, https://doi.org/10.5194/amt-14-4335-2021, 2021.

Sicard, M., Pérez, C., Rocadenbosch, F., Baldasano, J. M., and García-Vizcaino, D.: Mixed-layer depth determination in the Barcelona coastal area from regular lidar measurements: methods, results and limitations, Boundary-Layer Meteorology, 119, 135–157, https://doi.org/10.1007/s10546-005-9005-9, 2006.

Toledo, D., Córdoba-Jabonero, C., and Gil-Ojeda, M.: Cluster analysis: a new approach applied to lidar measurements for atmospheric boundary layer height estimation, Journal of Atmospheric and Oceanic Technology, 31, 422–436, https://doi.org/10.1175/JTECH-D-12-00253.1, 2014.

3. At only one point in the paper it is mentioned that backscatter data provide just a proxy, but not directly the ML. It would be very helpful to give a short description why backscatter shows information about the state of the ABL (different aerosol content in the ML and above in the RL or the FT, ...). Such a description then could be used to argue when and why any algorithm based on aerosol backscatter alone can perform good, and when it has its shortcomings (no aerosol difference between ML

and layer above, effects of RH, ...). And of course when and why the algorithm performs good when compared to the reference algorithms (STRATfinder, and instrument build in), Examples for such a discussion can be found in Schween et al. (2014) or references in Kotthaus et al (2023).

The Introduction has been expanded to provide further information on why backscatter shows information about the state of the ABL.

The following has been added to the manuscript:

"Finally, aerosol-based MBLH methods attempt to observe the result of the mixing process via a proxy, as aerosol properties of different atmospheric layers can vary due to recent mixing processes. Hence, the boundaries of atmospheric layers can be approximated by detecting changes in aerosol properties. Generally, a rapid drop in aerosol content occurs beyond the top of the mixing layer." (see lines 44–47)

As noted in the comment, this approach has shortcomings when no difference in aerosol content exists between the mixing layer and the layer directly above.

The following has been added to the manuscript:

"Aerosol-based detection is also difficult when no or limited differences in aerosol content exist between the mixing layer and the layer directly above." (see lines 52–54)

Further, algorithms that are based on backscatter alone have shortcomings at night, as particles are still visible in the atmosphere but may not be related to mixing. The following text has been added to the manuscript to note this:

"During nighttime, the pollution rich layers may drop to very low altitudes into the incomplete-overlap region of the CHM15k ceilometer and vertical mixing in fact ceases to exist. However, the concentration levels of pollutants remain layered and, therefore, Cabauw mast measurements were used to aid in the identification of the presence and height of nocturnal layers." (see lines 170–173)

When the algorithm performs well when compared to the reference algorithms (STRATfinder, and manufacturer algorithm), has been addressed by comparing performance for different ABL development phases. See the response to comment #4, below. Several arguments have been provided for why the algorithm performs well. For example:

"The Lufft algorithm obtained higher estimates than both other algorithms during the evening, night and early morning growth phases, as it had a tendency to follow aerosol or moist layers in the residual layer (see Sect. 3.1)." (see lines 323–324)

"On some days, STRATfinder reached higher peak values than Deep-Pathfinder. In case of multiple cloud layers, our annotations typically followed the lower layer, while STRATfinder followed the higher layer." (see lines 311–313)

4. These difficulties in MLH detection are strongly connected to the different phases of the diurnal development of the ABL: (1) early morning growth of the ML into the stably stratified, unmixed nocturnal boundary layer at the surface (NBL). (2) Hours till noon: fast growth into the more or less neutrally stratified residual layer (RL), remainder of the ML of the previous day. (3) Noon and early afternoon: fully developed ML growing slowly into the free troposphere (FT). (4) Later afternoon: more or less sudden breakdown of convective turbulence and thus breakdown of mixing. (5) evening and night: development of a new NBL and RL. These ideal development can be found in text books (Stull, 1988 or Garratt, 1992) but also in current articles (Kotthaus et al 2023). The comparison of the different NN based algorithms has been done on three different timeslots for different ABL stages (tables 3-5). But I would recommend to evaluate the performance in comparison with STRATfinder and the instruments algorithm based on these five phases as well, i.e. expand the analysis presented around table 1 and 2 to at least some of these BLH development phases, and discuss results with respect to them: is it possible to detect the ML in every of these phases with equal chance or are there different problems at different phases. And can recommendations for retrieval algorithms derived from this?

The analyses have been performed again based on the different phases of diurnal development.

First, the suggested description of these phases has been added to the manuscript, as follows:

"Importantly, performance varied for the different phases of the diurnal development of the ABL. In the early morning, the ML grows into a stably stratified, unmixed NBL at the surface (i.e., roughly 4–8h UTC). This is typically followed by fast growth into the more or less neutrally stratified RL (8–12h UTC). In the early afternoon, the fully developed CBL grows slowly into the free troposphere (12–16h UTC). During the late afternoon a more or less sudden breakdown of convective turbulence and thus breakdown of mixing occurs (16–19h UTC). Finally, in the evening and night a new NBL and RL develops (19–4h UTC)." (see lines 316–321)

The analysis presented around Tables 1 and 2 has been expanded. Methods were compared during the different phases of development of the ABL by computing the mean absolute difference in each phase. The following table has been added to the manuscript:

Table 3. Method intercomparison showing the mean absolute difference (metres) for the various phases of development of the ABL, obtained using the test set.

| Method | Overall | Time of day (UTC) | | | | |
| --- | --- | --- | --- | --- | --- | --- |
| | | 4–8h | 8–12h | 12–16h | 16–19h | 19–4h |
| Deep-Pathfinder vs. STRATfinder | 189.0 | 127.6 | 178.1 | 251.5 | 316.5 | 156.7 |
| Deep-Pathfinder vs. Lufft | 323.5 | 392.9 | 219.9 | 117.7 | 293.3 | 396.5 |
| STRATfinder vs. Lufft | 369.8 | 437.2 | 285.4 | 220.9 | 306.4 | 435.2 |

Further, the following text has been added to the manuscript:

"Table 3 shows that the mean absolute difference between Deep-Pathfinder and STRATfinder was lowest during the evening, night and early morning growth phases. In contrast, the early and late afternoon is where they were least similar (see explanation above). The Lufft algorithm obtained higher estimates than both other algorithms during the evening, night and early morning growth phases, as it had a tendency to follow aerosol or moist layers in the residual layer (see Sect. 3.1). However, Lufft and Deep-Pathfinder estimates were substantially more similar in the early afternoon. During this period, the manufacturer algorithm can be used more confidently." (see lines 321–326)

Finally, the Tables in section 3.4 'Using alternative neural network architectures' have also been changed to use the same time bins, for consistency (see our response to comments #125 and #126)

5. It is mentioned in the discussion that Deep-Pathfinder prefers in situations with multiple cloud layers, the lower cloud, while STRATfinder tends to the upper cloud layer. As discussed in several papers (Schween et al. 2014, Kotthaus and Grimmond 2018 and Kotthaus et al. 2023 and references therein) clouds are a challenge for MLH detection, not only because of the more conceptual problem whether a cloud base is really the top of the ML. The evaluation of the algorithm could therefore also be restricted to situations with no or few clouds to see whether the algorithms agree better in such cases. And again the question: can recommendations for retrievals derived from that?

The differences do not specifically occur for the situation "clouds" versus "no clouds", but mainly for "single cloud layer" versus "multiple cloud layers". Examples of these situations are provided in Figure 7.

In the case of a single cloud layer, the MBLH is treated in a similar way as STRATfinder, putting the annotation at the top of the cloud layer. In fact, it is not difficult to detect a single cloud layer using computer vision, as the white areas are distinctive and the edges are clearly visible. Therefore, there is no need to exclude them.

In the case of multiple cloud layers, we know the differences between Deep-Pathfinder and STRATfinder are caused by the use of different assumptions. Deep-Pathfinder was trained to follow the first cloud layer rather than the second. If it was labelled otherwise, it would learn the alternative behaviour. Excluding cases with multiple cloud layers will reduce the differences between the methods, but this is to be expected.

The conclusion of our current evaluation is that situations with multiple cloud layers can lead to differences between the methods. Therefore, a recommendation is that for multiple cloud layers it should be kept in mind that differences may exist with other MBLH methods.

The following has been added to the manuscript:

"Hence, in case of multiple cloud layers, users should be aware that the methods may produce different MBLH estimates." (see lines 293–294)

6. A widespread misconception is that the high backscatter in the lower levels during night identifies the NBL or even a mixing layer. The high backscatter layer is result of the fact that aerosol backscatter does not only depend on aerosol particle number concentration but also on backscatter cross-section, which depends via Mie Theory on particle size, which in turn depends on relative humidity (RH, see e.g. Tuomi 1976 or Fitzgerald 1984 for the exact mathematical formulation and example plots). When the NBL develops in the evening, air is cooling from the bottom and a very stable stratification develops (I am sure this can be seen in the tower data at Cabauw, see also van Ulden and Wieringa 1996). Stable stratification means no, or only very, very weak local vertical mixing. Decreasing temperature lets RH increase and accordingly backscatter increase without a change in number concentration and without mixing of aerosol. This can be nicely seen in fig.2 where RH in the lowest layers nearly reaches 100%. The increase in backscatter is not linear with RH, and only RH values above 70-80% (depending on aerosol properties and wavelength of the lidar) lead to a significant increase in backscatter. As only the lower levels in the NBL reach high RH values above 70% high backscatter values can only be expected in the lower part of the NBL. In summary the increased backscatter values are a nonlinear combination of several parameters. They do not identify the upper border of the NBL and they do not mark a mixed layer. Therefore a backscatter based identification of a surface based layer during night has no meaning at all. I recommend to drop all the night time retrievals except the authors can show on the basis of the Cabauw tower data (potential (Tpot) temperature and water vapor mass mixing ratio (qv), but not RH) that the region of enhanced backscatter really identifies a mixed layer (Tpot and qv constant with height).

We agree that the backscatter information is not sufficient to fully characterize layers and that the applicability of ceilometer data alone for the nocturnal boundary layer retrievals is debatable. However, being able to differentiate and track different sub-layers during the night is still important as it reviews characteristics of the ABL development, and being able to consistently follow selected layers is a crucial step for layer attribution and further quality control/flagging.

The ABLH review paper by Kotthaus et at. (2023) describes the capabilities and limitations of methods based on attenuated backscatter, but it still applies the term "mixing boundary layer height" for ceilometer-based retrievals during day, night or whenever no information on atmospheric stability is available. That is also the case for the STRATfinder paper (Kotthaus et at. 2020) that uses a slightly different terminology. For consistency with existing studies, the same layer definition is applied in our paper.

The manuscript has been adjusted accordingly:

"In our study, the same definition to characterise layers is used as described in Kotthaus et al. (2023). The term mixing boundary layer (MBL) is used to refer to the ABL sub-layer closest to the ground. Its height (MBLH) may indicate either the convective boundary layer (CBL) height or stable boundary layer (SBL) height, whichever is present at the given moment. The MBLH terminology is applied when no information on atmospheric stability is

available to differentiate between SBL and CBL (Kotthaus et al., 2023, p. 435)." (see lines 22–26)

Further:

"Note that the attenuated backscatter retrievals during nighttime are not equivalent to the SBL or nocturnal boundary layer (NBL) defined by thermodynamics. Our MBLH definition (see Sect. 1) has been adopted for consistency with existing methods for MBLH detection and for comparison of results with these reference methods." (see lines 164–167)

We also need to clarify that the RH profile information from the tower was used for reference purposes, and it was only available for the training days at the Cabauw station (not available for De Kooy or De Bilt). The layer annotation was performed by strictly following features in the RCS data. Our manual layer identification and attribution strategy during nighttime was done in a similar fashion to what STRATfinder does for its layer attribution: the nighttime layer is selected by tracking backwards in time from the high-confidence MLH identified after sunrise. That way, the results and definitions from both algorithms are comparable.

One of the objectives of this paper is to demonstrate that the Deep-Pathfinder architecture can learn, with a limited amount of annotated training data, to consistently follow selected layers - at day and at night - which is a crucial step for the of layer attribution and quality control/flagging. As stated in the introduction of this document, a thorough debate on the existence of the MLH at night is beyond the scope of the paper.

Further, various other changes to the manuscript have been made based on this comment. For example, throughout the manuscript, all references to nocturnal mixing have been removed, MLH has been changed to MBLH, the description of the use of RH data has been clarified, etc. (see our responses to various comments below).

Reference:

Kotthaus, S., Bravo-Aranda, J. A., Collaud Coen, M., Guerrero-Rascado, J. L., Costa, M. J., Cimini, D., O'Connor, E. J., Hervo, M., Alados-Arboledas, L., Jiménez-Portaz, M., Mona, L., Ruffieux, D., Illingworth, A., and Haeffelin, M.: Atmospheric boundary layer height from ground-based remote sensing: a review of capabilities and limitations, Atmospheric Measurement Techniques, 16, 433–479, https://doi.org/10.5194/amt-16-433-2023, 2023.

7.  The authors write that they do not address limitations of the instrument due to its optical design. This is of course dangerous: Instruments have their limitations and any analysis should consider them. This is especially the case with the here used CHM15k ceilometer which is known to have a rather large region of incomplete overlap and an imperfect built in overlap correction (see Hervo et al. 2016). As a result there appear spurious gradients at heights of about 100m and 200m to 300m (see fig.2.iii, i.e. the gradient panel) but also in fig.1, fig.5, fig.6 and fig.8). The no-overlap zone, i.e. the range close to the instrument where no backscatter can be measured, is mentioned but its range is not given in the text. The instrument delivers

in this region noise (see e.g.fig.2) and the question arises whether this influences the algorithm. Another speciality of the CHM15k is the output variable which has the name 'beta_raw' in the original netcdf data files and which is described as 'normalized backscatter' and which I guess is used here as 'range corrected signal' (RCS). This variable is the overlap, background and range corrected signal. It lacks a correction for the temporal variability of emitted photos and diverse state parameters of the instrument (see Schween et al. 2014). And more important this variable lacks a conversion to physical units for backscatter (1/(Sr m)). Although this is not a full calibration it would make output of the instrument comparable to other ceilometers and it would allow to generalize the algorithm. All these points should be mentioned in the text.

This comment can be broken down into these questions:
- Overlap and overlap correction
- Beta_raw
- Calibration to physical units

We agree that the region of incomplete overlap is generally an issue to be considered in lidar applications. This applies in particular to applications where the lidar's data is used to derive or estimate physical quantities that rely on the optical parameters included in the lidar data, such as optical extinction. However, the Deep-Pathfinder approach does not rely on optical information. The algorithm is trained to find the layer structure using annotated data. The annotation uses human guidance and implicitly takes the lidar signal properties in the region of incomplete overlap into consideration. That is, if the algorithm is not taught to see a layer in a particularly noisy region, it will also not find it there. In other words, the Deep-Pathfinder approach has an implicit correction, if you will, for the incomplete overlap. We do not claim that we apply an overlap correction.

The beta_raw data product that is provided in the NetCDF datafiles from the Lufft CHM15k is corrected for an instrument specific overlap function. This overlap function is determined by the manufacturer and is unique for each optical module. Also included in the beta_raw data product is a normalisation for laser power fluctuations. The units of beta_raw are in units of backscatter (1/(Sr-m)), but are not corrected for optical extinction. Having said this, as for the argument regarding the region of incomplete overlap, the Deep-Pathfinder is not aware of the optical properties in the lidar data, nor of the physical units, and it doesn't need this information. In that sense, the Deep-Pathfinder algorithm can be trained on background, range and laser power corrected backscatter lidar data from other ceilometers.

The following has been added to the manuscript in relation to the overlap correction:

"The region of incomplete overlap is generally an issue to be considered in lidar applications, in particular when the lidar data is used to derive or estimate physical quantities that rely on the optical parameters included in the lidar data, such as optical extinction. However, the Deep-Pathfinder approach does not rely on optical information, as is explained in the following sections. Due to the training process, Deep-Pathfinder implicitly takes the incomplete overlap into account and does not require an additional overlap correction to be applied before analysis (see Hervo et al., 2016)." (see lines 121–126)

Further, the following has been added to the Discussion section of the manuscript:

"Deep-Pathfinder is not aware of the optical properties in the lidar data, nor of the physical units, because it does not require this information. In that sense, the algorithm can be trained on background, range and laser power corrected backscatter lidar data from other lidars or ceilometers." (see lines 385–387)

Finally, the range of the no-overlap zone has been added to the manuscript:

"For the vertical profile the 'beta_raw' variable was used (firmware v1.010 and v1.040) as it readily provided a normalised range-corrected signal (RCS), with instrument-specific overlap correction (usable above 80–100 metres altitude) and harmonisation between different CHM15k systems." (see lines 119–121)

Reference:

Hervo, M., Poltera, Y., and Haefele, A.: An empirical method to correct for temperature-dependent variations in the overlap function of CHM15k ceilometers, Atmospheric Measurement Techniques, 9, 2947–2959, https://doi.org/10.5194/amt-9-2947-2016, 2016.

8. And one minor point:

   Despite the similar name, the presented algorithm has nothing in common with the PathfinderTURB algorithm presented in Poltera et al (2017). Please clarify that in the text.

The following has been added to the manuscript:

"Deep-Pathfinder has a similar goal to other algorithms that attempt to find the path the MBLH follows based on ceilometer observations (e.g., Pathfinder, PathfinderTURB), although using a completely different approach based on deep learning." (see lines 96–97)

*Comments to text:*

9. line 21:

   "MLH is not constant, ... depending on a myriad of factors ..."

   The physical model of Ouwersloot and Vilà-Guerau de Arellano (2013) reduces this 'myriad of factors' to a handful: surface sensible heat flux, temperature difference over the inversion layer at top, temperature gradient of the atmosphere above the ML, subsidence.

The following has been changed in the manuscript:

"MBLH is not constant, but varies throughout the day and can range from less than 100 metres to a few kilometres, depending on the surface sensible heat flux, temperature difference over the inversion layer at the top, temperature gradient of the atmosphere above the mixing layer, and subsidence (Ouwersloot and Vilà-Guerau de Arellano, 2013)." (see lines 26–29)

Reference:

Ouwersloot, H. G. and Vilà-Guerau de Arellano, J.: Analytical solution for the convectively-mixed atmospheric boundary layer, Boundary-Layer Meteorology, 148, 557–583, https://doi.org/10.1007/s10546-013-9816-z, 2013.

10. line 39:

"... measuring the back-scattering by aerosols to eventually obtain estimates of particle concentrations"

Backscatter not only depends on particle number concentration but also on backscatter cross section which depends via Mie Theory on particle size which depends on relative humidity.

We fully agree with this comment. Note that we are not stating that the backscatter information alone, without any other assumptions, is sufficient for particle concentration estimates. To make this sentence clearer we replaced the word "obtain" for "support".

Now, the sentence reads as follows:

"This phenomenon can be observed using ceilometers that employ the lidar (light detection and ranging) measurement principle, emitting short laser pulses and measuring the back-scattering by aerosols to eventually support estimating particle concentrations at different altitudes." (see lines 47–49)

11. line 40:

".. incomplete optical overlap of many lidar systems results in a blind spot at low altitudes,"

To be precise: it is not a spot but a range of distances and this blind zone is due to no-overlap between laser beam and field of view of the receiver telescope. The incomplete overlap region can in principle be corrected (see Hervo et al 2016).

Thank you for pointing this out. To make this sentence more precise, we replaced the word "spot" for "zone", separated no-overlap and incomplete overlap, and we added Hervo et al. (2016) as a reference here.

The manuscript has been revised as follows:

"The laser beam and field of view of the receiver telescope often have a no-overlap area, resulting in a blind zone at low altitudes. Further, many lidar systems have an incomplete-overlap region, which can in principle be corrected (Hervo et al., 2016). Possibilities and limitations depend mainly on the instrument optical design, but estimating the height of shallow mixing layers can be challenging." (see lines 49–52)

Reference:

Hervo, M., Poltera, Y., and Haefele, A.: An empirical method to correct for temperature-dependent variations in the overlap function of CHM15k ceilometers, Atmospheric Measurement Techniques, 9, 2947–2959, https://doi.org/10.5194/amt-9-2947-2016, 2016.

12. line 42:

"Possibilities and limitations depend mainly on the instrument optical design, which will not be addressed here"

Some points need to be addressed (see above).

This comment was addressed in the response to the previous comment.

13. line 44:

" ... methods occasionally experience sudden jumps in the MLH profile from one layer to another."

I remember several attempts to avoid this kind of jumps (see e.g. Haeffelin et al 2011).

We have added a sentence about one of the methods from Haeffelin et al. (2011), the THT method, illustrating an approach of how this kind of jumps have previously been limited.

The following sentence has been added to the manuscript:

"This undesirable behaviour could be reduced by setting limits for the maximum altitude difference between successive MBLH estimates (Martucci et al., 2010)." (see lines 60–61)

Reference:

Martucci, G., Milroy, C., and O'Dowd, C. D.: Detection of cloud-base height using Jenoptik CHM15K and Vaisala CL31 ceilometers, Journal of Atmospheric and Oceanic Technology, 27, 305–318, https://doi.org/10.1175/2009JTECHA1326.1, 2010.

14. line 48:

"it is common to reduce the temporal resolution of the input data to one or two-minute segments"

There are two reasons for this: averaging reduces noise in the data which is important for gradient based approaches.

And secondly: mixing is an average process. It is not important how far up single convective plumes reach ang how far down single 'dry tongues from the FT reach (e.g. Couvreux et al 2007 or Traeumner et al. 2011).

Response to the first reason: we have now incorporated this in the manuscript. Note that reducing noise through averaging is not required for neural network-based methods, as they can learn to handle the noise. A high resolution is actually an advantage for neural network-based methods, as no important information is averaged out. Averaging can always be applied later on model output if a user wishes to do so, in order to reduce model uncertainty.

The following has been added to the manuscript:

"This reduces noise in the data, which is important for these gradient-based methods." (see line 66)

Response to the second reason: besides characterising the mixing process, plumes can be important for other applications. For instance, these high-resolution features give an indication of the entrainment zone. Further, for large-eddy simulation (LES) models, physical processes can be resolved more accurately using high-resolution boundary layer estimates.

15. line 57:

"Vivone et al. (2021) used edge detection techniques ..."

A former version of the STRAT algorithm named STRAT-2D already used edge detection (Haeffelin et al 2011). And is edge detection already machine learning?

Edge detection is a traditional computer vision technique. By itself, it is not machine learning. The study by Vivone et al. (2021) was mainly mentioned here as we use deep learning techniques for computer vision, so the link is on computer vision rather than machine learning. This sentence has been moved and rephrased for clarification. The study by Haeffelin et al. (2011) has also been added.

The following change has been made to the manuscript:

"Several studies also used techniques from computer vision, such as edge detection, to identify layer boundaries (e.g., Haeffelin et al., 2012; Patel et al., 2021; Vivone et al., 2021)." (see lines 75–77)

Reference:

Haeffelin, M., Angelini, F., Morille, Y., Martucci, G., Frey, S., Gobbi, G. P., Lolli, S., O'Dowd, C. D., Sauvage, L., Xueref-Rémy, I., Wastine, B., and Feist, D. G.: Evaluation of mixing-height retrievals from automatic profiling lidars and ceilometers in view of future integrated networks in Europe, Boundary-Layer Meteorology, 143, 49–75, https://doi.org/10.1007/s10546-011-9643-z, 2012.

16. line 65:

> End of introduction and literature review:
>
> All presented machine learning based algorithms combine several parameters. I am missing a strong argument why we go now back to the single parameter aerosol backscatter. I would also recommend to describe the different strategies to derive MLH from backscatter profiles (gradient, fit of a step function, wavelet analysis, etc. see e.g. Haeffelin et al. 2011 or Kotthaus et al. 2023 for overview of these methods).

This paragraph has now been split in two paragraphs, as only the second half was related to machine learning methods that combine several parameters.

A single parameter model will be easier to apply in many locations, because of the large existing ceilometer network (e.g., E-Profile). This means there are no requirements to have additional instruments at the same location as the ceilometer. Our research also contributes to existing initiatives to develop new methods for this ceilometer network.

This has now been made more specific in the manuscript, as follows:

"These approaches have been shown to generally improve prediction accuracy, although the use of multiple data sources may complicate the large-scale implementation in existing real-time detection networks, such as E-Profile (Haefele et al., 2016)." (see lines 83–84)

Further, the Discussion section has been adapted to explain how data from other instruments can be incorporated to further improve model accuracy:

"In addition, ceilometer measurements can be accompanied with other data, such as profiles of horizontal and vertical wind speed from a Doppler wind lidar. We envision the input image to have an extra channel to capture both the ceilometer and wind lidar data." (see lines 409–411)

Regarding the second half of this comment, a description of different strategies to derive MLH from backscatter profiles has been added to the manuscript. This is described as follows:

"A variety of methods has been developed that derive MBLH from backscatter profiles. For example, regions showing substantial change in the attenuated backscatter have been detected based on negative vertical gradients and inflection points (e.g., Sicard et al., 2006), wavelet covariance transform (e.g., Cohn and Angevine, 2000), and spatio-temporal variance (e.g., Menut et al., 1999)." (see lines 54–57)

References:

Cohn, S. A. and Angevine, W. M.: Boundary layer height and entrainment zone thickness measured by lidars and wind-profiling radars, Journal of Applied Meteorology, 39, 1233–1247, https://doi.org/10.1175/1520-0450(2000)039<1233:BLHAEZ>2.0.CO;2, 2000.

Haefele, A., Hervo, M., Turp, M., Lampin, J.-L., Haeffelin, M., and Lehmann, V.: The E-PROFILE network for the operational measurement of wind and aerosol profiles over Europe, in: Proceedings of the WMO Technical Conference on Meteorological and Environmental Instruments and Methods of Observation (CIMO TECO 2016), pp. 1–9, World Meteorological Organization, Madrid, 2016.

Menut, L., Flamant, C., Pelon, J., and Flamant, P. H.: Urban boundary-layer height determination from lidar measurements over the Paris area, Applied Optics, 38, 945–954, https://doi.org/10.1364/AO.38.000945, 1999.

Sicard, M., Pérez, C., Rocadenbosch, F., Baldasano, J. M., and García-Vizcaino, D.: Mixed-layer depth determination in the Barcelona coastal area from regular lidar measurements: methods, results and limitations, Boundary-Layer Meteorology, 119, 135–157, https://doi.org/10.1007/s10546-005-9005-9, 2006.

17. line 83:

"... and limits manual feature engineering,"

What do you mean by feature engineering. Please give examples.

Machine learning is very good at automatic feature extraction, meaning that based on the data it finds the most important patterns to look for. Manual feature engineering is the opposite. An example of manual feature engineering is specifying which values represent a cloud. A machine learning approach would not need this specification, as it can learn what clouds looks like (all the different shapes & patterns of it). Note that we're not trying to teach the algorithm to predict/recognise clouds in this study, but it is an example.

A drawback of manual feature engineering is that the defined features (i.e., variables) may not be complete and some important features/variables are missing. It is also subjective in the sense that different researchers may not come up with the same set of features. Machine learning provides a more objective approach to feature extraction.

The following changes have been made to the manuscript:

"This promotes temporal consistency of MBLH estimates, while using the maximum resolution of the ceilometer. It also limits manual feature engineering, meaning there is no need to explicitly define the important characteristics that will be used to create the MBLH profile (e.g., which values represent a cloud)." (see lines 102–104)

18. Figure.1:

I am missing a date and annotations on the axes and a color scale.

The image also makes the overlap problems of the CHM15k ceilometer visible (change from dark blue to green in the lower fifth and to the right of the center).

This Figure is meant to have no date, annotations and colour scale. The goal of this figure is to show how we frame the MLH detection problem in a computer vision framework. They are two images: the first one is an input image, the second one an output image. In a computer vision approach, the axes and legend will not be supplied to the model.

The following has been added to the caption of Figure 1:

"The plot axes and colour scale are not supplied to the model and have therefore been omitted."

19. line 95:

"...recorded between June 2020 and February 2022"

Why do you not use full years of data but instead one year and nine months, covering two winters, one and a half summers and only one spring.

Deep learning requires very large amounts of data, and this was everything that was available at the time of this research.

The full dataset from June 2020 until February 2022 was only used for model pre-training. Since the pre-training loss converged to zero, this indicates that all seasons were adequately captured.

The following was added to the manuscript:

"This dataset contained backscatter profiles from ceilometers in the KNMI observation network and the full dataset available at the time of this research was obtained, containing measurements from June 2020 to February 2022." (see lines 114–116)

20. line 98:

Range corrected signal:

Output of the CHM15k ceilometers depend on firmware version running on the instrument.

Please specify what output parameter of the instrument you are using (beta_raw ?) Does it include correction for laser power fluctuations, incomplete overlap (in which range), etc ?

We agree that this needs more information.

The following changes have been made to the manuscript:

"The ceilometers recorded data continuously at 12-second temporal and 10-meter vertical resolution. For the vertical profile the 'beta_raw' variable was used (firmware v1.010 and v1.040) as it readily provided a normalised range-corrected signal (RCS), with instrument-specific overlap correction (usable above 80–100 metres altitude) and harmonisation between different CHM15k systems." (see lines 118–121)

Reference:

User Manual Lufft CHM15k Ceilometer, https://www.lufft.com/fileadmin/lufft.com/07_downloads/1_manuals/Manual_CHM15k_EN_R19.pdf

21. line 99:

'manufacturer's algorithm'

There are two extended abstract of posters and talks presented at the ISARS 2010 conference about the manufacturers algorithm: Teschke and Poenitz (2010, TP2010) and Frey et al (2010, F2010).

My notes say that TP2010 is the algorithm used by the ceilometer from firmware version 0.719 onwards and F2010 was the version before. But you should check that with the operator and the manufacturer of the instruments.

Before the initial submission of the manuscript, we requested and received a document from the manufacturer with a description of the methodology. However, it was explicitly stated to treat this information as proprietary. Therefore, we cannot incorporate this information in the paper.

22. 'benchmark'

I do not understand what the difference between 'only ... a benchmark' and an evaluation is, please clarify.

"Benchmark" is typically used in computer science, but evaluation also works here. It is now referred to as a reference method, based on the suggestion in comment #95.

The following has been changed in the manuscript:

"Further, MBLH estimates from the manufacturer's layer detection algorithm were included for reference purposes." (see line 138–139)

23. line 102:

"RCS was capped to [0, 1e6] and rescaled to [0, 1]."

Scaling the RCS is of course a valid method, but as you do not provide a unit to the capping RCS value this scaling becomes arbitrary. If you want your method to be independent of the instrument you should provide at least an estimate for the physical units ((Sr m)-1).

We have now included the RCS units and an estimate for the corresponding attenuated backscatter interval of the training dataset using the calibration constant for the corresponding ceilometer.

The manuscript was changed as follows:

"First, the RCS was capped to [0, 1e6] $m^2$ counts $s^{-1}$ and rescaled to [0, 1]. This corresponds to an attenuated backscatter interval of about [0, 2.4–3.4] $Mm^{-1}$ $sr^{-1}$, depending on the specific ceilometer used and its respective calibration constant provided by E-Profile (Wiegner and Geiß, 2012; Haefele et al., 2016)." (see lines 146–149)

References:

Haefele, A., Hervo, M., Turp, M., Lampin, J.-L., Haeffelin, M., and Lehmann, V.: The E-PROFILE network for the operational measurement of wind and aerosol profiles over Europe, in: Proceedings of the WMO Technical Conference on Meteorological and Environmental Instruments and Methods of Observation (CIMO TECO 2016), pp. 1–9, World Meteorological Organization, Madrid, 2016.

Wiegner, M. and Geiß, A.: Aerosol profiling with the Jenoptik ceilometer CHM15kx, Atmospheric Measurement Techniques, 5, 1953–1964, https://doi.org/10.5194/amt-5-1953-2012, 2012.

24. line 105:

"The software package labelme (Wada, 2022) was used for annotations"

This means annotation was done manually by visually inspecting the data ?

I recommend to point that out.

Yes, that is correct.

The following was changed in the manuscript:

"The software package labelme (Wada, 2022) was used for manually annotating a small part of the data (i.e., layer attribution)." (see lines 153–154)

The annotation process itself is described in the next paragraph, starting with:

"The annotation process started with a visual inspection of the RCS data" (see line 161)

25. line 108:

"The main location was Cabauw as this location also provided humidity profile information..."

Please give some more information - geographical coordinates, infrastructure like the tower and other instruments (former CESAR now Ruisdael Observatory etc) literature ?

Where does the humidity information come from ? (Tower, Microwave radiometer, ... ?)

Additional information, including geographical coordinates and infrastructure, has been added to the manuscript in response to this comment.

The following has been added to the manuscript:

"The majority of annotated data was from Cabauw, located in the western part of the Netherlands (51.971° N, 4.927° E) at 0.7 m below mean sea level. At this site a large set of instruments is operated to study the atmosphere and its interaction with the land surface (see Fig. 2). Here, KNMI and partners have been carrying out a continuous measurement program for atmospheric research since 1972 (see https://cabauw.knmi.nl). The measurement program has evolved with increasingly advanced measurement techniques and instruments, often in collaboration with other institutes and universities. For this study, the CHM15k ceilometer was used together with ancillary data such as relative humidity (RH) information that was collected at the 10, 20, 40, 80, 140 and 200 m levels of the 213 metres high meteorological tower (Bosveld, 2020)." (see lines 129–136)

Reference:

Bosveld, F. C.: The Cabauw in-situ observational program 2000 – present: instruments, calibrations and set-up, Technical Report TR-384, Royal Netherlands Meteorological Institute (KNMI), De Bilt, https://cdn.knmi.nl/knmi/pdf/bibliotheek/knmipubTR/TR384.pdf, 2020.

26. line 109:

"Instead of creating a specialised model for the Cabauw ceilometer, the model should generalise to new locations."

By restricting the training and analysis to Cabauw only, this sentence becomes an nonproven claim. I would be glad to see how the algorithm performs at any of the other sites.

Model training was not restricted to Cabauw only:

- The model pre-training phase used all downloaded data from all locations: Cabauw, De Kooy, Groningen Airport Eelde, Maastricht Aachen Airport and Vlissingen.
- The model fine-tuning phase used data from Cabauw, De Kooy, and De Bilt.

Hence, the trained model is not a specialized model for the Cabauw ceilometer.

For model evaluation, we chose one location (Cabauw) to explore the results in detail, but based on the training procedure there is no indication that we have arrived at a specialised model that would only work for Cabauw.

The goal of this sentence was not to make a claim about model performance, but to describe what we have chosen to do.

The manuscript has been revised as follows for clarification:

"Model pre-training (see Sect. 2.4) used all ceilometer data from Cabauw, De Kooy, Groningen Airport Eelde, Maastricht Aachen Airport and Vlissingen. Model fine-tuning used a subset of data from Cabauw, De Kooy and De Bilt and some days outside the June 2020 to February 2022 period (e.g., for validation), which were manually annotated. The majority of annotated data was from Cabauw, located in the western part of the Netherlands (51.971° N, 4.927° E) at 0.7 m below mean sea level." (see lines 127–131)

27. line 110:

"... several days of data from ceilometers at De Kooy and De Bilt were also annotated."

Could you give some more information (geographical coordinates, why these sites and not any other of the KNMI dataset, and how did you select these 'several days' ? what made you choosing them?). As far as I understand annotated training was done with Cabauw plus some few data from these two other sites. The analysis in section 3,3 and Fig7 deals only with Cabauw data. Thus the starting sentence of section 2.1 ("a large dataset was downloaded...") is misleading: You used nearly only data from Cabauw. How does the algorithm perform at the other sites ?

See our response to the previous comment. We used the entire dataset for model pre-training, containing data of five different locations.

Changes to the manuscript have been included in the comment above.

The annotator chose the days to arrive at a representative sample, as described in the manuscript:

"Representative cases under a variety of atmospheric conditions were selected for annotation to cover a broad range of boundary layer dynamics. The captured atmospheric conditions included clear days with a distinct CBL, cold days with a shallow boundary layer, days with cloud cover, days with multiple cloud layers, and precipitation events." (see lines 158–160)

28. Figure 2:

Please describe more accurate what is on top, middle and bottom plot. The enumeration (i, ii, etc) is not helpful. Helpful would be colour bars for the colour shading in the top and middle plot (Which colours code for min, middle, max, is white=cloud and brown=rain? I see more colors in the gradient plot than you mention in the text. The X-Axis annotation could be clearer (is this 'Month Day hour' or 'Day Month hour' is the hour in CET, CEST, LT or UCT ?). It would be a good idea to mark sun rise and sunset or at least provide the times in the figure caption. Minor tics on the X- and Y-axis at e.g. 100m steps and every hour would be helpful.

This figure has been updated according to the suggestions. Colour bars have been added for both the top and middle plot. The labels on the X-axis have been updated to avoid confusion. Minor tick marks have been added on the X- and Y-axis, every hour and at 100m steps.

Further, the caption has been updated as follows to describe the plots in more detail:

"Figure 3. Combining data sources for 8 June 2021 (left part of the plot) and the first half of 9 June 2021, to annotate the MBLH on 8 June. The top plot shows the RCS data with layers detected by the ceilometer (black lines), the thermodynamic MBLH from ECMWF (blue line) and the annotated MBLH (red line). The middle plot shows the negative (magenta) and positive (green) gradients of the RCS data, with layers detected by the ceilometer (black lines) and the annotated MBLH (red line). The bottom plot contains the measured relative humidity by the Cabauw mast at different altitudes (various colours). Sunrise was at 03:21 UTC and sunset at 19:55 UTC."

29. How do you deal with the noise in the data in the blind zone (below about 100m)? Would it not be more honest to remove it from the data as it cannot contain any information? The lidar is literal blind in this region. The laser beam is out of the field of view of the telescope. It cannot see the laser beam.

The noise itself is not directly an issue for the machine learning methodology, as the method will learn that there is no relation between the data below 100m and the MLH (i.e., it learns to ignore the noise).

Since we use an annotation of 0 as a special case, it is actually useful to have a small buffer (i.e., the altitude region near the surface) at the bottom of the image.

No changes have been made to the manuscript.

30. Top and middle plot nicely show the problems with the incomplete-overlap-correction of the CHM15k: there is a step in the top colour plot, and a vertical gradient in the middle plot at about 200m. This present as a horizontal mark starting at ~14:00 running through until the end of the plot. Your annotated MLH follows this artificial line around 21:00 i.e. this step generates an artificial MLH.

There is indeed a degree of subjectivity in the annotation process, which is especially apparent in areas where it is not 100% clear where the MLH is located.

In this case, the layer passed through the incomplete-overlap artefacts. One may argue that the layer was still visible around 9pm, as the signal looks quite different from just before 9pm. Annotating the MLH on the blue line around 9pm would also be subjective. Importantly, in some cases the true MLH is actually located at an altitude in the incomplete-overlap correction zone. These cases will inherently be difficult to annotate.

No changes have been made to the manuscript.

31. I do not understand how you could annotate the afternoon break down around 17:30 I do not see any hints why it should be there (no colour change in top plot no gradients in middle plot). And if there should be a faint gradient : could this not just be the end of the plume (or air mass ?) with lower backscatter (light green) rising from the ground at about 17:00. Similar for the annotated MLH from 06-09:00 and 09:30 to 10:00 when there seems to be no gradient.

This part is indeed difficult to annotate and leads to uncertainty in the annotations. The annotators looked for a path (layer edge), albeit with low gradient, that could connect the closest previously annotated neutrally stratified RL to the daytime convective boundary layer.

The paragraph describing this part of the annotation process has been rewritten as follows:

"To complete the annotations in this region, the thermodynamic data informed the gradual decline towards the nighttime MBLH. The annotators looked for a path (layer edge), albeit with low gradient, that could connect the closest previously annotated neutrally stratified RL to the daytime CBL. An example of this process can be seen in Fig. 3 between 4pm and 8pm. If the aerosol profile was too smooth, a sudden jump to the nighttime MBLH was annotated." (see lines 189–193)

32. line 111:

What are representative cases ? How do you select them, what are the criteria?

For model training it is important to not only select days with textbook-like behaviours for annotation. In that case, the model would not be capable of handling situations with very different conditions that it has not encountered before. Hence, it is important that the model is trained with a variety of conditions. It is less important that the frequency of each

specific condition is the same as in reality. Instead, the model should have seen a few events of each type of conditions.

The following has been added to the manuscript:

"The captured atmospheric conditions included clear days with a distinct CBL, cold days with a shallow boundary layer, days with cloud cover, days with multiple cloud layers, and precipitation events." (see lines 159–160)

33. line 115:

"The information on potential layer boundaries was combined with other data sources ... For example, ... "

I would recommend to give all sources you used and not just examples.

All sources used in the annotation process have now been described.

The manuscript was revised as follows:

"For this study, the CHM15k ceilometer was used together with ancillary data such as relative humidity (RH) information that was collected at the 10, 20, 40, 80, 140 and 200 m levels of the 213 metres high meteorological tower (Bosveld, 2020). Other data sources for annotation included MBLH information from ECMWF, based on bulk Richardson number (ECMWF, 2017), which was obtained via the Cloudnet model output for Cabauw (CLU, 2022) and indicated the general atmospheric conditions. Further, MBLH estimates from the manufacturer's layer detection algorithm were included for reference purposes." (see line 134–139)

References:

Bosveld, F. C.: The Cabauw in-situ observational program 2000 – present: instruments, calibrations and set-up, Technical Report TR-384, Royal Netherlands Meteorological Institute (KNMI), De Bilt, https://cdn.knmi.nl/knmi/pdf/bibliotheek/knmipubTR/TR384.pdf, 2020.

CLU: ECMWF icon-iglo-12-23 model data; 2021-01-01 to 2022-01-01; from Cabauw. Generated by the cloud profiling unit of the ACTRIS Data Centre, https://hdl.handle.net/21.12132/2.085d52ae0bcc4fc7, 2022.

ECMWF: IFS documentation CY43R3 – Part IV: Physical processes, European Centre for Medium-Range Weather Forecasts, Reading, https://doi.org/10.21957/efyk72kl, 2017.

34. line 116:

"... thermodynamic MLH information from Cloudnet (ECMWF) model output (CLU, 2022) ... "

You got this MLH from the cloudnet database for Cabauw, but it is generated in the Integrated forecast system (IFS) of the European Center for Midrange Weather Forecast (ECMWF). I am sure you can find an appropriate reference for this thermodynamic MLH in IFS and give also a short description how it is calculated (Parcel method, Bulk-Richardson, flux minimum, etc...). And your reference under CLU 2022 is not understandable nor traceable. Cloudnet provides on its website unique identifiers and a 'how to cite' description (see e.g. https://cloudnet.fmi.fi/file/8f6ab79b-4eeb-42ab-abbc-045f9e5e172e ).

The CLU reference has been updated, an additional reference was added, and it has been added that the calculation was based on bulk Richardson.

The manuscript was changed as follows:

"Other data sources for annotation included MBLH information from ECMWF, based on bulk Richardson number (ECMWF, 2017), which was obtained via the Cloudnet model output for Cabauw (CLU, 2022) and indicated the general atmospheric conditions." (see lines 136–138)

References:

CLU: ECMWF icon-iglo-12-23 model data; 2021-01-01 to 2022-01-01; from Cabauw. Generated by the cloud profiling unit of the ACTRIS Data Centre, https://hdl.handle.net/21.12132/2.085d52ae0bcc4fc7, 2022.

ECMWF: IFS documentation CY43R3 – Part IV: Physical processes, European Centre for Medium-Range Weather Forecasts, Reading, https://doi.org/10.21957/efyk72kl, 2017.

35. line 118:

"MLH estimates from the manufacturer's layer detection algorithm were included for comparison purposes."

Is it really comparison? Or does the person doing the annotation see the manufacturer MLH and eventually follows it if he or she thinks it is 'a good one'? In this case it would be an additional information. And I am wondering what this MLH really is: in figure 2 there are at certain times several MLH's visible. I.e. there is not only one MLH. I guess the instrument provides a first, second, third candidate. Please describe in more detail what 'the MLH' from the instrument is and how you use it.

The MLH from the instrument is the result of a wavelet covariance transform method (proprietary algorithm). This method detects multiple layers. The Lufft manual describes that the lowest identified layer can be interpreted as mixing layer height.

The annotator uses the manufacturer's MLH estimate as extra information to understand the atmospheric development during the day. Therefore, 'for comparison purposes' has

been changed to 'for reference purposes'. Annotation was always performed on the high-resolution RCS data, not merely accepting the MLH estimates from the manufacturer.

The following has been added to the manuscript:

"Further, MBLH estimates from the manufacturer's layer detection algorithm were included for reference purposes. This proprietary algorithm, based on wavelet covariance transform, identified multiple candidate layers from the RCS data, where the lowest identified layer can typically be interpreted as the MBLH." (see lines 138–141)

36. line 119:

"... the nocturnal MLH ..."

This is a common misconception: there is no nighttime mixing layer except in the heat islands of cities. In a rural location like Cabauw, a strong inversion develops during night, called the nocturnal boundary layer (NBL). This is a very stably stratified layer in the lowest few hundred meters of the BL (see Stull 1988 or Garrett 1992). Stable stratification means no vertical turbulent exchange, no mixing. And because there is no mixing the strong vertical temperature gradient can become even stronger. The surface decouples from the layers above and becomes due to radiative cooling and surface evaporation cooler and cooler. Any mixing would destroy this gradient (see van Ulden and Wieringa 1996). As temperature decreases relative humidity increases, aerosol particles grow and backscatter increase. There is no mixing, no transport of aerosol involved in this increase in backscatter. The height of the strong backscatter layer is not connected to the NBL it just represents the layer where depending on the type of aerosol RH is above 60-80%. Accordingly the high backscatter at night at the surface is not a mixing layer and it also does not identify the NBL.

See our response to comment #6.

The main changes to the manuscript have been included below. First, in the Introduction the following definition was added:

"In our study, the same definition to characterise layers is used as described in Kotthaus et al. (2023). The term mixing boundary layer (MBL) is used to refer to the ABL sub-layer closest to the ground. Its height (MBLH) may indicate either the convective boundary layer (CBL) height or stable boundary layer (SBL) height, whichever is present at the given moment. The MBLH terminology is applied when no information on atmospheric stability is available to differentiate between SBL and CBL (Kotthaus et al., 2023, p. 435)." (see lines 22–26)

Just above the line mentioned in this comment, the following has been added:

"Note that the attenuated backscatter retrievals during nighttime are not equivalent to the SBL or nocturnal boundary layer (NBL) defined by thermodynamics. Our MBLH definition

(see Sect. 1) has been adopted for consistency with existing methods for MBLH detection and for comparison of results with these reference methods." (see lines 164–167)

Further, the phrase 'nocturnal MLH' has been removed. This sentence has been revised as follows:

"The layer attribution started with identification of the nocturnal layer structure before sunrise by tracking backwards in time from the high-confidence MBLH identified after sunrise." (see lines 168–169)

If an end user of the algorithm would be more interested in following different annotations (e.g., an MBLH of 0 for the situation described in the comment), the algorithm is sufficiently flexible to use such annotations during the training process. This has been described in the Discussion:

"Fine-tuning can be an iterative process, where the current shortcomings of the model are used to slightly improve specific annotations in the training data. The model can then be retrained using the updated annotations to enhance performance." (see lines 395–396)

37. line 120:

"... the humidity profile ..."

I guess you mean relative humidity (RH).

Yes, this is indeed relative humidity and it has been changed in the manuscript. (see line 169)

38. line 122:

" ... the partially visible range of the ceilometer,..."

You mean either the 'no-overlap' or the 'incomplete-overlap' region.

In the no-overlap region the Laser beam and the field of view (FOV) of the lidar telescope do not overlap. The Lidar can not see light backscattered from the laser beam it is literally blind. In this height range there is no information in the ceilometer data about the state of the atmosphere. The incomplete-overlap region is where the laser partially lies within the FOV of the telescope. It is possible to correct this (see Hervo et al 2016 or Wiegner and Geiss 2012 (note that the latter is about the CHM15kx - with just a note about the CHM15k)).

For the CHM15k the no-overlap region ends at a range of roughly 100m the incomplete overlap ends at roughly 350m. As very small details of the optics determine the overlap, the manufacturer uses for every ceilometer an individual build-in overlap correction function. This function is not perfect (see remarks for fig.2). Hervo et al (2016) provided a method to correct it including a temperature

sensitivity. To get precise numbers for the overlap regions refer to the operator or the manufacturer of the instrument.

This has been changed to 'incomplete-overlap':

"During nighttime, the pollution rich layers may drop to very low altitudes into the incomplete-overlap region of the CHM15k ceilometer and vertical mixing in fact ceases to exist." (see lines 170–172)

Further, the following has been added to the Introduction section:

"The laser beam and field of view of the receiver telescope often have a no-overlap area, resulting in a blind zone at low altitudes. Further, many lidar systems have an incomplete-overlap region, which can in principle be corrected (Hervo et al., 2016)." (see lines 49–51)

Finally, the overlap correction function has now been mentioned in section 'Data':

"For the vertical profile the 'beta_raw' variable was used (firmware v1.010 and v1.040) as it readily provided a normalised range-corrected signal (RCS), with instrument-specific overlap correction (usable above 80–100 metres altitude) and harmonisation between different CHM15k systems." (see lines 119–121)

Reference:

Hervo, M., Poltera, Y., and Haefele, A.: An empirical method to correct for temperature-dependent variations in the overlap function of CHM15k ceilometers, Atmospheric Measurement Techniques, 9, 2947–2959, https://doi.org/10.5194/amt-9-2947-2016, 2016.

39. line 124:

"Humidity levels were similar when there was sufficient mixing ..."

Relative humidity (RH) is not conserved during vertical mixing processes. When RH stays constant with height this just means that air temperature and dew point increase or decrease (both is possible, especially if specific humidity varies with height) at the same rate with height. RH is thus not a reliable indicator for the height of a mixed layer. Reliable indicators are virtual potential temperature and specific humidity.

We apologise for putting text in the manuscript that is easily understood different from what was intended to describe. We have changed the text in the manuscript to describe the process better.

In section 2.2 where we describe the pre-processing and annotation it is mentioned that we chose the Cabauw data set to learn how to best annotate the data manually. Because of the availability of ancillary data, such as relative humidity (RH) information at various height levels from the in-situ observation program in the tower. In some cases we used the RH

tower data to check our manual annotations. These were cases where a layer could be seen by eye in the graphical representations of the ceilometer data at close range, i.e. low altitudes within the height range of the mast. So, for most cases that were used to annotate, the RH data from the tower was available, but we should like to mention here that the RH data was only used in those cases where the RH data could actually be used to support our manual assessment of the atmospheric conditions.

The manuscript has been changed as follows:

"The layer attribution started with identification of the nocturnal layer structure before sunrise by tracking backwards in time from the high-confidence MBLH identified after sunrise. At this point, the RH profile confirmed when the unstable boundary layer rose above 200 m (which generally coincided with the convective ejection patterns in the RCS). During nighttime, the pollution rich layers may drop to very low altitudes into the incomplete-overlap region of the CHM15k ceilometer and vertical mixing in fact ceases to exist. However, the concentration levels of pollutants remain layered and, therefore, Cabauw mast measurements were used to aid in the identification of the presence and height of nocturnal layers. When RH at various height levels were similar, we considered this to be a more or less homogeneous nocturnal layer, and the layer height was approximated as the altitude where the RH values of higher layers with respect the lower layers started to diverge. For example, Fig. 3 indicates a MBLH between 40 and 140 meters around 4am UTC on 8 June, which slowly rose to 200 meters around 7am. Note that for annotated layer heights below 100 metres it would be more difficult for the detection algorithm to recover the exact layer height, because of the no-overlap region of the ceilometer. If the MBLH of the following day was clearly visible in RCS data, it was followed backward in time to identify its formation after sunset." (see lines 168–179)

Further, the following has been added to the Discussion section:

"Combining measurements from different sensors, such as Doppler wind lidar, ceilometer and microwave radiometer could further improve the accuracy of annotations. For example, measurements of specific humidity and virtual potential temperature are useful indicators with respect to the height of the mixing layer." (see lines 369–372)

40. Figure 3:

Ceilometer in the front, tower in the background (only lower half of the tower (van Ulden and Wieringa 1995)

One can also see a scanning microwave radiometer (MWR, left), a scanning cloud radar (CR), a micro rain radar (MRR) and a second microwave radiometer (center), a scanning doppler wind lidar (DWL) and two Distrometers (right).

The caption has been adjusted as follows:

"The ceilometer (in front) and 213-meter mast (background) at the same location at Cabauw support high-quality annotations. This site also includes a scanning microwave radiometer

(left), a scanning cloud radar, a micro rain radar and a second microwave radiometer (center), a scanning Doppler wind lidar and two distrometers (right)." (see Figure 2)

41. line 136:

"Fig. 2 indicates a MLH between 40 and 140 meters around a 4am UTC on 8 June, which slowly rose to 200 meters around 7am"

As above explained there is no mixing layer during the night. Increased backscatter signal in the lower levels just represents the layer with high RH. Beside this any information below 100m can only be derived from the tower data. The Ceilometer is blind in this region. Does it really make sense to train the algorithm with this data?

Regarding the first part of this comment, the manuscript has been adjusted to state that there is no vertical mixing at night.

The following has been added to the manuscript:

"During nighttime, the pollution rich layers may drop to very low altitudes into the incomplete-overlap region of the CHM15k ceilometer and vertical mixing in fact ceases to exist. However, the concentration levels of pollutants remain layered and, therefore, Cabauw mast measurements were used to aid in the identification of the presence and height of nocturnal layers." (see lines 170–173)

The deep learning technique is able to filter out the signal from the noise and therefore builds a robustness against noise in the incomplete overlap region. Note that the Deep-Pathfinder results compare well to STRATfinder in the lower bounds of the altitude range where overlap is incomplete. STRATfinder needs an overlap correction function, while Deep learning learns a function from input image to output image, thereby implicitly taking the instrument behaviour in the incomplete overlap region into account.

For the no overlap region, if there is just noise in the input data, the algorithm could predict the top of the noise layer or it may still predict something more accurate by using valid measurements at times around this point and the patterns observed above 100m altitude. For example, if the layer height was above 100m in the five minutes before, this layer progression would also be visible in the 45-minute input image.

The following has been added to the manuscript:

"Note that for annotated layer heights below 100 metres it would be more difficult for the detection algorithm to recover the exact layer height, because of the no-overlap region of the ceilometer." (see lines 176–178)

42. line 137:

"The transition region from the unstable diurnal to the stable nocturnal MLH may not be clear from aerosol data (Wang et al., 2012)."

You mean the transition from the daytime convective (mixed) boundary layer to a neutrally stratified residual layer (RL) with a stable NBL below. That purely backscatter based MBL retrievals have difficulties to detect this transition has been shown on a statistical basis in Schween et al. (2014).

This has been changed accordingly in the manuscript:

"The transition region from the daytime convective (mixed) boundary layer to a neutrally stratified residual layer (RL) with a stable NBL below, may not be clear from aerosol data (Schween et al., 2014)." (see lines 188–189)

Reference:

Schween, J. H., Hirsikko, A., Löhnert, U., and Crewell, S.: Mixing-layer height retrieval with ceilometer and Doppler lidar: from case studies to long-term assessment, Atmospheric Measurement Techniques, 7, 3685–3704, https://doi.org/10.5194/amt-7-3685-2014, 2014.

43. line 140:

"Fig.3"

You mean figure 2.

This has been corrected. Note that during the revision process the order of figure 2 and 3 was also changed, so in the text it still states "Fig. 3", but it does refer to a different figure now. (see line 192)

44. line 143:

"To differentiate between the stable and convective boundary layer, a night time variable was included."

Two comments on this:

1: There is no reason to identify a surface based layers during night - there is no mixing in the rural NBL. Any increased backscatter is related to certain values of relative humidity but not to mixing.

2: When during night the RL stabilizes layers with varying backscatter develop, probably to height dependent advection of layers with different aerosol (content and hygroskopy). These layers persist in the morning until the growing ML reaches their respective height (see figure 2). All algorithms have problems with this. A good example is the manufacturer algorithm which nicely identifies all of these backscatter layers.

I therefore would exclude the night from the analysis and if necessary provide a

morning hour flag, or even just a variable 'time after sunrise' such that the algorithm can learn by itself what to do when.

As detailed in our response to comment #6, references to mixing at night have been removed. Further, we have now described in the manuscript that the definition of MBLH has been adjusted for consistency with existing methods for MBLH detection and for comparison of results with these reference methods. These reference methods also generate profiles for the full day.

To address this specific comment, we have also performed an experiment by training the model only using daytime data (i.e., only samples were used where the nighttime indicator was False, while the nighttime indicator itself was removed from the architecture). This alternative model achieved good performance for daytime predictions. However, a post-processing approach with quality control flags was considered a bit more flexible for two reasons. First, Deep-Pathfinder results will be in line with other methods which also provide full-day profiles. Second, end users can still decide which information to keep as a post-processing step. For example, quality control flags could be used to flag nighttime estimates for rural NBL, but keep these estimates at built-up locations.

The following has been added to the Discussion section:

"Further, quality control flags could be set for specific atmospheric conditions, for example, so end users can exclude circumstances without vertical mixing. Other examples of useful flags are the occurrence of clouds (e.g., a binary indicator based on any detected cloud base height) and rain, since MBLH is not clearly defined during periods of precipitation." (see lines 401–404)

Finally, the nighttime indicator itself is a valuable component of the neural network architecture. As described in the comment, there are differences in physical processes during daytime and at night and this variable helps the model identify what information is important to focus on.

45. I would like to see some more details of the annotation process. What does the annotater see: the same as we see here in figure 2 including RCS, gradient, the manufacturers MLH detection, the ECMWF data and the RH line plot at the bottom of fig.2 ? Does the annotator follow the MLH for every timestep (12sec) or does the software, used for the annotation, do some interpolation?

The annotator indeed sees the information as in Figure 3. This information was gathered before the annotation process started. The RCS data is leading for the annotations. We have annotated using the open source labelme software using a high-resolution RCS image with 12-sec temporal resolution. Labelme allows a user to select points to create a mask. This was done at the original resolution to follow all the patterns in the RCS data (as presented in Figure 1, but then the high-resolution version). We tried to select the points in such a way that the interpolated line followed the profile visible in the RCS data. If required, more points were added so this was the case.

The following has been added to the manuscript:

"Specifically, for a single day of data, many consecutive points were selected to follow small-scale changes such as convective plumes, intrusions and extrusions visible in the RCS data." (see lines 155–156)

46. And a provocative question: does the color scale of the backscatter image have an influence on where the annotator puts the MLH?

The annotation process looked for regions with high contrast, not for a specific colour. Therefore, it should work, as long as no contrast-minimising colour scale is used.

No changes have been made to the manuscript.

47. I would also recommend to order this section more clearly:

first describe which data you use:

1: Ceilometer from Cabauw +two other,

2: ECMWF MLH extracted from cloudnet data,

3: Ceilometer manufacturer MLH

4: Tower RH

5: and then describe the annotation process.

The structure of this section has been improved. The sentences in subsection "Pre-processing and annotation" that were related to data have now been moved up to the subsection "Data". The same applies to the figure with instruments at Cabauw. As a result, the subsection "Pre-processing and annotation" now mainly describes the annotation process.

The new paragraph in subsection data describes the datasets used for annotation. (see lines 127–144)

The annotation process is described in the following section (see Section 2.2 'Pre-processing and annotation').

48. Fig.4:

I am not an expert in computer based vision or image segmentation and this looks more like a flow chart to me (from input to output). I guess the left branch is the decoder and the right one the encoder. I would appreciate more description: what mean the arrows down, the arrows up, the horizontal arrows, the numbers in the boxes (first two numbers are divided by two from box to box. They start at input

with the dimension of images mentioned in the text. Third number changes in some way but product does not stay constant. First two numbers in boxes connected by horizontal arrows are identical but third number not ...).

The caption has been adjusted as follows to explain this figure more clearly:

"Deep-Pathfinder neural network architecture with input (green), output (blue), skip connections (horizontal arrows), encoder (downward arrows) and decoder (upward arrows from 7x7x321). In each box, the first two numbers indicate the spatial dimensions and the third indicates the number of channels, determining the capacity of the neural network layer." (see Figure 4)

For your information, some existing neural network architectures, such as MobileNetV2 (Howard et al., 2017), provide a width/depth multiplier to scale all the 'third numbers' in the network by a constant to easily adjust the capacity of the whole network. This has not been added to the manuscript.

Reference:

Howard, A. G., Zhu, M., Chen, B., Kalenichenko, D., Wang, W., Weyand, T., Andreetto, M., Adam, H. Mobilenets: Efficient convolutional neural networks for mobile vision applications. CoRR, abs/1704.04861, 2017.

49. line 151:

"latent space representation"

Please explain what this is.

The term 'latent space' is indeed jargon. It is used to refer to the point in the network where the features have been extracted (i.e., the end of the encoder). The values in the 'latent space' can be seen as the compressed data.

For clarification, this sentence has been revised as follows:

"From the latent space representation (i.e., a representation of compressed data), the dimensions are increased again to obtain an output mask (i.e., the decoder)." (see lines 203–204)

50. line 156:

"The input dimensions of the RCS image were 224×224 ..."

This means that the analysis is based on chunks of 224*12sec=44min:48sec - you mention that later but here would be a good point to do that. And I understand that 224 is used because it can be split in many prime factors (224=2^5*7) and because it

fits to the number pixels in the vertical - which of course was chosen for the same reason.

The 224x224 size was indeed chosen for the reasons mentioned in the comment.

The description of the 224x224 image has now been moved here:

"The input dimensions of the RCS image were 224x224 pixels, representing a time period of 44 minutes and 48 seconds and a fixed altitude range of 0 to 2240 meters, preserving the temporal and spatial resolution of the ceilometer data." (see lines 208–210)

51. I learn now that the third number in the boxes in fig.4 is the number of features. How do you choose the number of features for every step of the decoding and encoding process ?

We have experimented with different layer sizes (i.e., number of features). This is common practice in applied neural network research. Typically, if the numbers are too low, the model will have bad performance as it cannot capture all important features in the data. If the number of features is too high, the network will contain a lot of trainable parameters. This has two consequences: (i) more input data is required to avoid overfitting, (ii) model training will require a lot more computation resources. The first can be diagnosed by monitoring performance on the validation set.

For the encoder, we used the same layer dimensions as MobileNetV2. We experimented with the width multiplier parameter to scale the layer depths, but the basic MobileNetV2 layer depths already performed well, so those were kept. For the decoder, we also obtained layer depths through experimentation. The details of these experiments are a bit too much detail for the paper since it is a common process.

For the encoder description, the following has been added to the manuscript:

"Using a sequence of MobileNetV2 layers, a 7x7 block with 320 features was extracted from the input image." (see lines 210–211)

The following was added to the manuscript after the decoder description:

"Layer depths (i.e., the number of features per layer) were obtained through experimentation." (see lines 215–216)

52. line 164 / Section 2.4. Model Calibration

I am missing an information which part of the data was used for this pretraining:

All, the first year, only selected days ?

All data was used (~125 GB).

For clarification, the following was added to subsection 'Data':

"Model pre-training (see Sect. 2.4) used all ceilometer data from Cabauw, De Kooy, Groningen Airport Eelde, Maastricht Aachen Airport and Vlissingen." (see lines 127–128)

In subsection 'Model calibration', the phrase "using all data" was added to the text:

"Therefore, part of the network was pre-trained using all data to aid the calibration on limited annotated samples afterwards." (see line 222)

> 53. line 168:
>
> "Removing the skip connections was necessary as information would otherwise flow directly from input to output without passing through"
>
> I do not understand why it is then build into the algorithm ?

The usefulness of the skip connections depends on the task of the algorithm.

In our research, there are two tasks:
- The pre-training task is to reconstruct the RCS data (input image = output image)
- The fine-tuning task is to create a mask based on the RCS data

For the first task the skip connections are not good, because information would otherwise flow directly from input to output without passing through.

For the second task the skip connections are very helpful, as they allow for increased local details. That is, when increasing the spatial dimensions in the second half of the network, information in the corresponding location of the (partly) encoded image is used to determine where exactly to draw the details.

The latter is explained in more detail in the original U-Net paper by Ronneberger et al. (2015), which is referenced in the manuscript.

The manuscript has been changed to clarify that this is undesirable behaviour for the pre-training task:

"Removing the skip connections was necessary as information would otherwise flow directly from input to output without passing through the encoder/decoder structure, which is undesirable behaviour for the pre-training task." (see lines 225–227)

Reference:

Ronneberger, O., Fischer, P., and Brox, T.: U-Net: convolutional networks for biomedical image segmentation, in: Medical Image Computing and Computer-Assisted Intervention – MICCAI 2015, edited by Navab, N., Hornegger, J., Wells, W. M., and Frangi, A. F., pp. 234–

241, Springer International Publishing, Munich, https://doi.org/10.1007/978-3-319-24574-4_28, 2015.

54. line 172:

"Given the temporal resolution of 12 seconds, a total of 6976 different images can be extracted from a full day of data,..."

This means you generate for every timestep (12sec) a new image and between subsequent images you have an overlap of 99.6%. Is this large overlap necessary ?

Yes, it is in line with typical data augmentation strategies for neural network training.

Assume you have a fixed set of images which have no overlap. Rather than showing the same image multiple times during model training, it is better to slightly shift the image each time so the model has to learn how to deal with a different combination of pixel values. This will make the model more robust.

For further reading, please see the following guide to data augmentation for computer vision, which has a more elaborate description:
https://towardsdatascience.com/complete-guide-to-data-augmentation-for-computer-vision-1abe4063ad07

No changes have been made to the manuscript.

55. line 177:

"the model reached convergence with a reconstruction loss close to 0."

Can you explain what this means.

At the start of training a loss function is defined to measure the performance on the task you have specified. In this case the pre-training task was to reconstruct the RCS image. When training the model, this loss was minimised using a standard optimisation procedure. The sentence "the model reached convergence with a reconstruction loss close to 0" means that the minimisation of the loss function was successful and the final value of the loss function of the trained model was close to 0. This means that the model accomplished the task it was given. In this case, it led to very good image reconstruction (i.e., the output image from the model looked very much like the input image).

The following has been changed in the manuscript:

"with a loss close to 0 for the image reconstruction task (see Fig. 5)" (see line 233)

For clarification, a new figure has been added to the manuscript to visualise the results and show that the reconstruction loss was indeed very low:

[Figure]

"Figure 5. Visualisation of the completed pre-training process, showing in each column a grayscale input image (top) and the image reconstructed by the autoencoder (bottom)."
(see new Figure 5)

56. line 180:

"Unsupervised pre-training of encoder weights outperformed randomly initialising weights or loading ImageNet weights (results not presented)."

Based on which metric do you know that it outperformed the untrained model ?

Performance was evaluated on the fine-tuning task, initialising the network with either the pre-trained weights, random weights, or ImageNet weights. We evaluated performance on the validation set. Two metrics were used: the value of the loss function (binary crossentropy) and the accuracy. The accuracy metric determined for each pixel whether the prediction was white or black, using a threshold of 0.5. Then, the percentage of pixels that was the same as the mask was computed.

This statement better fits the Discussion section, so it has been moved there.

The following has been changed in the manuscript:

"Several experiments also indicated that unsupervised pre-training of encoder weights outperformed randomly initialising weights or loading ImageNet weights (Deng et al., 2009). This was assessed using the validation set, based on the binary crossentropy loss and the accuracy of the generated masks (results not presented)." (see lines 389–392)

Reference:

Deng, J., Dong, W., Socher, R., Li, L.-J., Li, K., and Fei-Fei, L.: ImageNet: a large-scale hierarchical image database, in: 2009 IEEE Conference on Computer Vision and Pattern Recognition, pp. 248–255, IEEE, Miami, FL, https://doi.org/10.1109/CVPR.2009.5206848, 2009.

57. What did the result of this pretraining look like ?

For clarification, the new Figure 5 has been added to the manuscript to visualise the results. Also see our response to comment #55.

58. Does the network at this stage already recognize the ML ?

See the new Figure 5 in the manuscript.

The network recognises all information in the RCS image. However, it is not aware yet which layer we find important to follow. Therefore, the fine-tuning step was developed.

59. Why is additional training on annotated data necessary ?

This has been addressed by adding the new Figure 5, which also answers this question. The pre-trained model just reconstructs the input image; it is a different task. The task that is useful for end users is mask prediction, not image reconstruction. However, the image reconstruction tasks helps the model to learn to extract useful features from RCS data, leveraging all the information in the available unlabelled data. This then leads to a better model for the task of mask prediction and limits the amount of labelling that is required.

60. I am also not sure how the terms 'pre-training' and 'model calibration' are related to each other.

Please clarify.

Model calibration is the complete process of optimising the weights inside the model (i.e., pre-training and fine-tuning combined). Therefore, the subsection header is 'Model calibration'. Pre-training is the first step in the model calibration process.

The following has been added at the start of the subsection 'Model calibration' for clarification:

"The model calibration process consisted of two steps: (i) pre-training and (ii) fine-tuning." (see line 220)

61. I also do not understand how unsupervised training leads to a meaningful segmentation of the images. Please explain.

Unsupervised pre-training does not directly result in a segmentation model. The resulting model of this first step can only reconstruct images.

However, it assists in performing the second task (i.e., the fine-tuning process), where the segmentation takes place.

For clarification, the pre-training task has been more clearly described in the manuscript:

"The unsupervised pre-training task was to reconstruct the ceilometer data, meaning the input RCS image was also used as the target image." (see lines 223–224)

62. line 182:

"transfer training"

What is that ?

For clarification, the following sentence has been added to the manuscript:

"Transfer learning is a learning paradigm for adapting a model to perform a similar task, exploiting the knowledge captured during the previous task." (see lines 236–238)

63. line 182:

Sentence starting with

"During model calibration, each batch ... "

This sentence is not clear to me. Please reformulate.

If I understand right you extract from every annotated day 16 subimages of size 224x224.

These images are then used as training material. Why not use the full set of annotated images ?

This explanation requires some additional background information. The model fitting process of a deep learning model is typically performed on one or more graphics processing units (GPU), which can handle many computations in parallel. This is commonly implemented by supplying batches of samples to the GPU (rather than one by one) to keep the GPU close to full utilisation.

This section in the manuscript describes how these batches were constructed. We randomly extracted 16 images from a day, as this was the maximum batch size that fitted in the available GPU memory. After processing one batch and updating the model weights, a new batch was supplied to the GPU, etc. The next batch will again be one of the (randomly selected) 50 training days and a random selection of 16 samples from this day. During model training, the model will process a very large number of batches. So, as training progresses, the model will over time see all the annotated images.

For clarification, the order of sentences has been changed and some additional information has been added. This paragraph has been rewritten as follows in the manuscript:

"For the mask prediction task, the neural network model now had to predict an annotated mask of 224x224 pixels. The model was provided with both the RCS image and the nighttime indicator as inputs, while the corresponding annotated mask was used as the target image. For illustration purposes, Fig. 6 shows three sample training image pairs. Samples were grouped in batches for computation purposes. These batches were constructed by randomly selecting a day from the training data and subsequently extracting a random batch of 16 image pairs of 224x224 pixels from the full 24-hour RCS image and corresponding annotated mask. Although optimal randomisation would be obtained if one batch contained samples from various days, this implementation choice resulted in limited pre-processing overhead and full GPU utilisation, while still iterating through the full set of annotated images." (see lines 238–245)

64. line 184:

image pairs are RCS image and image mask from the manual annotation ?

Yes, that is correct.

This paragraph has been rewritten for clarification. See our response to the previous comment.

65. line 186:

Sentence starting with

"For each 45-minute sample ..."

seems to be incomplete.

This sentence has been removed from the manuscript.

Further, the time period that one sample covers (i.e., 44 minutes and 48 seconds) has now been described in section 'Model architecture'. See the response to comment #50.

66. line 189:

"... several ..."

Fig 5 shows exactly three pairs

'Several' has been replaced by 'three' in the manuscript:

"For illustration purposes, Fig. 6 shows three sample training image pairs." (see line 240)

67. line 190:

"350,000 samples"

> I thought for every of the 50 days with annotated masks 16 pairs where extracted that would give only 800 pairs or samples. Where does this large number come from ?

As explained above, we extract 16 random pairs from an annotated mask many times during training. Over time, all annotations are seen by the model. The description in the manuscript was changed to make this more clear:

"These batches were constructed by randomly selecting a day from the training data and subsequently extracting a random batch of 16 image pairs of 224x224 pixels from the full 24-hour RCS image and corresponding annotated mask. Although optimal randomisation would be obtained if one batch contained samples from various days, this implementation choice resulted in limited pre-processing overhead and full GPU utilisation, while still iterating through the full set of annotated images." (see lines 241–245)

The number of samples was calculated as follows. There are (24 hours * 60 minutes * 60 seconds) / 12 seconds = 7200 horizontal pixels in one 24-hour RCS image. By moving an image with 224 pixels horizontally one pixel at a time, we can extract 7200 – 224= 6976 different images per day. This number is also mentioned in the manuscript in subsection 'Model calibration'. For 50 annotated days, this gives 50 * 6976 = 348,800 images.

Therefore, no changes have been made to the phrase "approximately 350,000 samples". (see line 246)

68. line 191:

   labelled data is equal to annotated images ?

Yes, that is equivalent.

69. line 192:

   "A small validation dataset for one additional annotated day was used to tune model hyperparameters"

   what are your "hyperparameters" ?

The main hyperparameters that we experimented with were the learning rate, batch size, and layer depth multiplier.

The following has been added to the manuscript:

"A small validation dataset for one additional annotated day ($n$ = 1396 samples, including 765 daytime and 631 nighttime images) was used to tune model hyperparameters, such as the learning rate, batch size, and layer depth multiplier." (see lines 247–249)

70. The additional day mentioned here is a 51st day which has been annotated?

Yes, that is correct.

71. How was this day selected?

This day had a variety of conditions at the start of the day (e.g., multiple residual layers, clouds) and a clearly defined convective mixing layer later on. It was important not to select a day that was very ambiguous for annotation, as the 1396 samples extracted from this day were used to provide an indication of model performance and tune hyperparameters.

72. How can one single day with its specific situation be used to tune and improve the results?

Hyperparameters influence model performance in a more generic way. They are not used to tune the model to perform well for a specific day or situation (at least not in our study).

For example, one of the hyperparameters is the learning rate of the algorithm. This determines how much to update the weights inside the model for each batch of images. Setting it too low will mean the model will take a long time to converge. Setting it too high means the loss function will not reach the global minimum, because it will overshoot the target. So, the hyperparameters influence the optimisation process. Different settings can have quite a large impact on results. We also want to make sure that we do not overfit the model on training data. That is why a small holdout set (i.e., the validation set) is necessary to fine-tune these hyperparameters and ensure the optimisation process works well.

Note that the full evaluation is performed on a separate test set. So, three independent sets are used in this research: the training set, the validation set, and the test set.

No changes have been made to the manuscript.

73. lines 193 - 199:

"The main selection criterion for model evaluation was the mean accuracy of the generated masks in the validation set,..."

I cannot follow:

You are talking here about a //validation dataset// which is used to //tune 'hyper'parameters//. And this set is selected because it performs good.

In my humble opinion an evaluation should use randomly selected data which has not been used for the regression or training to test the performance of a method.

Please clarify.

Our setup is correct, as we use the standard approach that is used in computer science.

To clarify:
There are three independent dataset: a training set, a validation set, and a test set.
- The training set is used for model training.
- The validation set is used to check model performance on hold-out data. However, in deep learning research there are also hyperparameters that need to be tuned through different experiments. These should not be tuned on the training set, as it will lead to overfitting
- Therefore, a third dataset is required for evaluation, which is referred to as the 'test set'

We have described this in the manuscript as follows:

"Finally, a test set was constructed with ceilometer data not used for mask prediction, spanning the second half of 2020 at Cabauw, to obtain the unbiased performance of the final tuned model." (see lines 255–256)

Our approach is standard practice. There are many resources available describing the process, the following is an example which could be used for further reading in case of interest:

https://kili-technology.com/training-data/training-validation-and-test-sets-how-to-split-machine-learning-data

*Figure 5.:*

74. The figure shows the outcome of the visual-manual annotating process. It therefore would rather fit into section 2.2.

Figure 3 shows the visual-manual annotation process. The figure referred to in this comment (currently Figure 6) is the next step, showing the construction of randomly cropped samples that are used for model calibration.

For a previous comment we have added the new Figure 5 with the results of the pre-training phase. It makes more sense to have this new Figure 5 appear before the current Figure 6, but the new Figure 5 cannot be moved to section 2.2. Therefore, no changes have been made.

75. The letters a,b,c used in the caption are not in the figure.

The letters a, b and c have now been added below the figure.

76. An annotation of time and height would be helpful,

This is not provided for a reason, as this information is not supplied to the model. Only the data shown in Figure 6 plus the binary nighttime indicator is provided to the model. We do

not pass details such altitude or time information separately to the model, other than what is encoded in the pixels.

77. Are times given in LT or UTC?

We have now converted all times to UTC, for consistency throughout the manuscript.

The second sentence of the revised caption has been changed as follows:

"The presented samples are from Cabauw on (a) 15-Sep-2019 at 8:49am–9:34am UTC, (b) 03-Sep-2021 at 9:33am–10:18am UTC, and (c) 14-Jun-2021 at 9:46pm–10:31pm UTC." (see Figure 6)

78. These three samples are manually annotated images which are used for the training - please include this information in the caption..

The caption has been revised to include this information.

The first sentence of the caption is now:

"Three samples of image pairs for model fine-tuning, consisting of a grayscale input image (top), a nighttime indicator (not shown) and a manually annotated output mask (bottom)." (see Figure 6)

79. All three examples show cloud free morning to noon hours. It would be interesting to see also other times like e.g. morning (sunrise, start of convection), early afternoon (fully developed ML), late afternoon (breakdown of convection), evening (sunset...), scenes with clouds etc.

This is not an illustration of the annotation process. We just show here how some samples look like, so readers can understand the model calibration process.

80. Subfig.a is from June 2019. although according to section 2.1. data was from *June 2020* to February 2022.

That is true, some data outside of the June 2020 to February 2022 period was used. For example, the validation set was also data from 2019 so it was independent from the other sets. This has now been described more accurately in the manuscript.

In section 2.1 the following was added:

"Model fine-tuning used a subset of data from Cabauw, De Kooy and De Bilt and some days outside the June 2020 to February 2022 period (e.g., for validation), which were manually annotated." (see lines 128–129)

81. Why has example c the lowest MLH although it is noontime in Summer ?

Fig. 6c shows RCS data recorded from 11:46pm – 12:31am local time (9:46pm – 10:31pm UTC). This is nighttime, rather than noon.

No changes have been made to the manuscript.

> 82. I even have the feeling that inside the identified ML ondulating layers with different RCS are visible. That would mean that there is no mixing.

See our response to comment #6 for the definition of the layer that we are annotating during nighttime. At nighttime an aerosol layer was annotated, not necessarily have mixing characteristics.

> 83. Why is the noisy layer at the bottom in example a higher than in the other two ?

Fig. 6a used measurements from 2019, while Fig. 6b and 6c used data from 2021. The ceilometer at Cabauw had the optical module replaced during that period, and that is why the measurements show different noise levels. The optical module's ID used for Fig. 6a was TUB150037 and for Fig. 6b and 6c this was TUB160060.

These variations in the training data help the model in becoming robust against changes in instrument characteristics, such as optical modules.

> 84. In the middle of b an intrusion of light gray (=aerosol less air?) from above is visible. It persists some minutes at lower levels while above again darker pixels are visible. This is not represented by the ML mask below. What are the rules to deal with such intrusions and similar with extrusions?

The rules for the annotation process were to provide guidance at the highest level we thought was possible (i.e., following the intrusions and extrusions at the highest available resolution of 12 seconds).

For clarification, the following was added to the 'Pre-processing and annotation' section:

"Specifically, for a single day of data, many consecutive points were selected to follow small-scale changes such as convective plumes, intrusions and extrusions visible in the RCS data." (see lines 155–156)

With the provided annotations, the algorithm started recognizing these high-resolution fluctuations. Annotations can subsequently be improved in further iterations. This has been discussed in the Discussion section, as follows:

"Fine-tuning can be an iterative process, where the current shortcomings of the model are used to slightly improve specific annotations in the training data. The model can then be retrained using the updated annotations to enhance performance." (see lines 395–396)

> 85. What would show the outcome of the non supervised training?

The outcome of the unsupervised training has been added in the new Figure 5.

86. line 201:

"correspond" not corresponding

This sentence has been changed as follows:

"The generated masks correspond to a 45-minute period, rather than providing a 24-hour sequence of MBLH values." (see lines 259–260)

87. line 202:

"First, output masks were generated ... using one-minute intervals ..."

I thought model output is a 224x224x1 ML mask (see fig.4) representing about 45min in time.

How can the model produce one minute intervals with a length of 5pixels ?

This requires some clarification:

The input data is processed as follows. The first image of the day is processed by the model to generate a mask. Then, the input image is shifted five pixels (i.e., one minute) and this image is processed by the model. Then, the input image is shifted again by five pixels, etc.

So the input images overlap and, hence, the generated masks overlap. These are then averaged (i.e., the multiple predictions for each time point) to obtain the full 24-hour mask.

For clarification, this sentence has been expanded in the manuscript as follows:

"First, multiple output masks were generated via model inference. Specifically, the input image was repeatedly shifted by five pixels (i.e., one minute) and processed by the model, until the entire day was processed." (see lines 260–262)

88. line 203:

"... overlapping predictions for each time step, which were averaged"

If I understand right you average the discrete black (in ML) or white (outside ML) values of the ML mask to something in between (some grey tone). Is this done by just arithmetically averaging the white and black values (0 and 1 or 0 and 255 or whatever)

Please explain that.

The model is trained with discrete black/white masks. However, the calibrated model already predicts grey tones for a single image. This is an indication of the model's uncertainty. If the model assesses it is highly likely that a pixel is located within the mixing layer (e.g., at low altitudes), the model output for that pixel will be (near) black. Pixels near the top of the image are typically (near) white. In more difficult situations, it could be any grey tone.

For clarification, an additional sentence has been added at the start of the 'Model inference and post-processing' subsection:

"Model inference using the calibrated model resulted in a generated mask with for each pixel a prediction of (near) white or black, or a grey tone in case of uncertainty." (see lines 258–259)

Further, the following has been added:

"This led to overlapping predictions for each time step. Multiple predictions at each unique time and altitude combination were arithmetically averaged to obtain a full 24-hour mask." (see lines 262–263)

> 89. line 205 - 210:
>
> "A loss function was formulated ..."
>
> With this text I do not understand what is done. Please be more precise.
>
> I guess you mean that pixels are counted and this count is interpreted as a loss function and you search the minimum of it. But are only pure black and pure white pixels considered or also grey ones lying above/below a certain threshold (0,0.5,1)? Where comes the softmax or sigmoid function in play? Do you use 'the sigmoid function' or 'the softmax function' or a sigmoid like function?

The sigmoid function is the final step in the neural network. For each pixel, it converts the output in the final node (ranging from -infinity to infinity) to a range of [0, 1]. Hence, the prediction is black, white or any grey tone in between. The loss function was formulated to also consider the effect of these grey predictions. Hence, it takes into account the model uncertainty when extracting the MLH estimate.

For clarification, these sentences have been rewritten as follows:

"Note that model predictions display varying degrees of uncertainty, with pixel predictions being either black, white or any grey tone in between (i.e., the result of the pixel-wise sigmoid function at the end of the neural network). The loss function took into account model uncertainty by proportionally penalising the number of pixels below $p$ that were predicted as non-black, plus the pixels above $p$ that were predicted as non-white. MBLH at time $t$ was estimated as the value $\hat{p}$ that minimised this loss, multiplied by the spatial resolution of 10 metres." (see lines 266–271)

90. line 211:

This trailing paragraph of section 2.5 is rather an introduction to section 3.1.

It has now been moved to the start of section 3.1, as an introduction to that section.

Note that several sentences from this paragraph have been moved to section 2.1 in response to the comments below. Further, the sentence about the test set was related to methodology and has now been removed, as it did not fit well in the 'Results' section and was already described in section 2.4.

The remaining sentences were changed slightly and now form the introduction to section 3.1, as follows:

"Deep-Pathfinder performance was compared to MBLH estimates from STRATfinder and manufacturer Lufft for all days in the test set, showing the out-of-sample performance on new data not used for model calibration." (see lines 274–275)

91. line 211:

"... proprietary algorithm of the ceilometer manufacturer based on wavelet covariance transform ... "

You already mentioned the algorithm in section 2.1. Add the information about the principle there and refer here only to the respective section.

This has been moved to section 2.1, where it has now been described as follows:

"This proprietary algorithm, based on wavelet covariance transform, identified multiple candidate layers from the RCS data, where the lowest identified layer can typically be interpreted as the MBLH." (see lines 139–141)

92. line 213:

"... STRATfinder was still in active development ..."

The same strategy for STRAT as for the manufacturer algorithm: describe it in section 2 and refer here only to the section.

The following has now been described in section 2.1:

"Model evaluation used MBLH estimates from two reference methods: (i) Lufft's wavelet covariance transform algorithm and (ii) a state-of-the-art detection algorithm, STRATfinder (Kotthaus et al., 2020). At the time of publication, STRATfinder was still in active development and the STRATfinder data for this study was received from Institute Pierre Simon Laplace (IPSL) on 6 May 2022." (see lines 141–144)

93. line 214:

Acronym IPSL not explained.

This has been changed in the manuscript as follows:

"Institute Pierre Simon Laplace (IPSL)" (see lines 143–144)

Further, in the 'Acknowledgements' section "Institute Pierre Simon Laplace" has been replaced by the acronym "IPSL". (see line 450)

94. line 215:

"... data from the second half of 2020 at Cabauw was used as a test set,..."

I lost oversight which parts of the dataset are used for (pre-)training, annotation and which for this inter comparison, please clarify that at the beginning, at best in section 2. Second half of 2020 means July-December and thus 2/3 of the Summer, full Autumn and first month of winter. Why not a full year ? This means calm autumn days dominate the statistics of this comparison.

This sentence was removed to avoid confusion. It has now only been described in section 2, where it has been adjusted as follows:

"Finally, a test set was constructed with ceilometer data not used for mask prediction, spanning the second half of 2020 at Cabauw, to obtain the unbiased performance of the final tuned model." (see lines 255–256)

Regarding the length of the test set, we agree with the comment. We also preferred to use a full year of data. The reason that we did not use a full year is that we needed data for all reference methods to be available. However, the STRATfinder MLH estimates that were made available to us were not complete for the first half of 2021.

95. line 220:

"the benchmark methods"

I am not very content with the term 'benchmark': it has several very different meanings in geography, business or computer technology. I would rather call them 'reference methods' and introduce them with this name in section 2.

Occurrences of "benchmark method" have been replaced with "reference method", as follows:

"Qualitative and quantitative results showed competitive performance compared to two reference methods: the Lufft and STRATfinder algorithms, applied to the same dataset." (see lines 13–14)

"These methods provide good baseline performance and are frequently used as a reference for the development of new methods." (see lines 40–41)

Further the methods are now introduced as reference methods in section 2:

"Model evaluation used MBLH estimates from two reference methods: (i) Lufft's wavelet covariance transform algorithm and (ii) a state-of-the-art detection algorithm, STRATfinder (Kotthaus et al., 2020)." (see lines 141–142)

> 96. line 224:
>
>> "... Fig. 6, top row"
>>
>> Use letters a, b, c, ... to identify sub plots.

Both the figure and manuscript text have been updated.

The following changes have been made to the manuscript text:

"In Figs. 7a and 7b, before sunrise, the Lufft algorithm jumped between several aerosol layers which developed inside the residual of the mixing layer of the day before." (see lines 278–279)

"For example, a sudden jump in MBLH is visible in Fig. 7b for both Deep-Pathfinder and STRATfinder, although at a different time." (see lines 285–286)

"An example is provided in Fig. 7c, where Lufft estimates were only available after 8pm UTC and not during low cloud conditions." (see lines 288–289)

"In a few cases, STRATfinder predictions were missing due to quality control flags (e.g., Fig. 7d)." (see line 290)

"During the precipitation event in Fig. 7a around 7 to 9pm UTC, Deep-Pathfinder has been trained to predict 0 (i.e., not applicable), while Lufft predictions jumped to about 2,500 meter altitude." (see lines 290–292)

"The example of Fig. 7e shows that for multiple cloud layers Deep-Pathfinder and STRATfinder typically followed a different layer." (see lines 292–293)

"When a clear CBL was not apparent (e.g., Fig. 7f), Deep-Pathfinder and STRATfinder were still able to track the shallow MBLH as intended throughout the day." (see lines 294–295)

97. line 224:

"... jumps between several residual layers ..."

These are not different RL's but aerosol layers which developed inside the residual of the ML of the day before. This is a typical problem of solely aerosol based algorithms: these aerosol layers show strong gradients or steps in backscatter which are then detected as 'the MLH' although they are result of no mixing between the layers.

This has been changed in the manuscript:

"In Figs. 7a and 7b, before sunrise, the Lufft algorithm jumped between several aerosol layers which developed inside the residual of the mixing layer of the day before." (see lines 278–279)

98. line 225:

"Deep-Pathfinder and STRATfinder correctly identified the nocturnal MLH around 100-200 meters..."

I repeat: There is no mixing at night. What you can see sometimes is increased backscatter due to RH above 60-80%. What you see in the first hours of 02-10 and 06-08 is that Deep-Pathfinder sticks to the top of the noisy layer i.e. the lowest layer were the ceilometer can detect anything. You probably annotated the top of the noisy layer as a ML during nights with low aerosol. In such situations the algorithm has to be trained that there is no detection possible - as you did for precipitation.

See our response to comment #6, where the issue related to no mixing at night was addressed.

It may still be useful to annotate a layer height here for consistency with most other published algorithms, which also produce full day profiles. Annotating as 0 is a possible solution, but it can also be addressed in post-processing. For example, a quality control flag can be defined that notes if the top of the noisy layer is predicted during nighttime. The section about post-processing in the Discussion has been extended to give a few examples of when quality control flags can be useful.

The following has been added to the Discussion:

"Further, quality control flags could be set for specific atmospheric conditions, for example, so end users can exclude circumstances without vertical mixing. Other examples of useful flags are the occurrence of clouds (e.g., a binary indicator based on any detected cloud base height) and rain, since MBLH is not clearly defined during periods of precipitation." (see lines 401–404)

Finally, 'correctly' has been removed from the sentence mentioned in the comment.

The manuscript was changed as follows:

"Deep-Pathfinder and STRATfinder identified the nighttime MBLH around 100–200 meters altitude," (see lines 279–280)

> 99. I realize that STRATfinder provides no MLH below ~100m (minor ticks would be helpful). If it does not find a MLH above it seems to provide this height. Is this a valid strategy? Check the description of STRATfinder and discuss.

Minor tick marks have been added and the visibility of the major tick marks has also been improved. (see Figure 7)

We believe annotating the top of the noisy layer is better in this case. If one cannot find any MLH above the noisy layer, the error would be smaller if the top of the noisy layer is annotated rather than a fixed height above it. Further, sometimes the visible layer height is below the STRATfinder minimum but above the no-overlap region. In this case Deep-Pathfinder can still detect this height, while STRATfinder will not.

The following has been added to the manuscript:

"Instead, Deep-Pathfinder was still able to process the noisy signal in the incomplete-overlap region." (see line 281–282)

> 100.     line 229:
>
> "All algorithms had difficulties capturing the decline in MLH around sunset, which is a typical limitation for MLH detection based on aerosol observations."
>
> ... see Wang et al 2012 or Schween et al 2014.

This has been added to the manuscript:

"All algorithms had difficulties capturing the decline in MBLH around sunset, which is a typical limitation for MBLH detection based on aerosol observations (see Wang et al., 2012; Schween et al., 2014)." (see lines 284–285)

References:

Schween, J. H., Hirsikko, A., Löhnert, U., and Crewell, S.: Mixing-layer height retrieval with ceilometer and Doppler lidar: from case studies to long-term assessment, Atmospheric Measurement Techniques, 7, 3685–3704, https://doi.org/10.5194/amt-7-3685-2014, 2014.

Wang, Z., Cao, X., Zhang, L., Notholt, J., Zhou, B., Liu, R., and Zhang, B.: Lidar measurement of planetary boundary layer height and comparison with microwave profiling radiometer observation, Atmospheric Measurement Techniques, 5, 1965–1972, https://doi.org/10.5194/amt-5-1965-2012, 2012.

101.    line 231:

"... a considerable amount of MLH estimates of the Lufft algorithm were missing due to quality control flags. An example is provided for 9 July 2020,..."

Quality control flags should ensure that a meaningful retrieval is possible. If the quality is bad no retrieval should be done. You could discuss which flags lead to no retrieval.

Especially 09-07 is very difficult and I am surprised that deep pathfinder provides a MLH:

The white areas below your retrieved MLH is precipitation partly not reaching the ground (check data from the Cabauw MRR and the Distrometers). Somewhere above it was stated that in case of precipitation the Deep-Pathfinder should deliver zero as indication for no retrieval possible. It seems as if this works not here. Instead Deep-Pathfinder as well as STRATfinder identify cloud base as MLH (recognizable on the darkband above, result of saturation of the ceilometer receiver, and noise further up, you may also check the cloud radar). One may discuss whether this is a valid retrieval - I would say no.

This is related to how the training data was annotated. The annotations for cloud base and precipitation are indeed somewhat ambiguous; especially with precipitation partly not reaching the ground. It is possible to improve the model by updating the annotations, which we also describe in the Discussion section, as follows:

"Fine-tuning can be an iterative process, where the current shortcomings of the model are used to slightly improve specific annotations in the training data. The model can then be retrained using the updated annotations to enhance performance." (see lines 395–396)

Quality control flags have also been discussed later on in the manuscript. See Figure 9 in the Discussion section.

102.    line 237:

"When a clear convective boundary layer is not apparent (e.g., 10 December 2020), Deep-Pathfinder and STRATfinder were still able to correctly track the shallow MLH throughout the day."

How do you know that this is the correct detection of a mixed layer. A shallow layer with increased backscatter during night and during calm winterdays might be just the layer with highest RH in a layer with a strong temperature inversion. Beside of that, STRATfinder sticks here over long times to its lowest possible output around 100m, and Deep-Pathfinder sticks during several moments to the top of the noise layer. You may check the potential temperature and humidity profiles from the tower whether there is really a mixed layer or stable stratification with high RH.

First, the definition of layer detection has been changed (see our response to comment #6).

As suggested, we checked the potential temperature and humidity profiles from the tower to show whether there is really a mixed layer or stable stratification with high RH, and to corroborate the validity of our annotations in the specific case of this shallow layer.

[Figure]

Figure R1: Case study at Cabauw on 2020-12-10. From top to bottom: RCS data with Deep-Pathfinder annotations (black), potential temperature, specific humidity, and short wave downward radiation.

We plotted the meteorological data from the tower to better understand this particular case (2022-12-10). Please see Figure R1, where we have zoomed in on the y-axis, now showing 0 – 600 metres altitude. Looking at the potential temperature and specific humidity, we can see that in this example the manual annotation was mostly in line with thermodynamic definitions.

Note that this figure has not been added to the manuscript.

The sentence referred to in this comment was revised in the manuscript as follows:

"When a clear CBL was not apparent (e.g., Fig. 7f), Deep-Pathfinder and STRATfinder were still able to track the shallow MBLH as intended throughout the day." (see lines 294–295)

103.    line 243:

"STRATfinder scored an average correlation of 0.591"

This is a poor correlation considering that both algorithms use the same data source. But it is in agreement with former intercomparisons of MLH algorithms based on Aerosol backscatter. The question arises whether a solely backscatter based MLH detection algorithm is meaningful.

(see Hennemuth and Lammert 2006, Muenkel et al 2007, Traeumner et al 2011, Haeffelin et al. 2011,  Eeresmaa et al 2012, Caicedo et al 2017, Kotthaus and Grimmond 2018, Kotthaus et al 2020)

We think this is a misinterpretation of the results. Both algorithms use the same ceilometer data, but not the same labels. If we would train our algorithm on annotations from the STRATfinder algorithm, the correlation would be much higher.

Instead, we decided to create our own annotations, allowing for high-resolution estimates but also leading to a larger discrepancy between the two methods.

The following has been added to the manuscript for clarification:

"Note that achieving the highest possible correlation was not the goal of our study, as otherwise STRATfinder annotations could have been used for model training. This would have led to better alignment between the methods, although without the capability to create high-resolution predictions." (see lines 303–305)

104.    This is an average of daily correlation coefficients for ~210days. Why don't you provide an overall correlation ? Why don't analysis it season wize?

We have considered this comment and agree that computing an average daily correlation may understate the level of agreement between the methods. On some complex days there are very few observations from the reference methods (due to quality control flags) and

typically the correlation on these days is low. However, this then gets overweighted because these few observations still count as one full day (i.e., when computing the daily mean).

Therefore, the correlations have been recalculated using the full test set, rather than first computing day-by-day. The Table in the manuscript has been updated as follows:

Table 1. Pearson correlation between the time series of Deep-Pathfinder, STRATfinder and Lufft in the July to December 2020 test set.

|  | Deep-Pathfinder | STRATfinder | Lufft |
|---|---|---|---|
| Deep-Pathfinder | 1 | | |
| STRATfinder | 0.706 | 1 | |
| Lufft | 0.425 | 0.325 | 1 |

An analysis for all seasons was not possible, as the STRATfinder data we received was not complete in 2021.

105.     line 247:

(end of section 3.2. correlation analysis)

I would like to see a separate analysis of correlations for the different phases of the development of the ML (morning growth, mature ML or afternoon breakdown).

Eventually it is also worth to look at this during different seasons.

When computing day-by-day, this leads to the same issues as in the comment above, because of missing data of the reference methods.

It is also not possible to create a single time series with all times of 'mature ML' (for example) after each other. This would create sudden jumps in the time series from one day to the next, and results in an artificially increased correlation.

Therefore, an analysis using mean absolute difference instead of correlation seemed more appropriate. In this way, methods could be compared during the different phases of development of the ABL (also see our response to comment #4).

This analysis fits best in subsection 3.3 'Diurnal MLH patterns', where the following table has been added to the manuscript:

Table 3. Method intercomparison showing the mean absolute difference (metres) for the various phases of development of the ABL, obtained using the test set.

| Method | Overall | Time of day (UTC) | | | | |
|---|---|---|---|---|---|---|
| | | 4–8h | 8–12h | 12–16h | 16–19h | 19–4h |
| Deep-Pathfinder vs. STRATfinder | 189.0 | 127.6 | 178.1 | 251.5 | 316.5 | 156.7 |
| Deep-Pathfinder vs. Lufft | 323.5 | 392.9 | 219.9 | 117.7 | 293.3 | 396.5 |
| STRATfinder vs. Lufft | 369.8 | 437.2 | 285.4 | 220.9 | 306.4 | 435.2 |

Further, the following text has been added to the manuscript:

"Importantly, performance varied for the different phases of the diurnal development of the ABL. In the early morning, the ML grows into a stably stratified, unmixed NBL at the surface (i.e., roughly 4–8h UTC). This is typically followed by fast growth into the more or less neutrally stratified RL (8–12h UTC). In the early afternoon, the fully developed CBL grows slowly into the free troposphere (12–16h UTC). During the late afternoon a more or less sudden breakdown of convective turbulence and thus breakdown of mixing occurs (16–19h UTC). Finally, in the evening and night a new NBL and RL develops (19–4h UTC). Table 3 shows that the mean absolute difference between Deep-Pathfinder and STRATfinder was lowest during the evening, night and early morning growth phases. In contrast, the early and late afternoon is where they were least similar (see explanation above). The Lufft algorithm obtained higher estimates than both other algorithms during the evening, night and early morning growth phases, as it had a tendency to follow aerosol or moist layers in the residual layer (see Sect. 3.1). However, Lufft and Deep-Pathfinder estimates were substantially more similar in the early afternoon. During this period, the manufacturer algorithm can be used more confidently." (see lines 316–326)

106.  Figure 6:

Add identifiers like a, b, c, ... to the subplots.

This figure has been updated.

The identifiers (a), (b), (c), (d), (e) and (f) have been added to the subplots. (see Figure 7)

*Table 1:*

107.  The diagonal is of course 1 and the table symmetric about the diagonal. One could use the lower triangular part of the table to present another measure like e.g. standard deviation, mean absolute difference etc. ... (see e.g. Haeffelin et al 2011)

This is an interesting suggestion. Although this leads to better use of space and more dense information, we think this may also make the table more confusing for some readers.

We have decided to remove the upper triangle numbers to remove the duplication in this table. (see Table 1)

108.  I am missing an information on how many datarecords where used.

This has been added to the manuscript:

"Deep-Pathfinder and STRATfinder scored an average correlation of 0.706, based on 250,000 corresponding records of data (see Table 1)." (see lines 299–300)

*Fig.7:*

109.      These are "Mean diurnal patterns..."

The caption has been changed as follows:

"Mean diurnal MBLH patterns per month at Cabauw for Deep-Pathfinder (black), STRATfinder (orange) and Lufft (purple), including interquartile ranges." (see Figure 8)

110.      I am surprised to see that the average night time MLH values are much larger (3-400m) than in the 'typical' examples (e.g. 100-200m on 02-10, 06-08, 13-10, 10-12 in fig.6). These plots show that no MLH retrievals are possible below ~150m (especially the STRATfinder algorithm which 25% percentiles shows a sharp lower boundary).

The six examples were not selected based on average nighttime layer height. Because of the small sample size, some deviations from the mean are possible. For the first hour (0 – 01 UTC), we can see approximately 100m (2-Oct), 100m (6-Aug), 350m (9-Jul), 200m (13-Oct), 550m (5-Jul) and 300m (10-Dec). That is an average of 267m. As indicated in the comment, in Figure 8, the averages per month are around 300-400m. The difference is not unreasonable.

The second half of the comment refers to retrievals below ~150m. For STRATfinder this statement is correct, as no retrievals below 150m are possible due to the lower bound specified in the algorithm. For Deep-Pathfinder it is possible, but it occurs less than 25% of the time, which is why it does not appear in Figure 8 (i.e., the shaded region is [25%, 75%]).

No problems were identified in the computation of the statistics.

*line 254: "In contrast, nocturnal MLH conditions were more stable throughout the months in the test ..."*

111.      As I said: there is no ML during night.

This sentence has been removed. It was replaced by the following text:

"Table 3 shows that the mean absolute difference between Deep-Pathfinder and STRATfinder was lowest during the evening, night and early morning growth phases." (see lines 321–322)

112.      Aside of this night time values in fig.7 show that the STRATfinder never goes lower than about 150m. Similar Deep-Pathfinder MLHs go only occasionally below 150m - probably due to the noise layer due to no-overlap which is about here. Both lower limits confine MLH ranges during night.

First, in section 'Pre-processing and annotation' the following was added to the manuscript:

"Note that for annotated layer heights below 100 metres it would be more difficult for the detection algorithm to recover the exact layer height, because of the no-overlap region of the ceilometer." (see lines 176–178)

Second, related to the description of former Fig. 7 (currently Figure 8), the following has been added to the manuscript:

"Further, Deep-Pathfinder was able to capture low layers at night better than STRATfinder, although it was also limited by the no-overlap region of the ceilometer." (see lines 313–315)

113.       Additionally I am wondering how the algorithm deals with cases when there is no turbulence and accordingly no mixing, and MLH should be zero meters (whole atmosphere between 0 and 224m is not mixed, the mask should be everywhere white.) This seems never to be identified, neither by the traditional algorithms nor by Deep-Pathfinder. But I am sure that this happens: strong surface inversion can only develop if there is no mixing (see van Ulden and Wieringa, 1996). Or if you want to relate it to own sensual experience: imagine calm autumn nights when there is no wind.

This purely depends on the annotations. At the moment, the only white masks that have been annotated was for strong precipitation. This led to large white sections in the masks for some cases. In Fig 7a, the Deep-Pathfinder prediction was also 0 (i.e., a white output mask was generated).

Currently, no layer heights of 0 metres have been annotated for the case of no mixing at night. However, the machine learning methodology is sufficiently flexible to follow a variety of annotation schemes a user may be interested in.

114.       line 256:

"...a tendency to follow residual layers."

As above: these are aerosol or moist layers in the Residual layer.

This has been changed in the manuscript:

"The Lufft algorithm obtained higher estimates than both other algorithms during the evening, night and early morning growth phases, as it had a tendency to follow aerosol or moist layers in the residual layer (see Sect. 3.1)." (see lines 323–324)

115.       line 258:

"... in case of multiple cloud layers our annotations typically followed the lower layer, while STRATfinder followed the higher layer."

To investigate whether this influences the average MLH you could remove times

with a certain amount of clouds (see e.g. Kotthaus et al 2023) and repeat the analysis. I recommend to do this.

We have looked at the peak MLH of STRATfinder and Deep-Pathfinder profiles for many days in the test set, so we have a high degree of confidence in this statement. Fig. 7e provides an example of this situation, which shows what was typically observed. Mathematically, it is clear that a higher peak for STRATfinder influences the average MLH peak. No changes have been made in the manuscript.

116.     line 259:

"Secondly, Deep-Pathfinder MLH estimates fluctuated more by following short-term reductions in MLH, while STRATfinder did not."

But these fluctuations are much smaller than the mean differences between the algorithms. Behalf of that I would expect that an average of the fluctuating MLH over some minutes comes at least within some ten meters close to the MLH based on a smoothed backscatter signal.

This sentence has now been removed from the manuscript.

Further, the description in the manuscript has been updated as follows:

"A gradual decline in peak MBLH can be observed from July and August towards December. On some days, STRATfinder reached higher peak values than Deep-Pathfinder. In case of multiple cloud layers, our annotations typically followed the lower layer, while STRATfinder followed the higher layer. For example, this behaviour can be observed in Fig. 7e." (see lines 310–313)

117.     Line 262 - end of section 3.3

The section ending here contains some hypothesized reasons (different behaviour in case of clouds, short term variability of MLH) for the differences between the algorithms. It would be worth to investigate this in more detail: Does it help to exclude clouds from the analysis, do 1min, 15min, 1h means of the algorithm still deviate from each other ?

The second reason (short-term variability of MBLH) was removed from the manuscript; see our response to the previous comment.

Excluding circumstances with clouds will reduce the quantitative differences between the methods, but this is already known without performing this analysis. The differences do not specifically occur for the situation "clouds" versus "no clouds", but mainly for "single cloud layer" versus "multiple cloud layers". See our response to comment #5.

118.     line 271:

"Note that these experiments did not use any form of unsupervised pre-training."

Is this intercomparison of NN architectures than comparable to the pretrained version?

This is why we also included the results for Deep-Pathfinder without pre-training.

It was described in the following sentences:

"Therefore, the Deep-Pathfinder architecture without pre-training on unlabelled lidar data (referred to as U-Net Nighttime Indicator) was also included for comparison purposes. Further, a simpler architecture without nighttime indicator and no pre-training (U-Net Base) was included, as this indicator was also not implemented for the alternative architectures." (see lines 341–343)

For clarification, the following sentence has been added to the manuscript:

"Hence, results of the alternative architectures can be directly compared to the U-Net Base model." (see line 344)

Further, Tables 4 and 5 also allow for the following analysis. Starting from U-Net Base, one can see that adding the nighttime indicator improves performance (see U-Net Nighttime Indicator results). Subsequently, adding the pre-training step further improves performance (see Deep-Pathfinder results).

This is now described as follows in the manuscript:

"Further, both the U-Net Nighttime Indicator and Deep-Pathfinder architectures performed substantially better than the U-Net Base architecture. This shows the benefits of (i) incorporating sunrise and sunset information explicitly in the model and (ii) unsupervised pre-training on large-scale lidar data to improve feature extraction. For example, overall MAE decreased from 104.5 metres for the U-Net Base model to 74.5 and 64.2 metres for the U-Net Nighttime Indicator and Deep-Pathfinder architectures, respectively." (see lines 354–358)

119.     line 274:

"... was also not ... " instead of "... was no ..."

It would be of interest to shortly explain what the main differences between the neuronal network architectures is.

The word 'also' has been added:

"Further, a simpler architecture without nighttime indicator and no pre-training (U-Net Base) was included, as this indicator was also not implemented for the alternative architectures." (see lines 342–343)

Some more details have been added for the three best performing alternative architectures.

The following has been added to the manuscript:

"These models have substantial architectural differences. For example, TransUNet was based on vision transformers, $U^2$-Net used a nested U-structure, and ResUNet-a used residual connections, atrous convolutions and pyramid scene parsing pooling." (see lines 334–336)

120.        line 275:

"... the MLH was extracted for a full day."

For one single day or all day in the validation set (July - December 2020) ?

This is indeed a bit confusing. The July-December period is the test set, while this experiment reported results on the 1396 samples of the validation set.

The sentence has been rewritten as follows:

"After model training, masks were predicted for all samples in the validation set and the MBLH was extracted (see Sect. 2.5)." (see lines 337–338)

121.        line 276:

"These statistics were computed with respect to the annotations for the validation set,"

Do I understand right that you use a part of the training set (annotations from 2019-2021) to validate the models in the validation set (July-December 2020) ?

No, that is not right. The dataset from July-December 2020 was the test set. This was not used for this analysis (i.e., no annotated masks were available for the whole test set, so MAE and other statistics could not be computed).

This analysis used the training set to train models. Then, the validation set was used to get an indication of model performance. The training and validation sets had no overlap (i.e., they were independent).

122.        line 279:

"A full evaluation on six months of test data was not performed for the alternative architectures."

You try to get an idea on the quality of the different alternative NN architectures by investigating its performance in the validation set. But the validation data set has

been used to improve model performance. Is it not common technique to use an independent "test data set" which has not been used for training and validation of the model ?

Please clarify.

Yes, that is true. However, the performance on the validation set already provides a good indication of how well the model is able to be fitted on ceilometer data. Note that the validation set is used to tune hyperparameters that ensure the model converges (e.g., the learning rate of the algorithm, the batch size, etc.).

The goal of this section is to show what could be recommended for future research. For this goal, we consider the performed analysis as sufficient. This is also why we state the results are 'indicative' of the performance:

"However, the results are indicative of the performance of different model architectures." (see lines 339–340)

123.     line 284:

"nocturnal"

typo.

This sentence was removed.

124.     line 284:

"Mean absolute error was the lowest for the nocturnal MLH"

Of course MAE is smallest during this time of the day as ceilometer backscatter is limited at the bottom by the no-overlap region and at the top by the position of the moist non-mixed layer.

This sentence was removed and replaced by the following:

"The growth of the MBL in the early and late morning, and the fully developed CBL in the early afternoon were captured best." (see lines 348–349)

125.     line 284:

"MAE was substantially higher for the late afternoon decay in MLH,"

Your afternoon extends into the night (16-24UTC) where differences can be expected to be small again (see fig.7). A better split of time would be 16-19UTC . And I am still convinced that a night time analysis makes no sense.

For consistency, the analysis has been rerun to incorporate the same time splits as in Section 3.3. That is, MAE and correlation were computed for 4–8h UTC, 8–12h UTC, 12–16h UTC, 16–19h UTC and 19–4h UTC (see Tables 4 and 5). After adding these additional columns for the different ABL phases, we felt that this section had too many tables with many statistics. Therefore, the MSE table was removed, as it showed very similar patterns as the MAE results.

Further, the description in the manuscript was updated to describe the results in terms of the different ABL development phases.

The following has been added to the manuscript:

"The growth of the MBL in the early and late morning, and the fully developed CBL in the early afternoon were captured best. Correlation was highest during the late morning, when the high temporal resolution forecasts accurately followed the annotated CBL in the validation set. The breakdown of convective turbulence in the late afternoon (i.e., 16–19h UTC) was the main difficulty the models faced. The correlation analysis also showed that tracking MBLH was more difficult at night, where ceilometer-based detection has its limitations (e.g., see Sect. 2.2)." (see lines 348–352)

126.    line 285:

"Correlation was highest during daytime,"

You could even improve the performance by selecting the time of the fully developed ML (12-16UTC) when the top of the ML coincides with the top of the Aerosol layer topped by the free troposphere with low aerosol. This was at least the outcome of Schween et al (2014) who compared a wind lidar based retrieval with an aerosol backscatter based retrieval.

We have expanded the analysis, as suggested. In terms of MAE, performance during 8–12h UTC and 12–16h UTC was comparable. In terms of correlation, we obtained better results for the gradually increasing time series during 8–12h UTC.

See our response to the previous comments on how this section of the manuscript was changed.

127.    Table 3:

I am missing an information on how many data records where used (50 independent days with in total ~350thsnd records at 12sec resolution which are of course highly correlated, One could calculate a correlation length and from this an effective number of independent samples see Lenschow et al 1994)

The validation set was used for Tables 4 and 5. As mentioned previously in the manuscript, this set has 1396 samples. (see lines 247–249)

128.        Table 4:

is the mean squared error given in meters squared ?

The table with mean squared error results was removed from the manuscript. See our response to comment #125.

129.        Line 311:

"The Deep-Pathfinder methodology was robust against differences in annotation methods, leading to different results, but functioning appropriately regardless of the chosen dataset."

I do not understand: a different annotation method leads to different results but was appropriate?

Please explain in more detail:

What was the difference in the annotation technique?

What was different in the results?

Why were these differences acceptable?

There were a few differences in the annotation technique. Our annotations followed small-scale fluctuations in the RCS data closely at high temporal resolution, while the MeteoSwiss annotations were made once per minute and could be characterised as more stable over time. The settings for the temporal and spatial resolutions of the ceilometer were also different. Hence, for this experiment one pixel represented a period of 30 seconds and 15 metres of altitude.

The results followed the annotations. So, the model trained on MeteoSwiss annotations produced smoother profiles, that did not follow short-term fluctuations visible in the RCS data.

The following has been added to the manuscript:

"Our annotations followed small-scale fluctuations in the RCS data closely, while the externally sourced annotations were made once per minute and could be characterised as more stable over time. The settings for the temporal and spatial resolutions of the ceilometer were also different. Hence, for this experiment one pixel represented a duration of 30 seconds and 15 metres of altitude." (see lines 378–381)

130.        line 314:

"For different types of ceilometers (e.g., Vaisala CL31), it is recommended to repeat the unsupervised pre-training using unlabelled data of the corresponding

instrument. This should not be necessary when using Deep-Pathfinder at other locations with the same instrument type."

I would expect that in more complex environments than Cabauw (homogenous landscape over tens of kilometres) e.g. mountain valleys or coastal sites with a pronounced land-sea breeze system the Cabauw training will fail. But you have the experience with other dutch ceilometer data. You could give proofs for your statement.

Our statement is about model pre-training. The pre-training is attempting to reconstruct the ceilometer data and learn how to extract features from the data. Hence, for different ceilometer types (e.g., Vaisala), it should be repeated. However, capturing differences in annotations for mountain valleys should be possible during fine-tuning. Therefore, no change to this statement is necessary.

Further, as explained in our response to comment #26, the model was not trained only on Cabauw data. The unsupervised pre-training used data from five locations, while the supervised fine-tuning used data from three locations.

For clarification, this has now been described as follows in the manuscript:

"Model pre-training (see Sect. 2.4) used all ceilometer data from Cabauw, De Kooy, Groningen Airport Eelde, Maastricht Aachen Airport and Vlissingen. Model fine-tuning used a subset of data from Cabauw, De Kooy and De Bilt and some days outside the June 2020 to February 2022 period (e.g., for validation), which were manually annotated." (see lines 127–129)

*line 325: "leads to a grayscale output mask (see Fig. 8)."*

    131.       I guess this is the result of the method described following line 203.

        Figure 8 could thus go to that section to illustrate the method.

This is not the result, but the starting point of the MBLH extraction method. Since the figure does not show how the MBLH is extracted, we have updated the description of the extraction process instead (also see our response to comment #89).

For clarification, the following was changed in the manuscript:

"Note that model predictions display varying degrees of uncertainty, with pixel predictions being either black, white or any grey tone in between (i.e., the result of the pixel-wise sigmoid function at the end of the neural network). The loss function took into account model uncertainty by proportionally penalising the number of pixels below $p$ that were predicted as non-black, plus the pixels above $p$ that were predicted as non-white. MBLH at time $t$ was estimated as the value $\hat{p}$ that minimised this loss, multiplied by the spatial resolution of 10 metres." (see lines 266–271)

132. I am also not sure whether I understand fig 8: around 17-19UTC there are two distinct MLH can you explain what happens here and what the final MLH would be.

(Does deepPathfinder stick to the upper layer and at a certain moment realises MLH is lower and then from 17:45UTC on decides that previous MLHs must also be rather at 600m than at 1200m ?)

The predicted mask takes into account the temporal connections in the data, but sometimes this does not result in a clear separation of black and white in the output masks. After the output mask has been generated, MBLH extraction happens for each time step independently (see our response to the previous comment). Hence, Deep-Pathfinder initially follows the lower layer (i.e., from 17:00 UTC), but there is a jump between 18:30 and 19:00 to a higher layer, which is not correct. The value of the loss function will be higher in the red rectangle and allows to identify cases when predictions are less likely to be accurate.

133. line 326:

"value of the loss *function*"

Missing word

The word "function" has been added to the manuscript. (see line 400)

134. line 327: "Further quality control flags could be set if rain is detected,"

I thought you set MLH to zero if rain is detected.

The Deep-Pathfinder output should indeed be zero (e.g., see Fig. 7a). However, in case of precipitation that does not reach the ground (see comment #101), it can still be useful to have this quality control flag.

This section in the Discussion was rewritten as follows:

"Further, quality control flags could be set for specific atmospheric conditions, for example, so end users can exclude circumstances without vertical mixing. Other examples of useful flags are the occurrence of clouds (e.g., a binary indicator based on any detected cloud base height) and rain, since MBLH is not clearly defined during periods of precipitation." (see lines 401–404)

135. I would recommend the same for cloudy moments (see discussion of clouds in Kotthaus and Grimmond 2018 and Kotthaus et al 2023). Ceilometers deliver cloud base height that can be easily used for a cloud flag or when averaged over longer time periods as cloud fraction.

This suggestion has been added to the manuscript. See our response to the comment above for the changes that have been made to text.

136.     line 329:

"Instead of using only two output classes (i.e., mixing layer or not), image segmentation methods are suitable for the detection of multiple classes. ..."

This is indeed an interesting possibility and I guess you can give more details about your ideas how to approach this from ceilometer data alone (top of the Aerosol layer and residual layer), or with the help of other algorithms (e.g. PathfinderTURB, Poltera et al. 2017) or Manninen et al (2018), which of course would require a whole set of additonal data which, I guess, are all available at Cabauw (surface sensible heat flux, profiles of horizontal and vertical wind, cloud bases etc.).

We envision additional instruments would be required for such an extension, as not all these classes can be predicted using just ceilometer measurements.

The following has been added to the manuscript:

"In addition, ceilometer measurements can be accompanied with other data, such as profiles of horizontal and vertical wind speed from a Doppler wind lidar. We envision the input image to have an extra channel to capture both the ceilometer and wind lidar data. Hence, no major changes to the neural network architecture would be required, besides minor changes to the input and final layer." (see lines 409–412)

137.     Figure 9:

This is not your work. Did you asked for permission to publish it in your paper?

We have obtained permission to publish this figure in our paper.

For completeness, this email has been added as an additional file to our submission.

138.     line 337:

"U2-Net has so many skip connections that it would not be feasible to apply the pre-training approach used in our study."

I thought you cut all the skip connection for the pretraining in your model. Why not in U2net too ?

$U^2$-Net uses a nested U-structure (see the architecture in Figure R2). Each horizontal line represents a skip connection. So, there are skip connections between blocks, but also skip connections within blocks. By removing all of these connections, the network design becomes substantially different. Therefore, pre-training is expected to be less beneficial for $U^2$-Net.

[Figure]

Figure R2. U$^2$-Net architecture (Qin et al, 2020)

Although it is indeed technically feasible to remove all the skip connections, this approach may not be as suitable for pre-training.

The following change has been made in the manuscript:

"As U$^2$-Net uses a nested U-structure, it has so many skip connections that it would not be as suitable for the pre-training approach used in our study." (see lines 415–416)

Reference:

Qin, X., Zhang, Z., Huang, C., Dehghan, M., Zaiane, O. R., and Jagersand, M.: U$^2$-Net: Going deeper with nested U-structure for salient object detection, Pattern Recognition, 106, 107404, https://doi.org/10.1016/j.patcog.2020.107404, 2020.

139.      line 348:

"One challenge for model development is that no ground truth MLH data is available,"

But there are several different methods based on different physical parameters:

Radiosonde based temperature profiles, radar wind profilers based turbulence profiles, doppler lidar based wind profiles, etc. All of them could be used for intercomparison.

The point we were trying to make here is that in computer science applications using supervised learning the target variable is generally known. However, in this case, it has to be assessed whether the quality of the target variable is good based on an intercomparison with other methods.

The following changes have been made to the manuscript:

"One challenge for model development is that no ground truth MBLH data is available, although the quality of the labels can be assessed based on different physical parameters (e.g., radiosonde based temperature profiles, radar wind profilers based turbulence profiles, Doppler lidar based wind profiles). This complicates method intercomparison." (see lines 426–429)

> 140.      line 352:
>
> "However, manual feature engineering based on expert decisions is avoided,"
>
> But you used for the NN architecture intercomparison only learning based on annotated data. Annotating is highly based on expert knowledge: the annotating person knows (or learns quickly) during which time of the day which behaviour of the MLH can be expected, what to do in the case of clouds or rain etc.

We mean that during the modelling phase no additional expert knowledge is required. It is acknowledged that the annotation requires assumptions, which is described in the preceding sentence.

For clarification, the manuscript was revised as follows:

"The initial structured annotation process (see Sect. 2.2) involves assumptions to determine the exact location of the MBLH. However, in the modelling phase manual feature engineering is avoided, as the mapping between input and label is learned directly from large-scale data." (see lines 430–433)

> 141.      line 358:
>
> "The availability of real-time MLH estimates from a large-scale ceilometer network could be used for the advancement of NWP models via data assimilation."
>
> I am not sure whether MLH can be easily assimilated into NWP. Do you have references for this,

This sentence has been revised to:

"The availability of real-time MBLH estimates from a large-scale ceilometer network could be used for the advancement of NWP models." (see lines 438–439)
* * *
References provided by reviewer 2:

Caicedo et al 2017

  Comparison of aerosol lidar retrieval methods for boundary layer height detection using ceilometer aerosol backscatter data

  Atmos. Meas. Tech., 10, 1609-1622, 2017

  doi:10.5194/amt-10-1609-2017

Couvreux et al 2007

  Negative water vapour skewness and dry tongues in the convective boundary layer: observations and large-eddy simulation budget analysis

  Boundary-Layer Meteorol (2007) 123:269-294

  DOI 10.1007/s10546-006-9140-y

Eeresmaa et al 2012

  A Three-Step Method for Estimating the Mixing Height Using Ceilometer Data from the Helsinki Testbed

  JOURNAL OF APPLIED METEOROLOGY AND CLIMATOLOGY 51 pp2172

  DOI: 10.1175/JAMC-D-12-058.1

Fitzgerald 1984

  Effect of relative humidity on the aerosol backscattering coefficient at 0.694- and 10.6-μm wavelengths

  Applied Optics Vol. 23, Issue 3, pp. 411-418 (1984)

  DOI: 10.1364/AO.23.000411

*Frey et al. 2010*

*Detection of aerosol layers with ceilometers and the recognition of the mixed layer depth*

*Extended abstract on poster presented at 'International Symposium for the Advancement of Boundary Layer Remote Sensing' (ISARS), June 2010 at Paris, P-BLS/12)*

*http://www.isars2010.uvsq.fr/index.php?option=com_content&view=article&id=42&Itemid=36*

*http://www.isars2010.uvsq.fr/images/stories/PosterExtAbstracts/P_BLS12_Frey.pdf*

*Garratt 1992*

*The Atmospheric Boundary Layer*

*Cambridge University Press*

*ISBN 0 521 46745 4*

*Haeffelin et al. 2011*

*Evaluation of Mixing-Height Retrievals from Automatic Profiling Lidars and Ceilometers in View of Future Integrated Networks in Europe*

*Boundary-Layer Meteorol*

*DOI 10.1007/s10546-011-9643-z*

*Hennemuth and Lammert 2006*

*DETERMINATION OF THE ATMOSPHERIC BOUNDARY LAYER HEIGHT FROM RADIOSONDE AND LIDAR BACKSCATTER*

*Boundary-Layer Meteorology (2006) 120: 181-200*

*DOI 10.1007/s10546-005-9035-3*

*Hervo et al 2016*

An empirical method to correct for temperature-dependent variations in the overlap function of CHM15k ceilometers

Atmos. Meas. Tech., 9, 2947-2959, 2016

doi:10.5194/amt-9-2947-2016

Kotthaus et al 2023

Atmospheric boundary layer height from ground-based remote sensing: a review of capabilities and limitations

Atmos. Meas. Tech., 16, 433-479, 2023

https://doi.org/10.5194/amt-16-433-2023

Lenschow et al 1994

How Long Is Long Enough When Measuring Fluxes and Other Turbulence Statistics?

Journal of Atmospheric and Oceanic Technology 11(3):661-673

DOI: 10.1175/1520-0426(1994)011<0661:HLILEW>2.0.CO;2

Muenkel et al 2007

Retrieval of mixing height and dust concentration with lidar ceilometer

Boundary-Layer Meteorol (2007) 124:117-128

DOI 10.1007/s10546-006-9103-3

Ouwersloot and Vilà-Guerau de Arellano 2013

Analytical Solution for the Convectively-Mixed Atmospheric Boundary Layer

Boundary-Layer Meteorology

DOI 10.1007/s10546-013-9816-z

Poltera et al 2017:

*PathfinderTURB: an automatic boundary layer algorithm. Development, validation and application to study the impact on in situ measurements at the Jungfraujoch,*

*Atmospheric Chemistry and Physics, 17, 10 051-10 070,*

*DOI: 10.5194/acp-17-10051-2017*

*Schween et al 2014*

*Mixing-layer height retrieval with ceilometer and Doppler lidar: from case studies to long-term assessment*

*Atmos. Meas. Tech., 7, 3685-3704, 2014*

*doi:10.5194/amt-7-3685-2014*

*Stull, 1988*

*An Introduction to Boundary Layer Meteorology,*

*Atmospheric and Oceanographic Sciences Library, Springer Netherlands,*

*ISBN 978-94-009-3027-8*

*Teschke and Poenitz 2010:*

*On the Retrieval of Aerosol (Mixing) Layer Heights on the Basis of Ceilometer Data*

*Extended abstract on talk presented at 'International Symposium for the Advancement of Boundary Layer Remote Sensing' (ISARS), June 2010 at Paris, P-BLS/12)*

*http://www.isars2010.uvsq.fr/index.php?option=com_content&view=article&id=41&Itemid=35*

*http://www.isars2010.uvsq.fr/images/stories/PosterExtAbstracts/P_RET07_Ponitz.pdf*

*Traeumner et al 2011*

*Convective Boundary-Layer Entrainment: Short Review and Progress using Doppler Lidar*

*Boundary-Layer Meteorol (2011) 141:369-391*

*DOI 10.1007/s10546-011-9657-6*

*Tuomi 1976*

*Backscatter of light by Aerosol at high relative humdity*

*J. Aerosol Sci., 1976, Vol. 7, pp. 463 to 471.*

*DOI: 10.1016/0021-8502(76)90051-3*

*van Ulden and Wieringa 1996*

*Atmospheric boundary layer research at Cabauw*

*Boundary-Layer Meteorol 78, 39-69*

*DOI: 10.1007/BF00122486*

*Wiegner and Geiss 2012*

*Aerosol profiling with the Jenoptik ceilometer CHM15kx*

*Atmos. Meas. Tech., 5, 1953-1964, 2012*

*doi:10.5194/amt-5-1953-2012*

---

## Author Response (AR2)

**Manuscript**: amt-2023-80
**Title**: Deep-Pathfinder: A boundary layer height detection algorithm based on image segmentation
**Revision:** round 2

Dear Reviewers:

Thank you for your time and effort to carefully assess the revised version of the manuscript. In this second round, the manuscript has been thoroughly revised based on all remaining comments.

The following pages provide our detailed responses.

Kind regards,
The authors

**Reviewer 1**

Summary: *After the review, this paper improved significantly. I recommend to publish.*

Thank you very much for your feedback.

**Reviewer 2 (report #3)**

General comment: *The Authors explain a lot to the reviewer, but not to the reader of the paper, i.e. they do not put their arguments in the revised paper. Several changes are mentioned several times in the reply. This is of course ok if it fits, but one may get the impression that a lot of changes happened. Most of the changes are adaptions within the sentences, often just a few words.*

*I am mostly satisfied with the adaptions around the description of the algorithm and the training, fine-tuning and validation. What has been also clarified is what part of the dataset has been used for what. And i think the reorganization of section 2 is also helpful.*

Thank you very much for your feedback.

A point-by-point response to your remaining feedback is provided below.

1. Clouds (point 5, ):

   I suggested that the authors should repeat the Deep-Pathfinder - STRATFinder comparison with no or few clouds to investigate whether clouds have an impact on the retrievals. This could give insights how to deal with clouds in BLH retrievals. The authors provide a longer detailed discussion to the reviewer based on the case of fig.6e, but instead of explaining this in the articles text they add just one sentence to the manuscript. The authors miss the opportunity to prove or disprove on a statistical basis that clouds might pose a problem.

The suggested analysis has now been performed. For each day in the test set, the percentage of time was computed that clouds were present in the input image. This is referred to as the cloud overcast fraction. Subsequently, Deep-Pathfinder and STRATfinder performance was compared for different ranges of cloud cover.

The following text and table with statistics have been added to the manuscript:

"These daily fluctuations can be partly explained by the amount of cloud cover. To illustrate this point, the daily cloud overcast fractions were computed for all dates in the test set, looking only for clouds below 2245 meters (i.e., the vertical range captured by our model). Table 3 shows that on days with no or few clouds the Deep-Pathfinder and STRATfinder algorithms were more closely aligned, based on Pearson correlation and mean absolute difference statistics." (see lines 309–312)

**Table 3.** Comparison of Deep-Pathfinder and STRATfinder estimates for different ranges of cloud cover in the test set.

|  | Overall | Cloud overcast range | | |
|---|---|---|---|---|
|  |  | 0–10% | 10–30% | 30–100% |
| Number of days | 161 | 27 | 24 | 110 |
| Pearson correlation | 0.706 | 0.811 | 0.819 | 0.632 |
| Mean absolute difference (m) | 189.0 | 141.6 | 176.7 | 205.3 |

Finally, we indeed did not list all changes we made to the manuscript during the first revision in the response to comment #5. This could have been stated more clearly. For completeness, the changes describing the most important observations related to the case of Fig. 7e are listed below.

"In case of multiple cloud layers, our annotations typically followed the lower layer, while STRATfinder followed the higher layer. For example, this behaviour can be observed in Fig. 7e." (see lines 322–323)

"The example of Fig. 7e shows that for multiple cloud layers Deep-Pathfinder and STRATfinder typically followed a different layer. Hence, in case of multiple cloud layers, users should be aware that the methods may produce different MBLH estimates." (see lines 298–300)

2. Comment number 6:

   This is my remark that //there is no night time mixing layer// which was also brought forward by reviewer 1. I repeated this at several places in my comments. The Authors somehow agreed on that - but do not mention the RH/backscatter problem in the revised manuscript. And they refused to drop the data nor change the naming ('...mixing layer...' ) because they wanted to be "consistent with former papers." In other words they agree that it has been done wrong in the past but they want to go on with that.

   They discussed at no place the argumentation that with high relative humidity aerosol particle grow in size, backscatter increases and that this may result in a layer detection where no layer is. They could say that this mechanism exists and may result in faulty layer attribution - but they did not. In my opinion this argument must go in the paper. It should be noted that this high humidity layer does not even necessarily falls together with the stable nighttime surface inversion.
   If they want they can state in the paper that further investigation is beyond the scope of the paper - as they wrote somewhere in their reply. But this in an essential problem with backscatter based layer detection and it must be suspected that also artificial intelligence and computional vision is not able to get around this problem. The authors should discuss this.

First, we would like to mention that as part of the first revision, we incorporated in the manuscript that there is no mixing at night. This now reads as follows:

"During nighttime, the pollution rich layers may drop to very low altitudes into the incomplete-overlap region of the CHM15k ceilometer and vertical mixing in fact predominantly ceases to exist. However, the concentration levels of pollutants remain layered and, therefore, Cabauw mast measurements were used to aid in the identification of the presence and height of nocturnal layers." (see lines 176–179)

The following has now also been added to the manuscript:

"Finally, with high relative humidity aerosol particles grow in size, leading to increased backscatter which may result in a layer detection where no layer is (i.e., faulty layer attribution). The high humidity layer also does not necessarily coincide with the stable nighttime surface inversion, meaning MBLH retrieval during night by use of ceilometer backscatter data can be strongly biased. Further investigation of these mechanisms is beyond the scope of this paper. For further reading, refer to Kotthaus et al. (2023, section 3.3.2 and references therein) or Collaud Coen et al. (2014)." (see lines 54–59)

References

Kotthaus, S., Bravo-Aranda, J. A., Collaud Coen, M., Guerrero-Rascado, J. L., Costa, M. J., Cimini, D., O'Connor, E. J., Hervo, M., Alados-Arboledas, L., Jiménez-Portaz, M., Mona, L., Ruffieux, D., Illingworth, A., and Haeffelin, M.: Atmospheric boundary layer height from ground-based remote sensing: a review of capabilities and limitations, Atmospheric Measurement Techniques, 16, 433–479, https://doi.org/10.5194/amt-16-433-2023, 2023.

Collaud Coen, M., Praz, C., Haefele, A., Ruffieux, D., Kaufmann, P., and Calpini, B.: Determination and climatology of the planetary boundary layer height above the Swiss plateau by in situ and remote sensing measurements as well as by the COSMO-2 model, Atmospheric Chemistry and Physics, 14, 13 205–13 221, https://doi.org/10.5194/acp-14-13205-2014, 2014.

3. Figure R1:

   The authors provided a figure R1 in the response showing the course of backscatter, Temperature, specific humidity etc. from the Cabauw tower for one of the cases they presented in fig.6. I suggested in my review that they should investigate these parameters because I hoped that it would become obvious to them that their night time BLH is not correct. They present the plot and say in the response just very short:

   > "we can see that in this example the manual annotation was mostly in line with thermodynamic definitions.
   > Note that this figure has not been added to the manuscript."

   No arguments or explanation why they think that this plots support their retrieval. They changed the text, but they just changed one single word i.e. //correctly// to //as intended// (why do they talk here about their 'manual annotation' and not about the retrieval ?)

   But there is more than a missing argument.
   The temperature axis in this figure is annotated with:

   > "Potential Temperature * [K] *(my own estimative using metpy) "

   I.e. the author was not sure whether it is correct. I am sure he can find a colleague at KNMI who can support him in finding an exact instead 'estimative' approach.

If you ask me that cannot be Potential Temperature (Tpot) but instead it is rather air temperature (Tair) in Kelvin.

In the first hours 00-03 UTC , i.e. in the middle of the night, the temperature at 2m (grayish) is by more than 1K warmer than at 200m (blue). Even during daytime in summer under clear sky that would be an exceptional unstable stratification. If I would believe that this is Tpot I would expect a *mixing*layer to reach several thousand meter, but not only 300m as found by the retrievals. If you assume it is just Tair in K and add as an estimate deltaT from the adiabatic gradient (deltaT = +z*1K/100m) to calculate Tpot you get stable stratification with a stronger gradient at the surface - as you would expect during night. At about 3:30 this gradient inverts (advection of cold and dry air) - from then on T(2m) (grayish) is colder than T(200m) (blue) and T(140m) (orange) - i.e. if we take this temperature as Tpot we have stable stratification, no mixing and thus mixing layer height = 0m. And if we assume that this is not Tpot and add deltaT the gradient becomes even larger. But the BL retrievals find a BL of 150-170m - way higher.

I could go on with arguments like this for the whole day ...

So: the authors cannot use this plot as an argument that their retrieval works well - not if it would be Tpot and not if it is, as i suspect, Tair.

It is obvious that Temperature presented in figure R1 can not be potential temperature, but is rather something at least close to air temperature. Accordingly their statement that "...manual annotation was mostly in line with thermodynamic definitions. " is not supported by this figure. It rather supports the reviewers argument that a boundary layer height (BLH) retrieval during night by use of ceilometer backscatter data can be strongly biased. The reason for this is probably the RH-aerosol growth-backscatter mechanism described by the reviewer.
They should discuss this argument in the manuscript.

Figure R1 indeed contained a mistake related to the potential temperature. We thank the reviewer for noticing this and for the opportunity to correct this figure. For the record, a corrected version of this figure is added below (figure R1_v2).

*This figure has not been updated in the manuscript, as it does not appear in the manuscript (i.e., it only appeared in the response to reviewers document).*

[Figure]

Figure R1_v2: Case study at Cabauw on 2020-12-10. From top to bottom: RCS data with Deep-Pathfinder (black line) and STRATfinder (orange line) retrievals, potential temperature, specific humidity, and short wave downward radiation.

To resolve this comment, the discussion of Fig. 7f in the manuscript has been updated as follows:

"When a clear CBL was not apparent (e.g., Fig. 7f), Deep-Pathfinder and STRATfinder obtained similar estimates, although in Fig. 7f both were far above the stable nighttime surface inversion." (see lines 300–301)

Finally, the reviewer suggests to discuss that a boundary layer height retrieval during night by use of ceilometer backscatter data can be strongly biased, and that this may be caused by the RH-aerosol growth-backscatter mechanism.

As also noted in the response to comment #2, the following has now been added to the manuscript:

"Finally, with high relative humidity aerosol particles grow in size, leading to increased backscatter which may result in a layer detection where no layer is (i.e., faulty layer attribution). The high humidity layer also does not necessarily coincide with the stable nighttime surface inversion, meaning MBLH retrieval during night by use of ceilometer backscatter data can be strongly biased. Further investigation of these mechanisms is beyond the scope of this paper. For further reading, refer to Kotthaus et al. (2023, section 3.3.2 and references therein) or Collaud Coen et al. (2014)." (see lines 54–59)

References

Kotthaus, S., Bravo-Aranda, J. A., Collaud Coen, M., Guerrero-Rascado, J. L., Costa, M. J., Cimini, D., O'Connor, E. J., Hervo, M., Alados-Arboledas, L., Jiménez-Portaz, M., Mona, L., Ruffieux, D., Illingworth, A., and Haeffelin, M.: Atmospheric boundary layer height from ground-based remote sensing: a review of capabilities and limitations, Atmospheric Measurement Techniques, 16, 433–479, https://doi.org/10.5194/amt-16-433-2023, 2023.

Collaud Coen, M., Praz, C., Haefele, A., Ruffieux, D., Kaufmann, P., and Calpini, B.: Determination and climatology of the planetary boundary layer height above the Swiss plateau by in situ and remote sensing measurements as well as by the COSMO-2 model, Atmospheric Chemistry and Physics, 14, 13 205–13 221, https://doi.org/10.5194/acp-14-13205-2014, 2014.

**Reviewer 3 (report #2)**

Summary: *I believe the authors have answered most of the reviewers' comments, and the revised manuscript is substantially improved with respect to the first version.*

Thank you very much for your feedback.

A point-by-point response to your comments is provided below.

1. However, I feel the authors have not clearly addressed comment #7 of Reviewer 2.

   In particular, I see a potential contradiction in their breakout answers, i.e. "We do not claim that we apply an overlap correction", and then "with instrument-specific overlap correction".

   The authors should remove the potential contradiction and clarify their arguments.

The cause of this confusion is that there are two types of overlap corrections. First, the CHM15k instrument has a built-in overlap correction. This first overlap correction is not perfect, as also noted by reviewer 2 in comment #7. Second, some studies therefore use an additional overlap correction when processing the data from the ceilometer (e.g., Hervo et al., 2016). For clarification: we use the built-in overlap correction, but no additional overlap correction.

To clarify our initial response to comment #7, this should be rephrased as: "Besides the built-in overlap correction, we do not claim that we apply an additional overlap correction"

Further, the following sentence has been inserted in the corresponding paragraph of the manuscript for clarification:

"Note that prior research has indicated that the built-in overlap correction of the CHM15k is not perfect (Hervo et al., 2016)." (see lines 126–127)

Reference:

Hervo, M., Poltera, Y., and Haefele, A.: An empirical method to correct for temperature-dependent variations in the overlap function of CHM15k ceilometers, Atmospheric Measurement Techniques, 9, 2947–2959, https://doi.org/10.5194/amt-9-2947-2016, 2016.

Minor comments:

2. In addition to Milroy et al. (2012), I would suggest the authors refer to Collaud et al. 2014 for insightful discussion on the features of MBLH estimated from different instruments (openly accessible at: www.atmos-chem-phys.net/14/13205/2014/)

The suggested reference has been added to the manuscript:

"These methods are complementary, as each approach has different advantages and limitations for capturing certain features of the MBLH (Collaud Coen et al., 2014)." (see lines 37–39)

Reference:

Collaud Coen, M., Praz, C., Haefele, A., Ruffieux, D., Kaufmann, P., and Calpini, B.: Determination and climatology of the planetary boundary layer height above the Swiss plateau by in situ and remote sensing measurements as well as by the COSMO-2 model, Atmospheric Chemistry and Physics, 14, 13 205–13 221, https://doi.org/10.5194/acp-14-13205-2014, 2014.

3. Table 3: In addition to UTC, I suggest to explicit local standard time (LST) as well, as the latter is more meaningful for the diurnal cycle.

The following has been added to the caption of this table:

"Time of day is stated in UTC; note that the local standard time at Cabauw is UTC+1 or UTC+2 (daylight saving time)." (see page 16)